# MIRAGE: A Benchmark for Multimodal Information-Seeking and Reasoning in Agricultural Expert-Guided Conversations

Vardhan Dongre[1*]    Chi Gui[1*]    Shubham Garg[2]    Hooshang Nayyeri[2]
Gokhan Tur[1]    Dilek Hakkani-Tür[1]    Vikram S. Adve[1]
[1]University of Illinois Urbana-Champaign    [2]Amazon

## Abstract

We introduce MIRAGE, a new benchmark for multimodal expert-level reasoning and decision-making in consultative interaction settings. Designed for the agriculture domain, MIRAGE captures the full complexity of expert consultations by combining natural user queries, expert-authored responses, and image-based context, offering a high-fidelity benchmark for evaluating models on grounded reasoning, clarification strategies, and long-form generation in a real-world, knowledge-intensive domain. Grounded in over 35,000 real user-expert interactions and curated through a carefully designed multi-step pipeline, MIRAGE spans diverse crop health, pest diagnosis, and crop management scenarios. The benchmark includes more than 7,000 unique biological entities, covering plant species, pests, and diseases, making it one of the most taxonomically diverse benchmarks available for vision-language models, grounded in the real world. Unlike existing benchmarks that rely on well-specified user inputs and closed-set taxonomies, MIRAGE features underspecified, context-rich scenarios with open-world settings, requiring models to infer latent knowledge gaps, handle rare entities, and either proactively guide the interaction or respond. We evaluate more than 20 closed and open-source frontier vision-language models (VLMs), using an ensemble of reasoning language models as evaluators, highlighting the significant challenges posed by MIRAGE. Despite strong performance on conventional benchmarks, state-of-the-art VLMs struggle on MIRAGE, particularly in scenarios encountering rare entities and addressing open-ended user requests. To support model development, we fine-tune Qwen2.5-VL models on MIRAGE, observing measurable performance gains and demonstrating MIRAGE's potential as both a benchmark and a development suite for in-domain visual reasoning and conversational decision-making in real-world settings. Our dataset [1] and code [2] are all publicly available. Project Page: https://mirage-benchmark.github.io/

## 1 Introduction

Advances in large vision–language models (LVLMs) have significantly improved AI's ability to interpret images and generate natural-language responses. However, existing benchmarks predominantly focus on short-form visual question answering [69, 30, 12, 52, 72], captioning [54, 51], or grounded generation under constrained contexts [29]. These tasks fall short of capturing the interactive, decision-centric nature of real-world expert consultations, where users often present open-ended,

---

*Equal Contribution
[1]Dataset: https://huggingface.co/MIRAGE-Benchmark
[2]Code: https://github.com/MIRAGE-Benchmark/MIRAGE-Benchmark

ambiguous, and visually grounded queries. In knowledge-intensive domains such as agriculture, medicine, and engineering, expert consultations inherently span multiple modalities [50] and success hinges not just on perception or language fluency, but on the ability to reason causally, handle missing context, and make interaction-level decisions. This complex interplay between multimodal understanding and contextual reasoning in professional settings represents a significant gap in the current LVLM evaluation frameworks. LLMs and LVLMs are increasingly used in domains like healthcare, law, and plant care [58, 60, 38, 73, 17, 49, 64], yet current benchmarks underrepresent knowledge-intensive scenarios involving ambiguous, multimodal queries [30]. Agriculture illustrates this gap: farmers often seek image-based, expert-level guidance during critical events, but inaccurate or unsupported model outputs [70, 40, 68] can lead to serious consequences. This highlights the need for rigorous evaluation frameworks that assess VLMs in more realistic settings.

To address this need, we introduce MIRAGE, a comprehensive benchmark designed around four core principles. First, **underspecification**: unlike traditional VQA and multimodal understanding datasets, MIRAGE presents user turns with latent knowledge gaps, requiring inference of missing context. Second, **multimodality**: each task combines natural language, images, and real-world metadata like location and time, reflecting the inputs experts receive. Third, **decision-making**: MIRAGE evaluates not only factual accuracy but also a model's ability to simulate expert conversational behavior by deciding whether to ask clarifying questions or provide actionable answers. Lastly, **domain grounding**: built from real expert-user conversations, MIRAGE ensures ecological validity and high relevance for agricultural consultation tasks. It features problems sourced from 37,512 carefully selected high-quality user-expert conversations distilled from a corpus of 218,000 interactions collected between 2012 and 2025. Spanning more than 7000 unique biological entities (see Table 4) across plants, pests, and diseases, we evaluate models in two unique challenges absent in current benchmarks (Figure 1) A.) **MIRAGE-MMST** (Multimodal Single-Turn): Given a user query and associated images(s), can a model identify key biological entities, reason about causal symptoms, and produce actionable management recommendations? B.) **MIRAGE-MMMT** (Multimodal Multi-Turn): In an ongoing conversation, can a model decide whether to seek clarification or respond, and generate the appropriate follow-up utterance?

We evaluate 22 SOTA proprietary and open-source LVLMs covering 6 model families on MIRAGE. Our key contributions and findings are summarized below:

- MIRAGE-MMST is highly challenging: Even GPT-4.1 achieves only 43.9% Identification Accuracy.

- There is a pronounced disparity in performance between open-source LVLMs and GPT-4.1. The highest performing open-source models, such as Qwen2.5-VL-72B achieve approximately 29.8% in Identification accuracy and 2.47 (out of 4) reasoning score.

- We introduce a novel evaluation framework using an ensemble of reasoning-capable LLMs as judges, enabling interpretable, reproducible scoring across fine-grained criteria, including Accuracy, Relevance, Completeness, and Diagnostic Parsimony.

- MIRAGE exposes a substantial generalization gap: Even after LoRA fine-tuning, models like Qwen2.5-VL-3B achieve up to 28.4% accuracy on seen entities, but only 14.6% on unseen entities, revealing a persistent 14-point gap that underscores the difficulty of open-world generalization in long-tail settings.

- Decision-making under partial observability of user goals remains difficult: On MIRAGE-MMMT, even top models achieve only 63% decision accuracy, with frequent errors in determining when clarification is necessary.

  We believe MIRAGE will serve as a valuable benchmark for building next-generation multimodal assistants that are not only perceptually grounded but also capable of contextual reasoning, eliciting missing contextual information, and cautious recommendations in complex real-world settings.

## 2 The MIRAGE Benchmark

We introduce MIRAGE (**M**ultimodal **I**nformation-seeking and **R**easoning in **AG**ricultural **E**xpert-Guided conversations), a benchmark purpose-built to evaluate vision-language models on expert-level reasoning and decision-making in real-world in-domain consultations. It is a multimodal benchmark

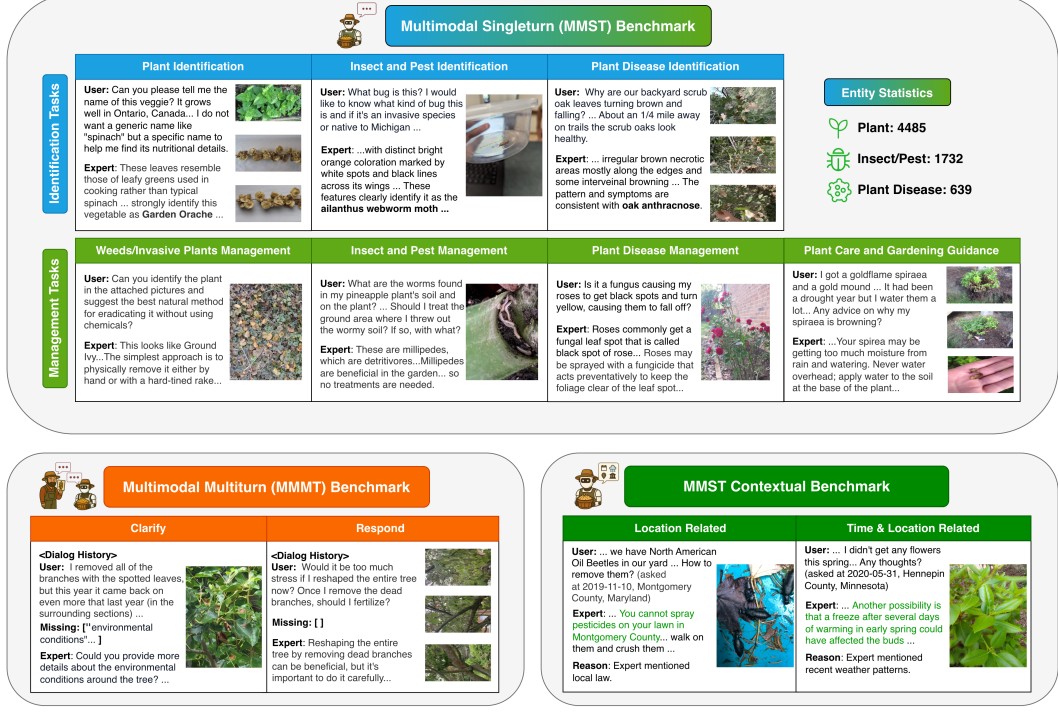

Figure 1: An overview of the MIRAGE benchmark, detailing its components. The benchmark includes: (1) The Multimodal Singleturn (MMST) Benchmark, with 8,184 interactions featuring 6,856 biological entities across seven agronomic categories. (2) The Multimodal Multiturn (MMMT) Benchmark, a corpus of 861 dialogues for evaluating 'clarify-or-respond' decision-making. Additionally, MIRAGE contains the MMST Contextual Benchmark, a specialized single-turn set of 3,934 interactions where expert responses are related to time and location metadata.

derived from over 200,000 raw interactions between real users and certified agronomy experts from AskExtension [3]. In addition to supporting studies on agricultural and knowledge-intensive LVLMs, MIRAGE uniquely assesses models across multimodal perception, causal reasoning, and clarification strategy in realistic, user-initiated scenarios.

## 2.1 Overview of MIRAGE

MIRAGE comprises two components: MMST, with a total corpus of **over 29,000** single-turn expert-user interactions, and MMMT, a multi-turn corpus of **6,306** dialogues ( 12,000 turns) requiring clarify-or-respond decisions. Figure 1 provides a visual overview of the evaluation benchmarks derived from this corpus. Tasks span identification, diagnostic reasoning, and issue management guidance across seven agronomic categories (Table 4). The MMST evaluation benchmark includes **Standard (8,184 interactions)** and **Contextual (3,934 interactions)** subsets (Section 3, Appendix C), while the MMMT benchmark contains 861 dialogues. A fine-grained evaluation framework assesses both accuracy and utility. With over 7,600 biological entities in the full corpus (6,856 in the benchmark set), including thousands of unseen species in the evaluation set, MIRAGE introduces a challenging open-world generalization setting for LVLMs. Refer Appendix C for more details.

## 2.2 Benchmark Comparison and Positioning

MIRAGE builds on a growing body of benchmarks in agriculture and multimodal reasoning, yet fills a critical gap left by prior efforts. Datasets like MMMU [69] focus on generalist multimodal reasoning but rely on constrained multiple-choice formats. AgMMU [30] targets agriculture, yet

---

[3]Ask Extension is an online service that connects the public with research-based answers from Cooperative Extension and university experts at U.S. Land-Grant institutions, offering guidance on agriculture, gardening, food safety, and more through a simple online form. https://ask.extension.org/

| Dataset | Type | Multimodal | Training Set | Expert Domain | Factuality | Expert Authored | Multi Turn |
|---|---|---|---|---|---|---|---|
| *iNat21* [59] | CLS | ✗ | ✓ | ✓ | – | – | – |
| *TreeOfLife* [55] | CLS | ✓ | ✓ | ✓ | – | – | – |
| *SimpleQA* [62] | OEQ | ✗ | ✗ | ✗ | ✓ | – | – |
| *MMMU* [69] | MCQ | ✓ | ✗ | ✓ | ✗ | ✗ | ✗ |
| *AgMMU* [30] | MCQ+OEQ | ✓ | ✓ | ✓ | ✓ | ✗ | ✗ |
| *CROP* [71] | LFQ/Conv | ✗ | ✓ | ✓ | ✓ | ✓ | ✓ |
| **MIRAGE** | LFQ/Conv | ✓ | ✓ | ✓ | ✓ | ✓ | ✓ |

Table 1: Comparison of MIRAGE with existing benchmarks across key characteristics. *CLS* = Classification, *MCQ* = Multiple Choice Questions, *OEQ* = Open-Ended Questions, *LFQ* = Long-Form Question Answering, *Conv* = Conversational multi-turn interactions. MIRAGE uniquely combines multimodality, expert-authored long-form responses, and multi-turn conversations grounded in factual and domain-specific reasoning.

uses synthetic short-form MCQs that are not representative of real expert-user interactions. Domain-specific datasets such as TreeOfLife [55] and iNat21 [59] emphasize fine-grained image classification but operate in closed-world, non-interactive settings. The CROP benchmark [71] introduces multi-turn crop science QA, but is limited to two crops, lacks visual input, and is fully text-based. In contrast, MIRAGE is the only benchmark constructed from large-scale, real-world agricultural consultations and supports both multimodal single-turn and multi-turn tasks grounded in naturally underspecified user queries. As summarized in Table 1, MIRAGE includes contextual question answering, real user-submitted images with varied quality and lighting, and domain-expert-authored responses as ground truth. It spans diverse query types—including identification, causal reasoning, and management—and introduces goal-state modeling, clarify-or-respond decision policies, and a multidimensional evaluation framework that moves beyond correctness to assess identification accuracy, causal justification, response quality, and diagnostic parsimony. These features make MIRAGE a uniquely realistic and challenging testbed for evaluating LVLMs in real-world consultation scenarios.

# 3 MIRAGE-MMST: Multimodal Singleturn Benchmark 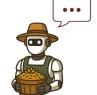

**MIRAGE-MMST** is a benchmark designed to assess expert-level, single-turn reasoning in multimodal agricultural consultations. The task setup is similar to a Long-form VQA task [37]. Each instance consists of a natural language question paired with one or more user-provided images and associated metadata (e.g., timestamp, location). Each instance consists of a natural language question $q$, an associated image set $I = \{i_1, \ldots, i_m\}$, and metadata meta $\in \mathcal{M}$. Formally, a single-turn instance is represented as a triplet $(q, I, \text{meta}) \in \mathcal{Q} \times \mathcal{I}^m \times \mathcal{M}$, and the model must generate a structured response $r = (e, c \vee m)$, where $e$ denotes identified entities (e.g., crop, pest, disease), $c$ is a causal explanation, and $m$ is a management recommendation, if requested. The task evaluates the model's ability to reason causally about visual symptoms, identify relevant agronomic entities, and, when prompted, generate detailed management recommendations grounded in the observed evidence.

To support varying levels of difficulty and contextual grounding, MIRAGE-MMST is divided into two subsets: a Standard subset, consisting of self-contained questions that can be answered using only the provided text and image, and a Contextual subset, where successful interpretation depends on implicit information such as time, location, or agricultural context not present in the input. The Standard subset further includes two task types—MMST-ID (Identification), which focuses on visual reasoning and recognizing entities, and MMST-MG (Management), which involves reasoning and generating recommendation-based responses. In contrast, the Contextual subset contains queries with latent information gaps and elliptical language, requiring models to reconstruct missing context using external priors. We first manually annotated a seed set of contextual examples and then adopted an automated classifier to separate the full dataset. See Appendix C.3 for details.

MIRAGE-MMST serves as a diagnostic tool to benchmark the capabilities of language and vision-language models in grounded agricultural reasoning. Evaluation focuses on the model's ability to perform fine-grained entity recognition, generate causal explanations, and provide accurate and contextually appropriate management advice, when applicable.

## 3.1 Dataset Curation

Our dataset was collected from Ask Extension[4], an online platform where users submit agricultural queries and receive expert responses from trained volunteers and professionals. We aggregated 218K single-turn dialogues from December 2012 to April 2025. To ensure high dataset quality and relevance, we implemented a rigorous four-step curation process as shown in Figure 5 (1) *Data Cleaning*, removing incomplete or unsuitable dialogues; (2) *Data Categorization*, organizing dialogues into two primary subsets, Subset 1: Identification (Plant Identification, Insect and Pest Identification, Plant Disease Identification) and Subset 2: Management (Plant Disease Management, Insect and Pest Management, Plant Care and Gardening Guidance, Weeds/Invasive Plants Management), extracting and collecting main entity name, and labeling context-aware responses; (3) *Data Reformatting*, involving the removal of personal information, enhancing questions with supplementary content from expert-provided URLs using GPT-4.1, and reconstructing Identification task answers with visual enhancements through GPT-4.1-mini to ensure consistent formatting, detailed visual descriptions, and comprehensive reasoning chains; and (4) *Data Splitting*, partitioning the curated dataset into Standard Benchmark (8,184 dialogues), Standard Training (17,532 dialogues), and Contextual Benchmark (3,934 dialogues) based on contextual labeling. Further curation details are provided in the Appendix C.2.

# 4 MIRAGE-MMMT: Multimodal Multiturn Benchmark

**MIRAGE-MMMT** is a multimodal decision-making task, grounded in real-world agricultural consultations. Users pose complex, often image-supported questions about plant health, pest identification, growing conditions, and other agronomic concerns. Each dialogue reflects a practical scenario in which the expert must reason over conversation history and visual context to decide: (1) whether to respond with guidance based on what is known, or (2) whether to pause and seek additional input to resolve a knowledge gap. This introduces a decision-making challenge tightly coupled with natural language generation.

**Task Formulation:** We formalize this as a joint decision making-generation task: Given a multi-turn dialogue $D = \{(s_1, u_1), \ldots, (s_n, u_n)\}$, where $s_i \in \{\texttt{user}, \texttt{expert}\}$ and $u_i$ are utterances, and an associated image set $I = \{i_1, \ldots, i_m\}$, the model must jointly predict a decision $a \in \{\text{Clarify}, \text{Respond}\}$ and generate the corresponding utterance $r$.

The model infers the user's goal $G$ and a goal state $S_G = (\texttt{known}, \texttt{missing})$. It selects:

$$a = \begin{cases} \text{Clarify}, & \text{if } \exists f \in \texttt{missing} \text{ essential for achieving } G \\ \text{Respond}, & \text{if } \texttt{missing} = \emptyset \text{ or non-essential} \end{cases}$$

Then it generates:

$$r = \begin{cases} \text{a clarification question}, & \text{if } a = \text{Clarify} \\ \text{an expert response}, & \text{if } a = \text{Respond} \end{cases}$$

The model is evaluated on decision quality and generation utility via LLM-as-a-Judge.

**Domain-General Task Modeling Framework:** While MIRAGE-MMMT is instantiated in the agricultural domain, the underlying task formulation, where an expert must decide between seeking clarification or issuing a response based on evolving multimodal dialogue context, generalizes to a broad class of consultative, decision-oriented interactions. This includes domains such as healthcare triage, customer support, legal advising, technical troubleshooting, and educational tutoring. Crucially, our framework is dataset-agnostic: given any multi-turn consultation corpus with follow-up user inputs (text and/or images), our pipeline can be applied to generate structured <Clarify> or <Respond> training data.

---

[4]https://ask.extension.org/

## 4.1 Dataset Curation

The goal of MIRAGE-MMMT is to model the decision-making process of an expert who must decide, based on the evolving multimodal context, whether to respond with actionable guidance or clarify by seeking more information. To simulate this, we extract segments of each dialogue such that the input context ends with a user utterance, and a later user message, typically written in response to an earlier expert query, is treated as a revealed fact. This allows us to reconstruct the decision point at which the expert, based on available context and images, would determine whether the current information is sufficient to respond or whether a clarification is necessary. We prompt a powerful LLM (e.g., GPT-4o) to generate a structured task instance consisting of dialogue context, referenced images, and metadata such as geographic location and topic. To ensure data safety, we perform automated PII sanitization while preserving domain relevance. (see Appendix G.2 for more details)

# 5 Experiments

We evaluate a diverse set of models on both MIRAGE-MMST and MIRAGE-MMMT benchmarks to assess expert-level multimodal reasoning as well as their ability to make accurate clarify-or-respond decisions and generate goal-consistent, grounded responses.

## 5.1 Model Selection

The LVLMs we evaluated can be divided into two groups: 1.) *Proprietary Models*: this group comprises SOTA GPT models (4.1, 4.1-mini, 4o, 4o-mini) [7], Claude-3 (3.5v2 Sonnet, 3.7 Sonnet, 3 Haiku) [11] available through API service. 2.) *Open-Source Models*: this group includes LLaMa-4 Scout-17B [45], LLaVa (7B mistral, 7B qwen2) [41], Qwen 2.5 VL models (3B, 7B, 32B, 72B) [5], Gemma 3 models (4b, 12, 27b) [57] and Intern VL3 (2B, 8B, 14B, 38B, 72B) [75]. 3.) *Finetuned Models* We perform multi-image multimodal fine-tuning of the Qwen 2.5 VL models on the MMST dataset. All experiments were conducted with NVIDIA A100-40GB/H200-141GB GPUs.

## 5.2 Evaluation Methodology

To facilitate consistent, interpretable, and reproducible evaluation of model outputs, we implement a structured LLM-as-a-Judge protocol. Rather than relying on single-model, single-pass assessments, we leverage an ensemble of reasoning-capable language models: *DeepSeek-R1-Distilled* [34], *Qwen3-32B* [5], and *Phi-4-Reasoning* [6], as judges. Each candidate's response is evaluated across three independent generations, with every generation scored by the full ensemble, resulting in nine total evaluations per sample. This design captures both linguistic variability across generations and scoring consistency across evaluators. To validate the fidelity of this framework, we quantify inter-rater agreement using Fleiss' $\kappa$ for categorical scoring and Kendall's W for ordinal coherence, ensuring both statistical reliability and scoring consistency across the ensemble. Fleiss' $\kappa$ scores for the ID task (binary classification) generally fall within the 0.75–0.88 range, indicating "good" to "excellent" agreement by standard interpretation guidelines. Full details, scoring rubrics, and agreement statistics are provided in Appendix D.1.1. The prompt for evaluation is showed in Figure 14.

Figure 30 illustrates our evaluation framework in action. The judge model is given the expert-authored gold response and systematically evaluates both the correctness of the model's prediction and the quality of its explanation. By reasoning step-by-step, the judge produces an interpretable "judgment trace" that makes the evaluation process transparent and highlights where the model's output aligns with, or diverges from, the expert's reasoning.

## 5.3 MIRAGE-MMST Results

We evaluate model performance on the **MMST-ID** task using two complementary metrics: 1.) *Identification Accuracy* which is a binary metric that measures whether the entity identified by the model matches the expert's answer. A response is scored as correct (1) if the predicted entity string exactly matches any of the reference fields: `entity name`, `scientific name`, or `common names`; otherwise, it receives a score of 0. *Reasoning Score*, evaluates the quality of the model's visual and linguistic justification for its prediction. It is graded on a 0–4 scale by the judges, based on the

presence of key visual clues, descriptive specificity, and causal coherence. Higher scores reflect more complete, interpretable, and expert-aligned justifications.

To evaluate model outputs on the **MMST-MG** task, we adopt a multidimensional scoring framework that assesses both factual correctness and communication quality. Each model response is rated on a 0–4 scale across the following four dimensions the judges: 1.) *Accuracy*: which measures the factual alignment with the expert's response. A perfect score (4) is awarded when the facts, terminologies, and recommendations fully align with the expert response. *Relevance* measures the ability of the model to stay on-topic with the user's original query and avoid tangential information. *Completeness* assesses whether the model covers all key information provided by the expert, and *Parsimony*, inspired by Occam's Razor; this score captures the conciseness, clarity, and actionability of the model response. We aggregate these scores in a 2:1:1:1 ratio. See Appendix D for more details.

## 5.4 Performance Comparison

**MMST-ID**: Table 2 presents the results of zero-shot prompting across 22 LVLMs on the MMST-ID task. All models are evaluated without task-specific fine-tuning, allowing for a fair comparison of out-of-the-box reasoning and grounding capabilities. We observe a consistent gap between closed-source and open-source models. GPT-4.1 achieves the highest Average Identification Accuracy (43.9%) and the top Reasoning Score (3.0), with other proprietary models like Claude 3 and GPT-4o performing competitively. In contrast, the best open-source model, Qwen2.5-VL-72B, reaches only 29.83% accuracy and 2.47 reasoning score.

**MMST-MG**: For cross-model comparisons, we compute a weighted aggregate score over the four evaluation metrics. Table 2 and Figure 3 indicate GPT-4.1 consistently outperforms all models across judges and metrics, achieving the highest scores in Accuracy (3.24) and Parsimony (3.01). Among open-source models, Qwen2.5-VL-32B and Gemma-3-27B perform best.

Performance improves steadily with model scale within families, especially for open models as seen in Figure 9. However, scaling gains are not uniform across families and show early saturation in some cases. Qwen2.5-VL series show the strongest scaling behavior, Gemma and InternVL3 also improve with size, but lag behind Qwen, particularly in identification accuracy, suggesting weaker visual grounding. None of the open models approach GPT-4.1 performance, which retains a sizable lead in both accuracy and reasoning. Notably, reasoning improvements appear to saturate earlier (32B), indicating that scale alone may not bridge the gap without better visual-linguistic alignment or task supervision. Open-source models still lag by 0.1–0.15 in weighted score compared to GPT-4.1, especially due to verbosity (low parsimony) and subtle factual gaps (accuracy).

**Model Finetuning:** We utilize the MMST training dataset to fine-tune Qwen 2.5 VL models (3B, 7B, 32B) and evaluate performance on both MMST-ID & MMST-MG tasks. Figure 2 illustrates the effects of LoRA-based fine-tuning on Qwen2.5-VL-3B model across four checkpoints vs the base model (base-Instruct). Our results indicate consistent improvements in seen entity accuracy on MMST-ID after fine-tuning, rising from 22.3% (base-Instruct) to a peak of 28.4% (epoch 6). Across model sizes, as seen in Figure 10, we see a consistent convergence behavior across epochs, with 32B achieving the strongest peak performance (37.6% at epoch 6) and reasoning score (3.04). However, models struggle to generalize, as accuracy on unseen entities remains stagnant. On the MMST-MG task, from the base-Instruct model to LoRA-ep6, we observe steady improvement across all four metrics as well. *Relevance* is the highest-scoring metric throughout, indicating the model's ability to remain on-topic is strong even in zero-shot, and improves further with tuning. *Parsimony* also improves consistently, indicating models can learn concise delivery of recommendations.

**Performance on Contextual Subset**: The contextual subset of MIRAGE-MMST reveals that inferring latent context remains a major bottleneck for current LVLMs. Even top-tier models like GPT-4.1 perform well but not well, and the gap to open-source models is significant. Compared to the standard subset, scores are consistently lower across the board, indicating that reasoning under partial observability and implicit context reconstruction are significant failure points for current models, as seen in Table 8. We also observe that current LVLMs make limited use of metadata, such as location and time, despite their importance in real-world agricultural reasoning see Table 10 for comparison with and without the meta-data. Across both proprietary and open-source models, the inclusion of metadata yields minimal improvements in identification accuracy (e.g., +1.6% for GPT-4.1, +0.6%

Table 2: Performance of large vision–language models on the MMST *Standard* benchmark, averaged over three open-source reasoning judges (DeepSeek-R1-Distill-Llama-70B, Qwen3-32B, Phi-4-reasoning). **ID task:** Acc (Identification Accuracy, 0–1) and Reason (Reasoning Score, 0–4). **MG task:** Acc (Answer Accuracy), Rel (Relevance), Comp (Completeness), Pars (Parsimony), all 0–4. Composite score W-Sum is computed as $W\text{-}Sum = (2*Acc + Rel + Comp + Pars)/20$ and ranges 0–1.

| Model | MMST-ID Task | | MMST-MG Task | | | | |
|---|---|---|---|---|---|---|---|
| | Acc (%) | Reason | Acc | Rel | Comp | Pars | W-Sum |
| gpt-4.1 | **43.9** | **3.01** | **3.24** | **3.60** | **3.22** | **3.01** | **0.82** |
| gpt-4.1-mini | 34.6 | 2.75 | 2.94 | 3.37 | 2.83 | 2.98 | 0.75 |
| gpt-4o | 39.3 | 2.49 | 2.77 | 3.16 | 2.43 | 3.00 | 0.71 |
| gpt-4o-mini | 22.4 | 2.18 | 2.65 | 3.02 | 2.28 | 2.78 | 0.67 |
| claude-3-7-sonnet | 33.9 | 2.64 | 2.82 | 3.23 | 2.69 | 2.88 | 0.72 |
| claude-3-5-sonnet | 32.0 | 2.51 | 2.75 | 3.17 | 2.57 | 2.92 | 0.71 |
| claude-3-haiku | 17.6 | 1.79 | 2.40 | 2.89 | 2.05 | 2.84 | 0.63 |
| Llama-4-Scout-17B-16E-Instruct | 20.1 | 2.11 | 2.51 | 2.93 | 2.27 | 2.61 | 0.64 |
| llava-v1.6-mistral-7b-hf | 7.1 | 1.34 | 2.20 | 2.50 | 1.86 | 2.20 | 0.55 |
| llava-onevision-qwen2-7b-ov-hf | 9.4 | 1.59 | 2.23 | 2.57 | 1.94 | 2.23 | 0.56 |
| Qwen2.5-VL-3B-Instruct | 17.2 | 1.48 | 2.09 | 2.38 | 1.78 | 2.08 | 0.52 |
| Qwen2.5-VL-7B-Instruct | 22.1 | 1.85 | 2.38 | 2.72 | 2.14 | 2.31 | 0.60 |
| Qwen2.5-VL-32B-Instruct | 25.1 | 2.43 | 2.87 | 3.19 | 2.88 | 2.43 | 0.71 |
| Qwen2.5-VL-72B-Instruct | 29.8 | 2.47 | 2.72 | 3.09 | 2.56 | 2.61 | 0.69 |
| gemma-3-4b-it | 10.4 | 1.84 | 2.28 | 2.71 | 2.32 | 2.16 | 0.59 |
| gemma-3-12b-it | 15.9 | 1.98 | 2.63 | 3.00 | 2.74 | 2.30 | 0.67 |
| gemma-3-27b-it | 18.9 | 2.22 | 2.77 | 3.14 | 2.87 | 2.43 | 0.70 |
| InternVL3-2B | 9.0 | 1.65 | 2.09 | 2.41 | 1.80 | 2.15 | 0.53 |
| InternVL3-8B | 11.9 | 1.77 | 2.35 | 2.71 | 2.10 | 2.46 | 0.60 |
| InternVL3-14B | 14.2 | 1.91 | 2.49 | 2.85 | 2.21 | 2.66 | 0.64 |
| InternVL3-38B | 19.2 | 2.12 | 2.56 | 2.95 | 2.28 | 2.75 | 0.66 |
| InternVL3-78B | 22.4 | 2.24 | 2.60 | 2.98 | 2.31 | 2.87 | 0.67 |

*Scores are averaged over the three judge models. Closed-source rows are red/light-red, open-source rows blue/light-blue. Bold purple numbers mark the overall best for each metric.*

for Qwen2.5-VL-72B) and virtually no gains in reasoning scores. In some cases, performance even degrades slightly, suggesting that models may treat metadata as irrelevant or distracting.

## 5.5 MIRAGE-MMMT Results

Before presenting experimental results, we first outline the structure of the decision task and its dependence on input observability. Under full observability, where the model is given access to an oracle goal state that explicitly enumerates known and missing information, a rule-based oracle achieves 98.45% decision accuracy. This illustrates that the challenge lies in the model's ability to infer missing context from dialogue history. In real-world settings, such goal states are unavailable at inference time. MIRAGE is therefore designed to evaluate models under partial observability, where decisions must be made using only the dialogue history. To study the effect of structured supervision, we trained logistic regression classifiers with progressively enriched inputs (Table 26). Performance improves modestly with access to the user-stated goal (69.79% → 71.34%), and substantially with access to the oracle goal state (89.27%), confirming that inferring implicit context is the key difficulty in this task.

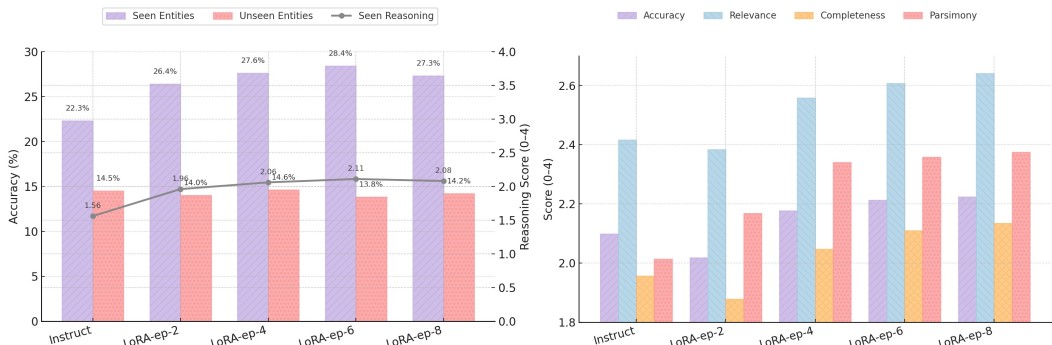

Figure 2: LoRA fine-tuning results on identification and management tasks. **Left:** ID Accuracy (%) on *seen entities* and *unseen entities* for Qwen2.5-VL-3B at epochs: Instruct (0), LoRA-ep-2, 4, 6, 8. The grey line marker ● traces the Reasoning Score (0–4 scale) on seen entities. **Right:** Finetuning performance on management task across four metrics.

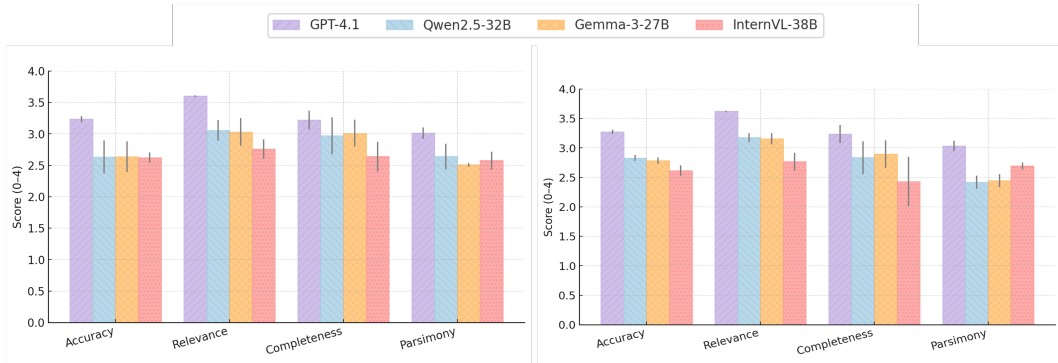

Figure 3: Mean performance of closed-source and leading open-source LVLMs on the Standard-MG (Left) and Contextual-MG (Right) subsets. Each bar is the average over three judges ($\text{Score}_{m,k} = \frac{1}{3}\sum_{r=1}^{3}\text{Judge}_r(m,k)$) for Accuracy (Acc), Relevance (Rel), Completeness (Com), and Parsimony (Par) on a 0–4 scale.

## 5.6 Performance Comparison

We evaluate the model's high-level decision and its alignment with user goals in three complementary ways. First, *Decision Accuracy* captures how often the model's chosen action, whether to ask a clarification question or to respond directly, matches the gold annotation; To ensure that when the model generates an utterance, it is doing so meaningfully, we measure *Clarify Relevance*: for every turn labeled <Clarify> or *Respond Relevance* for all turns labeled <Respond> using the LLM judges.

We conduct experiments under Zero-Shot and COT [61] prompt settings on MMMT shown in Table 3. In the zero-shot setting, GPT-4o achieves the highest Decision Accuracy (62.98%) and leads in both Clarify Goal Relevance (70.75%) and Respond Goal Relevance (77.07%), showing strong overall reasoning and generation alignment. Open models like LLaMA 4 Maverick and LLaMA 4 Scout perform competitively, reaching 53–49% decision accuracy. Introducing chain-of-thought (CoT) reasoning consistently improves performance across models. GPT-4o gains +2.5% in Decision Accuracy (to 65.5%) and improves goal relevance in both Clarify (+2%) and Respond (+1.5%) settings. Other models, especially Claude 3.7 and Qwen 72B, benefit more substantially (improvements in clarify goal relevance by +10.5%).

## 6 Conclusion

We introduce MIRAGE, a high-fidelity benchmark for evaluating vision-language models (VLMs) in expert-level agricultural consultations. MIRAGE addresses key limitations of existing benchmarks

Table 3: Comparison of Zero-Shot and CoT-Reasoning Performance on MIRAGE-MMMT. The table shows Decision Accuracy (Acc%), Clarify Goal Relevance, and Respond Goal Relevance. Arrows (↑/↓) indicate change from Zero-Shot prompting. Absolute Δ values are reported on the right.

| Model | Zero-Shot | | | CoT | | | Δ (CoT–ZS) | | |
|---|---|---|---|---|---|---|---|---|---|
| | Acc% | Clarify | Respond | Acc% | Clarify | Respond | Acc | Clarify | Respond |
| GPT-4o | 62.98 | 70.75 | 77.07 | **65.50**↑ | **72.80**↑ | **78.50**↑ | 2.52 | 2.05 | 1.43 |
| Claude 3.7 Sonnet | 57.80 | 24.39 | 23.45 | 62.40↑ | 34.90↑ | 28.70↑ | 4.60 | 10.51 | 5.25 |
| LLaMA 4 Maverick 17B | 53.75 | 65.08 | 70.65 | 59.80↑ | 69.10↑ | 74.20↑ | 6.05 | 4.02 | 3.55 |
| LLaMA 4 Scout 17B | 49.81 | 61.79 | 68.67 | 56.00↑ | 66.40↑ | 71.30↑ | 6.19 | 4.61 | 2.63 |
| LLaMA 3.2 90B | 45.09 | 52.43 | 62.27 | 50.40↑ | 56.20↑ | 65.10↑ | 5.31 | 3.77 | 2.83 |
| Qwen 72B | 31.33 | 58.76 | 75.00 | 37.40↑ | 63.90↑ | 76.50↑ | 6.07 | 5.14 | 1.50 |

by incorporating real-world, multimodal, and context-rich interactions grounded in domain-specific decision-making. Despite these strengths, MIRAGE has a few limitations. It does not yet simulate interactive dialogue with real users or user simulators, limiting evaluation of adaptation and dialogue flow over time. Visual follow-ups are also not modeled; user turns beyond the initial user turn are assumed to be text-only. Looking ahead, MIRAGE opens several promising directions for advancing multimodal reasoning and conversational capabilities of models in knowledge-intensive domains. Improving performance on unseen entities will require better open-world generalization strategies, such as retrieval-augmented generation, compositional reasoning, or domain-adaptive pretraining. Limited gains from metadata highlight the need for explicit context modeling, including structured encodings of time, location, and integration with external knowledge bases capturing seasonal and biological priors. Persistent challenges in clarify-or-respond decision-making point to a need for models that can infer latent user goals and reason about missing information. We hope this benchmark enables the development of more context-sensitive, knowledge-intensive VLMs. MIRAGE is publicly released to support the development of vision-language systems that go beyond narrow question answering, toward models capable of engaging in natural interactions that involve ambiguity, a need for clarification, visual understanding, and decision-making when critical context is implicit or missing, addressing common challenges in real-world expert consultations.

## 7   Limitations & Future Work

While MIRAGE provides a comprehensive evaluation framework for multimodal expert-level reasoning within agricultural consultations, several opportunities for future improvement remain. Currently, the benchmark primarily reflects smallholder and backyard gardening scenarios, largely derived from the AskExtension database. In future work, we aim to expand its scope to better represent large-scale industrial agricultural practices and the broader logistical, economic, and environmental challenges characteristic of commercial farming operations. Moreover, agriculture is inherently multidisciplinary, encompassing soil science, agricultural engineering, economics, environmental sustainability, and policy considerations. Future iterations of MIRAGE will extend coverage across these dimensions to better capture the full breadth of agricultural reasoning. In the multi-turn scenario (MMMT), our present design assesses models' ability to reason over dialogue context, but it does not yet fully simulate dynamic, interactive exchanges between models and users. To address this, future versions will incorporate agentic capabilities that enable models to leverage time- and location-based context, as well as respond adaptively to evolving user input. Such extensions will allow MIRAGE to better evaluate interactive conversational reasoning, including conversational repairs, user feedback integration, and contextually grounded decision-making.

## 8    Impact Statement

MIRAGE introduces a high-fidelity benchmark that addresses a critical gap in evaluating vision-language models (VLMs) for real-world, expert-level decision-making in knowledge-intensive domains. By grounding tasks in over 35,000 authentic agricultural consultations involving multimodal, context-rich queries, MIRAGE pushes the boundaries of current LVLM capabilities beyond perception and language generation to include causal reasoning, clarification strategies, and open-world generalization. It provides both a rigorous testbed and a development suite for building safer, more reliable AI assistants in domains where decision quality has tangible real-world consequences. However, caution is advised in fully relying on MIRAGE results: the benchmark, while comprehensive, does not yet simulate dynamic user interactions or evolving contexts, and agriculture itself encompasses a vast range of specialized knowledge not fully captured in this dataset. Thus, models performing well here may still face significant challenges in broader, real-world agricultural deployments, particularly in novel or unforeseen scenarios.

## 9    Acknowledgments

This work was supported by the Amazon-Illinois Center on AI for Interactive Conversational Experiences Award, the AIFARMS National AI Institute, and the Center for Digital Agriculture at the University of Illinois. We are grateful to the AskExtension team for providing the dataset and to Dr. John F. Reid, Dennis Bowman, and Talon Becker for contributing their domain expertise and invaluable feedback. The contributions of Shubham Garg and Hooshang Nayyeri to this work are independent of their roles at Amazon. The views expressed and conclusions drawn herein are solely those of the authors and do not necessarily reflect the official policies or positions, expressed or implied, of any sponsoring organizations. This work used Delta advanced computing and data resource at University of Illinois Urbana-Champaign and its National Center for Supercomputing Applications through allocation CIS250434 from the Advanced Cyberinfrastructure Coordination Ecosystem: Services & Support (ACCESS) program, which is supported by U.S. National Science Foundation grants #2138259, #2138286, #2138307, #2137603, and #2138296.

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

# MIRAGE: A Benchmark for Multimodal Information-Seeking and Reasoning in Agricultural Expert-Guided Conversations
## Supplementary Material

**Table of Contents in Appendix**

# A    Data Source

The MIRAGE benchmark is constructed from a large-scale archive of real-world agricultural consultations obtained from Ask Extension [25], a U.S. national digital platform maintained by the Extension Foundation. Ask Extension is part of the broader Cooperative Extension System, a federally supported network of land-grant universities that delivers science-based, community-oriented education and services across the United States. The platform connects members of the public, such as farmers, gardeners, or homeowners, with university-affiliated experts who provide timely, research-backed responses to their questions.

Inquiries submitted through the Ask Extension portal are answered by a diverse pool of domain specialists, including university faculty, Extension educators, and trained volunteers such as Master Gardeners. These responses reflect both academic rigor and region-specific expertise, leveraging a unique model of public scholarship that blends localized agricultural knowledge with the latest findings from land-grant institutions. This institutional provenance ensures that the answers used in our dataset are highly reliable, authored by qualified experts, and grounded in scientifically validated practices.

We collected approximately 285,393 interactions (218,431 for single-turn; 66,962 for multi-turn) from the Ask Extension platform, spanning from December 2012 to April 2025. Each entry captures a real question from a user, along with the corresponding expert response, and may include user-uploaded images, time of submission, and geographic metadata.

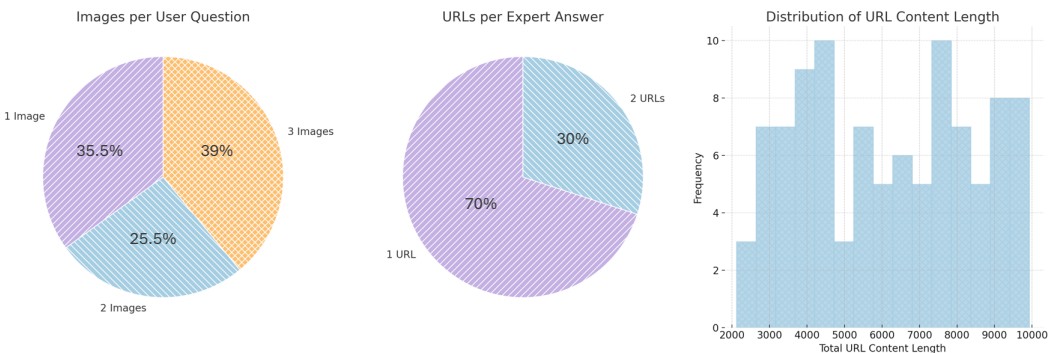

Figure 4: Filtered AskExtension data—(left) number of images per user question, (center) number of URLs per expert answer, and (right) distribution of total URL content length.

As seen in Figure 4, Our filtered AskExtension dialogues are strongly multimodal and reference-rich. Users typically include one to three images in their questions, about 35% of turns have a single image, 26% include two, and 39% include three. Experts in turn ground their advice in external sources: roughly 70% of answers cite a single URL, while the remaining 30% provide two. The total amount of content fetched from those links spans from about 2 000 up to 10 000 tokens per response, indicating that experts draw on substantial external context to support their guidance.

# B    Related Works

**Multimodal Large Language Models**: Recent advances in multimodal large language models (LLMs) have markedly expanded vision–language reasoning capabilities. Proprietary models such as *GPT-4* [8], *Claude 3 Sonnet* [10], and Google's *Gemini* [56] demonstrate strong capabilities in unifying visual and textual modalities, achieving notable success across diverse multimodal benchmarks. Concurrently, open-source models—including *Qwen-VL 2.5* [14], *Gemma 3* [57], and *InternVL-3* [75] have narrowed the performance gap while remaining publicly accessible. Although these models excel on general-domain benchmarks, they underperform in agriculture: they lack fine-grained visual expertise, agronomic terminology, and the ability to reason about rare biological entities and management practices. *MIRAGE* is designed to expose these weaknesses by providing domain-specific, multimodal tasks that require expert-level diagnosis and advice.

**Multimodal Agricultural Benchmarks**: Agricultural benchmarks [19, 26, 31] have progressed from controlled, image-only datasets to more realistic vision–language corpora. Early resources such as *PlantVillage* [36] and *PlantDoc* [53] emphasise leaf-level disease classification under laboratory and in-field conditions, respectively. The *AgMMU* [30] benchmark augments field images with farmer–expert dialogues, yet its evaluation is dominated by short, synthetic multiple-choice questions that limit open-ended reasoning. The recent *Crop Disease Domain Multimodal* (CDDM) [43] corpus contributes 137k crop–disease images paired with 1 M single-turn QA pairs, but it focuses narrowly on disease identification and management. In contrast, *MIRAGE* is derived from large-scale, real-world consultations covering more than 7000 biological entities; it offers both single-turn and multi-turn tasks, explicitly models clarify-or-respond decisions, and scores long-form answers along multiple dimensions of reasoning quality, thereby providing a far more ecologically valid and challenging test bed.

**Long-Form Question Answering Evaluation**: Lexical-overlap metrics such as ROUGE [39], BLEU [47], and BERTScore [74] are ill-suited to evaluating open-ended, knowledge-intensive answers, as they correlate poorly with human judgements. Recent work therefore adopts *LLM-as-Judge* paradigms, exemplified by *AlpacaEval* [22], *MT-Bench* [13], and *G-Eval* [44], in which powerful LLMs are prompted to assess responses along axes such as factuality, coherence, and completeness. While these methods better align with human preferences, single-model evaluations remain vulnerable to bias and opacity. *MIRAGE* employs an interpretable ensemble of reasoning-focused LLM judges *DeepSeek-R1-Distilled* [34], *Qwen 3-32B* [5], and *Phi-4-Reasoning* [6] and by conducting three independent generation–evaluation passes per sample. This multi-model, multi-run protocol enhances robustness, enables variance analysis, and provides publicly inspectable rationales, yielding transparent and reproducible long-form QA evaluation.

**State Tracking in Dialogue Systems**: State tracking has been a central component in task-oriented dialogue systems [16, 42], where it plays a critical role in maintaining a representation of user intent and progress toward the user's goal throughout the interaction. Dialogue state tracking (DST) typically involves predicting a belief state, a structured representation of slot-value pairs reflecting the current user intent at every turn. The widespread adoption of state tracking in dialogue systems has enabled robust multi-turn interactions, improved task success rates, and enhanced user satisfaction. Benchmarks like MultiWOZ [15, 24, 67] and Schema-Guided Dialogue (SGD) [48] have been instrumental in shaping the field by introducing multi-domain, open-schema challenges that pushed models to become more generalizable and scalable. These successes in dialogue suggest that structured, continuously updated representations of task progress can be highly beneficial in other interactive or sequential decision-making settings. Building on this insight, we introduce Goal State Tracking (GST), a mechanism that explicitly monitors whether all user-provided information is sufficient to generate a sound consultative recommendation; when gaps remain, GST triggers the agent to pose targeted clarifying questions, ensuring that advice is delivered only once the requisite context is complete.

# C MIRAGE-MMST

## C.1 Benchmark Details

We summarize the key statistics of MIRAGE-MMST dataset in Table 4, which forms the single-turn evaluation component of the MIRAGE benchmark. The dataset is partitioned into three subsets: Standard, Contextual, and Standard Training Data, each designed to support different evaluation and training objectives.

Table 4: Statistics for MIRAGE-MMST

| Overall Statistics | Standard | Contextual | Standard Training Data |
|---|---|---|---|
| Total Samples | 8 184 | 3 934 | 17 532 |
| Total Images | 15 069 | 8 069 | 33 120 |
| **Per-Sample Statistics** | | | |
| Avg. Question Words | 69.57 | 80.94 | 67.53 |
| Avg. Answer Words | 163.13 | 222.97 | 171.18 |
| Avg. Number of Images | 1.84 | 2.05 | 1.89 |
| **Category Statistics** | | | |
| Total IDENTIFICATION TASKS | 4 324 | - | 7 398 |
|    Plant Identification | 2 600 | - | 3 919 |
|    Insect and Pest Identification | 1 146 | - | 2 131 |
|    Plant Disease Identification | 578 | - | 1 348 |
| Total MANAGEMENT TASKS | 3 860 | 3 934 | 8 957 |
|    Plant Care and Gardening Guidance | 1 609 | 1 797 | 3 707 |
|    Insect and Pest Management | 725 | 641 | 1 689 |
|    Plant Disease Management | 1 047 | 1 184 | 2 445 |
|    Weeds / Invasive Plants Management | 479 | 312 | 1 116 |
| Others | - | - | 1 177 |
| **Entity Statistics** | | | |
| Plant Entities | 4 485 | 999 | 1 725 |
| Insect / Pest Entities | 1 732 | 306 | 840 |
| Plant Disease Entities | 639 | 200 | 320 |

**Overall Composition**: The dataset comprises a total of 29,650 high-quality user-expert interactions and over 56,000 user-submitted images. The Standard Benchmark subset contains 8,184 samples, while the Contextual Benchmark includes 3,934 samples that explicitly rely on implicit context (e.g., location-related, timing-related) not explicitly derivable from image or user query and metadata. An additional 17,532 samples are used for training and pre-tuning models.

**Per-Sample Characteristics**: Each sample includes both user and expert turns along with image inputs. On average, standard samples contain 69.6 words in user questions, 163.1 words in expert answers, and 1.84 images per sample. Contextual samples tend to be longer and more detailed, with 80.9 words per question, 222.9 words per answer, and a slightly higher image count (2.05 images per sample), reflecting the increased reasoning burden in these settings.

**Task Category Distribution**: MIRAGE-MMST covers a broad spectrum of expert tasks, which are grouped into two high-level categories: A.) Identification Tasks (7398) including, Plant Identification (3,919), Insect and Pest Identification (2,131), Plant Disease Identification (1,348) & B.) Management Tasks (8957) spanning, Plant Care and Gardening Guidance (3,707), Insect and Pest Management (1,689), Plant Disease Management (2,445), Weed and Invasive Plant Management (1,116)

**Entity Coverage**: The dataset includes over 7,000 unique biological entities, with fine-grained coverage across: 4,485 plant species, 1,732 insect/pest categories and 639 plant disease types.

## C.2 Data Curation Details

Our benchmark utilizes real-world dialogue data sourced from online platforms, necessitating a rigorous multi-step curation process to ensure the dataset's high quality and relevance. The curation process comprises four main steps (See Figure 5):

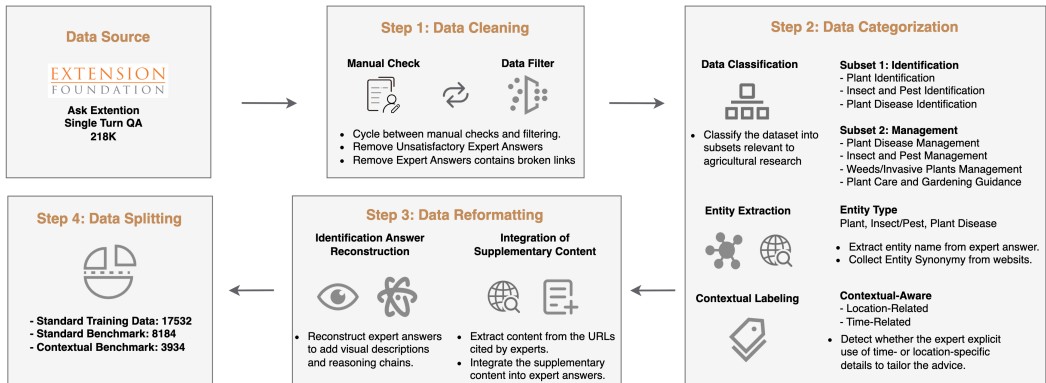

Figure 5: An Illustration of Data Curation Process for MIRAGE-MMST

## Step 1: Data Cleaning

The initial step involved removing unsatisfactory or incomplete data points. We first sampled the data and conducted manual checking, identifying four primary issues. Specifically, we excluded dialogues where the expert: (1) recommended contacting another individual or organization; (2) requested additional information from the user without providing a complete standalone response, as the benchmark focuses exclusively on single-turn dialogues; (3) expressed uncertainty regarding their response; or (4) explicitly indicated an inability to assist. Subsequently, we employed GPT-4o-mini to automate the filtering of these identified issues. Manual verification and automated filtering were iteratively performed to refine and optimize the filtering prompts. Final results of this process are described in Section C.3. Additionally, expert responses containing inaccessible or broken URLs were removed to ensure the dataset's integrity and usability.

Let the initial dataset of raw expert-user dialogues be denoted as:

$$\mathcal{D}_0 = \{(q_i, a_i, m_i)\}_{i=1}^N$$

where $q_i$ is the user query, $a_i$ is the expert response, $m_i$ is associated metadata (e.g., images, timestamp, location), and $N$ is the number of raw entries.

**Filtering Unvalid Samples**    We define a filtering function $f_{\text{valid}} : \mathcal{D}_0 \to \{0, 1\}$, which retains a dialogue only if the response:

- is complete (not asking for follow-up or deferring);
- does not express uncertainty or redirect the user;
- is not a broken or inaccessible URL;

The cleaned dataset is:

$$\mathcal{D}_1 = \{(q_i, a_i, m_i) \in \mathcal{D}_0 \mid f_{\text{valid}}(q_i, a_i, m_i) = 1\}$$

## Step 2: Data Categorization

**Data Classification**: We categorized the dataset into subsets relevant to agricultural language model research, resulting in seven primary categories grouped into two subsets: **Subset 1: Identification** (Plant Identification, Insect and Pest Identification, Plant Disease Identification) and **Subset 2: Management** (Plant Disease Management, Insect and Pest Management, Plant Care and Gardening Guidance, Weeds/Invasive Plants Management). Dialogues not fitting these categories were labeled as "Others." We used GPT-4.1 to classify the dataset.

Let $C : \mathcal{D}_1 \to \mathcal{C}$ be a classifier mapping each sample to one of $K = 7$ predefined agronomic categories. Define:

$$\text{cat}_i = C(q_i, a_i, m_i)$$

**Entity Extraction**: We extracted relevant entities based on the assigned category: plant names for Plant Identification, Plant Care and Gardening Guidance, and Weeds/Invasive Plants Management; insect or pest names for Insect and Pest Identification and Management; and disease names for Plant Disease Identification and Management. We used GPT-4.1-mini to extract the entities. These entities are then enriched with their synonyms (See Section C.4).

Let $E_{\text{plant}}, E_{\text{pest}}, E_{\text{disease}}$ be entity extractors for respective domains. Then:

$$e_i = \begin{cases} E_{\text{plant}}(q_i, a_i), & \text{if } \text{cat}_i \in \{\text{Plant ID, Plant MG, Care/Weeds MG}\} \\ E_{\text{pest}}(q_i, a_i), & \text{if } \text{cat}_i \in \{\text{Pest ID, Pest MG}\} \\ E_{\text{disease}}(q_i, a_i), & \text{if } \text{cat}_i \in \{\text{Disease ID, Disease MG}\} \end{cases}$$

**Contextual-Aware Labeling**: Contextual-aware labeling involved analyzing expert answers that leveraged implicit context such as user location and timing. These instances included cases where experts cited location-specific regulations or practices, provided location-dependent advice, referenced current weather conditions specific to the user's location and timing (e.g., recent drought, frost conditions), or offered time-specific recommendations. These were labeled as "location_related" or "time_related" data. We used GPT-4o and GPT-4o-mini to label the data. This step also involved manual checking. Final results of this process are described in Section C.3.

Let $\phi_{\text{ctx}} : \mathcal{D}_1 \to \{0, 1\}$ indicate whether a sample includes contextual elements (e.g., location-specific recommendations):

$$\phi_{\text{ctx}}(q_i, a_i, m_i) = \begin{cases} 1 & \text{if contextual reasoning is present} \\ 0 & \text{otherwise} \end{cases}$$

Split the dataset as:

$$\mathcal{D}_{\text{standard}} = \{(q_i, a_i, m_i, \text{cat}_i, e_i) \mid \phi_{\text{ctx}} = 0\}, \quad \mathcal{D}_{\text{contextual}} = \{(q_i, a_i, m_i, \text{cat}_i, e_i) \mid \phi_{\text{ctx}} = 1\}$$

**Step 3: Data Reformatting**

**Content Removal and Question Enhancement**: We reformatted the remaining data to enhance clarity and appropriateness for language model benchmarks. Specifically, we removed personal identification information, references specific to the "Ask Extension" service, and content unsuitable for interactions with language models (e.g., mentions of voicemails). Additionally, relevant details from dialogue titles were merged into questions when they provided additional context. We used GPT-4.1-mini model to reformat the data.

**Integration of Supplementary Content**: Approximately half of the expert responses contained URLs referencing supplementary information. To enrich these responses, we crawled and extracted the content from these URLs. Subsequently, we used the GPT-4.1 model to integrate this supplementary content with the original expert answers, producing more detailed and comprehensive responses.

**Identification Answer Reconstruction with Visual Enhancement**: Initial identification (ID) dataset expert responses included reasoning processes and conclusions but lacked consistent formatting, comprehensive visual descriptions, and complete reasoning chains. We utilized the GPT-4.1-mini model, capitalizing on its multimodal and information integration capabilities, to reconstruct and enhance these responses. This involved adding detailed descriptions of key visual characteristics (such as distinguishing features of plants and insects, or observable symptoms of plant diseases) and constructing clear, coherent reasoning chains. The standardized answers were formatted into a concise single-paragraph structure, clearly presenting both the reasoning process and the final result, thereby facilitating efficient benchmark evaluation.

Let $\mathcal{D}_{\text{ID}} \subseteq \mathcal{D}_{\text{standard}} \cup \mathcal{D}_{\text{contextual}}$ with $\text{cat}_i \in \{\text{Plant ID, Pest ID, Disease ID}\}$. We define an enhanced answer:

$$a_i' = \text{llm}(a_i, m_i)$$

The reconstructed dataset is:

$$\mathcal{D}_{\text{ID-enhanced}} = \{(q_i, a_i', m_i, \text{cat}_i, e_i) \in \mathcal{D}_{\text{ID}}\}$$

**Step 4: Data Splitting**

We first divided the dataset into standard and contextual subsets based on contextual-aware labeling.

**Standard Data**: The standard data subset was partitioned into benchmark (30%) and training (70%) datasets. The splitting aimed to maximize diversity by ensuring at least one sample per entity in the benchmark, which led to some entities appearing exclusively in the benchmark due to limited samples. Additionally, we preserved the original distribution across all seven categories and maintained the proportion of URL-based responses within each category. Ultimately, we obtained a **Standard Benchmark** dataset of 8,184 dialogues and a training dataset of 17,532 dialogues.

**Contextual Data**: Context-sensitive questions were augmented with explicit user location and timing details. Due to their complexity, these were entirely allocated to the **Contextual Benchmark**, totaling 3,934 dialogues.

We define a stratified sampling function $S : \mathcal{D}_{\text{standard}} \rightarrow \{0, 1\}$ for selecting 30% of data for benchmarking:

$$\mathcal{D}_{\text{benchmark}}^{\text{standard}} = \{x \in \mathcal{D}_{\text{standard}} \mid S(x) = 1\}, \quad \mathcal{D}_{\text{train}}^{\text{standard}} = \mathcal{D}_{\text{standard}} \setminus \mathcal{D}_{\text{benchmark}}^{\text{standard}}$$

### C.3 Manual Check and Model Selection for Data Sanitation

As part of our data curation pipeline, we conducted a targeted analysis of user-expert conversations to identify specific characteristics that affect data quality. This step was essential for filtering low-value interactions and retaining samples that reflect sensitivity to latent contextual reasoning typical in real-world agricultural consultations.

We focused on three key characteristics:

- **Unsatisfactory**: These include expert replies that fail to provide meaningful or actionable guidance. Common patterns include vague disclaimers such as "I'm not sure how to help you with this" or deferrals to third-party support ("You may want to contact your local extension office"). These represent non-informative speech acts and were marked for exclusion from the curated dataset to maintain high informational integrity.

- **Location-related:** Many expert responses assumed the user's geographic context, such as referencing local regulations, climate patterns, or soil characteristics, without this context being explicitly provided by the user. While this introduces contextual elision, these responses are not deficiencies; they reflect realistic, situated expertise. We retained these samples, recognizing their value in evaluating models' ability to interpret or recover latent geographic/location-related context.

- **Time-related**: Similarly, several expert responses implicitly relied on temporal context, such as seasonal crop cycles or pest development stages. These exhibited temporal under-specification, where the meaning of the advice depends on when the consultation occurred (e.g., "The pest is likely in its larval stage right now" during a spring consultation). These interactions were preserved as they reflect authentic domain-specific reasoning under temporal constraints.

To identify these characteristics at scale, we first manually annotated a stratified sample of 111 conversations. Each was labeled with one or more of the above characteristics. We then used few-shot prompting to evaluate a set of large language models (LLMs) on their ability to classify these characteristics. For each model, we provided a set of illustrative examples demonstrating how each characteristic manifests in expert replies. We measured performance using standard classification metrics: accuracy, precision, recall, and F1 score, broken down by characteristic type. (See Table 5)

Based on this evaluation, we selected gpt-4o-min plus analysis for filtering unsatisfactory data and labeling location-related data; gpt-4o plus analysis for labeling time-related data in our data curation workflow.

Table 5: Performance of various LLMs on conversational characteristic classification. "Model + Analysis" denotes that the model first generates an analysis before classification; "Model" indicates direct classification without intermediate analysis.

| Characteristic | Count | Model | Accuracy | Precision | Recall | F1 Score |
|---|---|---|---|---|---|---|
| Unsatisfactory | 51 | gpt-4o-mini | 0.8559 | 0.8302 | 0.8627 | 0.8462 |
| | | gpt-4o | 0.8198 | 0.7925 | 0.8235 | 0.8171 |
| | | gpt-4o-mini + analysis | 0.8559 | 0.7778 | 0.9608 | **0.9176** |
| | | gpt-4o + analysis | 0.8288 | 0.7963 | 0.8431 | 0.8333 |
| | | gemini-2.0-flash | 0.8739 | 0.8364 | 0.9020 | 0.8880 |
| | | gemini-2.0-flash + analysis | 0.8829 | 0.8654 | 0.8824 | 0.8789 |
| Location-related | 25 | gpt-4o-mini | 0.8288 | 0.5714 | 0.9600 | 0.8451 |
| | | gpt-4o | 0.9189 | 0.8333 | 0.8000 | 0.8065 |
| | | gpt-4o-mini + analysis | 0.9279 | 0.8400 | 0.8400 | **0.8400** |
| | | gpt-4o + analysis | 0.9099 | 0.8571 | 0.7200 | 0.7438 |
| | | gemini-2.0-flash | 0.8378 | 0.5946 | 0.8800 | 0.8029 |
| | | gemini-2.0-flash + analysis | 0.8829 | 0.6875 | 0.8800 | 0.8333 |
| Time-sensitive | 16 | gpt-4o-mini | 0.8559 | 0.5000 | 1.0000 | 0.8333 |
| | | gpt-4o | 0.9189 | 0.6522 | 0.9375 | 0.8621 |
| | | gpt-4o-mini + analysis | 0.8559 | 0.5000 | 0.9375 | 0.7979 |
| | | gpt-4o + analysis | 0.9640 | 0.8750 | 0.8750 | **0.8750** |
| | | gemini-2.0-flash | 0.6757 | 0.3077 | 1.0000 | 0.6897 |
| | | gemini-2.0-flash + analysis | 0.8468 | 0.4848 | 1.0000 | 0.8247 |

## C.4  Biological Entity Synonymy Collection

To facilitate fair evaluation of biological entity identification in model outputs, we developed a comprehensive name collection pipeline that aggregates all valid references, both scientific and vernacular, for biological entities in our dataset. Rather than standardizing to a single canonical form, our objective was to compile an exhaustive synonymy for each entity, enabling robust matching regardless of the nomenclatural variant used. The system queries multiple authoritative taxonomic databases via their APIs in a hierarchical approach, beginning with the Global Biodiversity Information Facility (GBIF) [2] and extending to iNaturalist [59] Encyclopedia of Life (EOL) [1], Integrated Taxonomic Information System (ITIS) [3], Wikipedia [63], and NCBI Taxonomy [4]. Queries are enhanced with category-specific context (e.g., kingdom Plantae for botanical entries) to improve retrieval accuracy. For each entity, we preserve all retrieved scientific names (including accepted names, synonyms, and historical nomenclature) and vernacular names across languages, with metadata indicating the source authority. This comprehensive approach prevents unfair penalization during model evaluation when, for instance, a model correctly identifies an organism using its scientific name (Phytolacca americana) while the reference answer uses a common name ("pokeweed"), or vice versa. The resulting enriched dataset maintains the original hierarchical structure while appending all valid nomenclatural alternatives, thereby supporting more equitable assessment of biological entity recognition capabilities.

## D  MMST Evaluation Criteria

### D.1  Reasoning LLM-as-Judge

#### D.1.1  Traditional LLM-as-Judge

Traditional metrics for evaluating long-form question answering have significant limitations. Research by Xu et al. [65] reveals that established approaches like ROUGE and BERTScore frequently diverge from human quality assessments, creating a fundamental evaluation challenge. A breakthrough solution has emerged in the form of language model-based evaluation frameworks. Cortes et al. [21] found that carefully designed prompting strategies with advanced models like GPT-4 can

achieve remarkable alignment with human judgment, particularly when assessing the thoroughness of responses. This paradigm shift suggests that AI systems themselves may offer the most effective tools for evaluating complex natural language generation tasks.

LLMs have become widely adopted as evaluators in benchmarks like AlpacaEval [23], MT-Bench [13], and Chatbot Arena [18], where models such as GPT-4 conduct pairwise preference assessments based on helpfulness, factuality, and engagement. Frameworks like G-Eval [44] further enable fine-grained scoring by prompting models to assess specific dimensions using structured rubrics. These LLM-based evaluations have shown stronger alignment with human judgments than traditional metrics, especially on long-form and knowledge-intensive tasks.

However, relying on a single model introduces concerns around opacity, bias, and instability, including self-preference and sensitivity to output variance. To address this, we implement a multi-model, multi-run evaluation protocols that improve interpretability, reduces bias, and yields more robust and reproducible assessments.

### D.1.2  Leveraging an Interpretable Ensemble of Reasoning LLMs

To address the limitations of single-model, single-pass evaluation pipelines, we propose an interpretable and robust evaluation framework based on an ensemble of reasoning-capable LLMs: Deepseek-R1-Distilled [34], Qwen-3-32B [5], and Phi-4-Reasoning [6]. These models were selected not only for their demonstrated strength in multi-hop reasoning and long-form comprehension but also for their open accessibility, ensuring that our evaluation protocol is fully transparent and reproducible without dependence on proprietary APIs. Our evaluation protocol is also distinguished by its multi-run robustness. For each benchmark sample, we perform three independent inference runs of the candidate model to capture natural generation variability. Each of these outputs is then evaluated independently by the full ensemble of three judge models. This results in nine total evaluations per sample, allowing us to report aggregated metrics that reflect not only average performance but also stability and consistency across generations and evaluators. We further analyze cross-model agreement and judgment variance to ensure evaluation fidelity. This interpretable ensemble-based and multi-run evaluation represents a significant step forward in the use of LLMs as evaluators. It brings together the benefits of scale and automation while maintaining experimental rigor.

### D.2  Reliability and Robustness of Multi-Judge Evaluation

To ensure the statistical robustness and reproducibility of our evaluation framework, we go beyond model-averaged scores and perform formal inter- and intra-judge reliability assessments. These analyses validate both the consistency across judges (inter-rater agreement) and stability within individual judges across multiple runs (intra-rater reliability), providing a more rigorous characterization of evaluation quality.

**Inter-Judge Agreements**: We assess agreement between our ensemble of LLM judges: Deepseek-R1-Distilled, Qwen-3-32B, and Phi-4-Reasoning, using two complementary statistical measures:

- **Fleiss' Kappa**: To measure categorical agreement across multiple judges, we binarize evaluation outcomes (e.g., correct vs. incorrect) and compute *Fleiss' $\kappa$* [28], a widely used metric for evaluating agreement on nominal data among fixed raters. This quantifies how consistently the LLM judges classify outputs beyond what would be expected by chance.

- **Kendall's W (Coefficient of Concordance**: For tasks involving ordinal scoring or ranking of generations, we compute *Kendall's W* [27], a non-parametric measure of rank correlation. This accounts for judges using different scoring scales by focusing on relative orderings. To accommodate tied ranks, we use the corrected-for-ties version of *Kendall's W*. The coefficient ranges from 0 (no agreement) to 1 (perfect agreement), enabling us to quantify the degree of concordance in evaluative judgments.

The results of our inter-judge reliability analysis, shown in Figure 6, reveal consistently high agreement among the three LLM judges across a diverse set of evaluated models. *Fleiss' $\kappa$* scores for the ID task (binary classification) generally fall within the 0.75–0.88 range, indicating "good" to "excellent" agreement by standard interpretation guidelines The bottom plot presents inter-judge agreement measured by *Kendall's W* across four evaluation dimensions: accuracy, completeness, parsimony, and relevance, for 23 vision-language models. Overall, models exhibit moderate to strong agreement

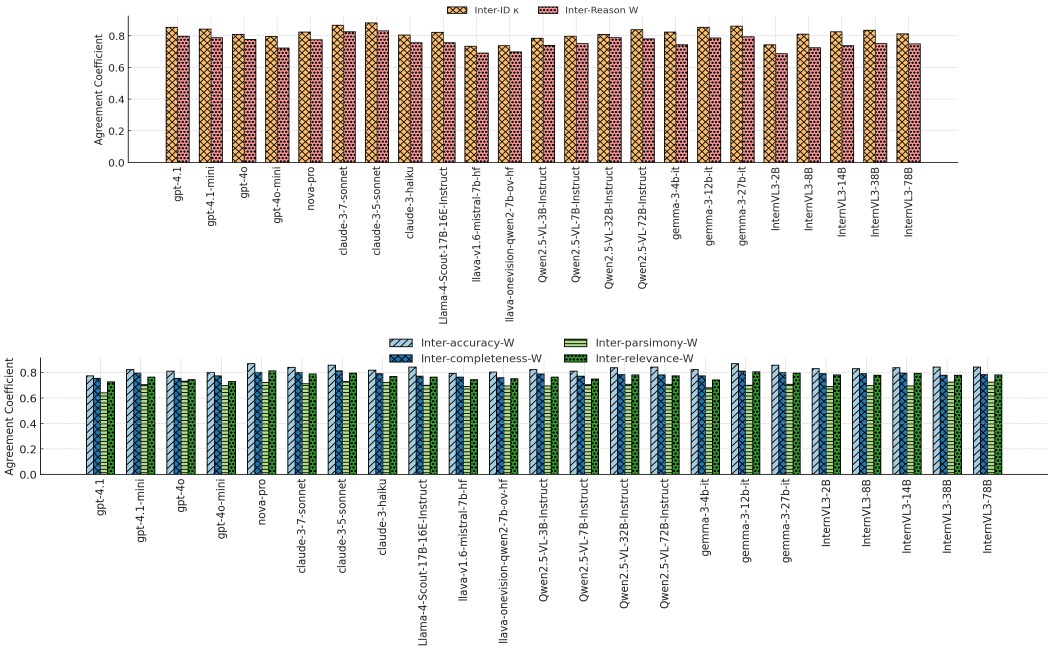

Figure 6: **(Top)**: Inter-judge reliability of LLM-based evaluation using *Fleiss' κ* (binary classification of ID correctness) across evaluated models on MMST-ID. Each bar represents the agreement among the three ensemble judges—Deepseek-R1-Distilled, Qwen 3 32B, and Phi-4-Reasoning. **(Bottom)**: *Kendall's W* across four evaluation dimensions in MMST-MD: *accuracy*, *completeness*, *parsimony*, and *relevance*—for 23 vision-language models. Higher values indicate stronger rank correlation among the three LLM judges.

(W = 0.69–0.87), indicating consistent ranking behavior among the three LLM judges. Agreement is highest for accuracy. In contrast, parsimony scores demonstrate lower concordance (W = 0.64–0.73), indicating greater variability in how judges interpret brevity or conciseness. Qwen, Claude, and Gemma families show relatively stable agreement across all four dimensions, highlighting their reliability as evaluated agents. The relatively lower agreement for some models, such as LLaVA-based variants and InternVL3-2B, particularly on reasoning metrics, highlights instances where judge interpretations diverged, possibly due to varied output styles or ambiguous task completions.

**Intra-Judge Reliability**: In addition to evaluating agreement across different models, we assess intra-judge reliability by running each judge model three independent times on the same set of samples. This allows us to measure the stability of each model's judgments under natural generation variability.

For this purpose, we compute the Intraclass Correlation Coefficient (ICC), which quantifies how consistently a single model scores the same item across multiple runs. ICC is particularly suited for this setting as it accounts for both within-subject and between-subject variability, providing a continuous-scale assessment of intra-rater consistency. Following established interpretative guidelines [20] we classify ICC values into bands (e.g., moderate, good, excellent) to report the strength of reliability for each judge.

To assess the consistency of our LLM-based judges, we conducted a three-run intra-rater reliability analysis using the Intraclass Correlation Coefficient (ICC2) shown in Figure 7, separately computed for binary ID judgments and ordinal reasoning assessments. The results indicate that both DeepSeek-R1 and Qwen3-32B exhibit strong intra-judge reliability. DeepSeek-R1 shows excellent agreement on ID assessments across almost all models, with ICC2 values typically in the 0.85–0.90 range. We also observe that LLaVA-based models tend to show more intra-model fluctuation, potentially due to less structured or variable outputs.

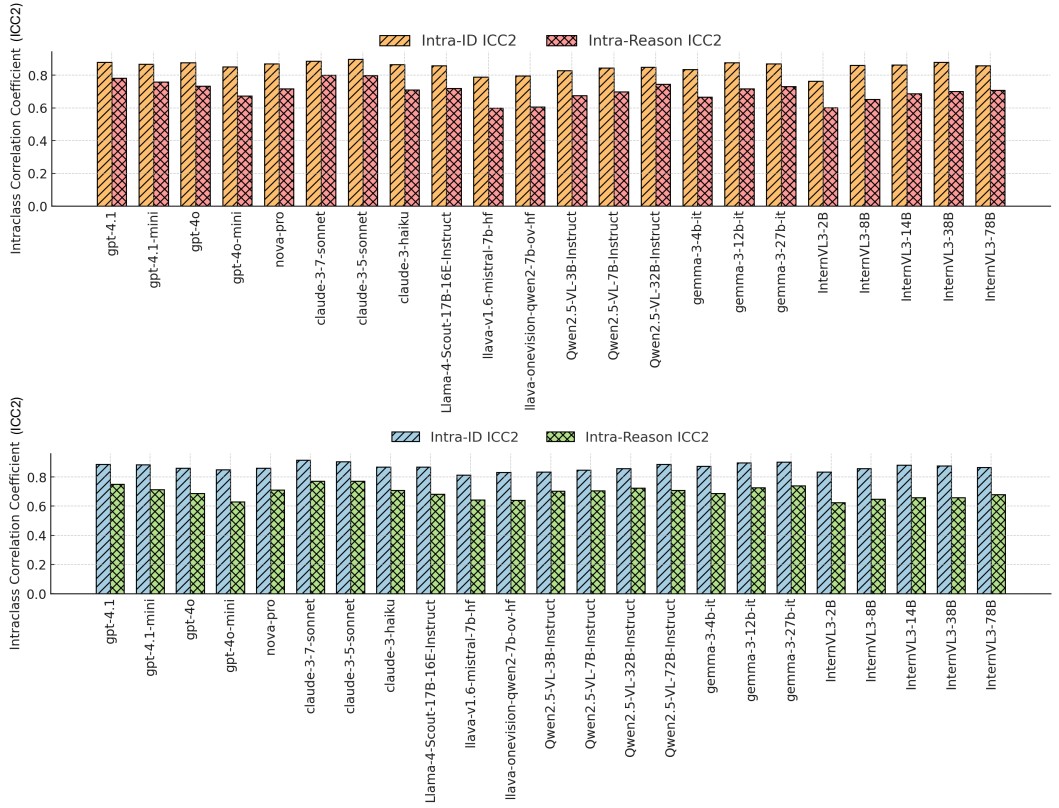

Figure 7: Intra-rater reliability (ICC2) of our interpretable ensemble judges over three independent runs, shown for two judge models. **(Top)**: DeepSeek-R1-Distill and **Bottom**: Qwen3-32B. Both models exhibit consistently high reliability of judgement on MMST-ID

## D.3 Management (MG) Evaluation Criteria and W-Sum Score

To evaluate model performance on the **Management (MG)** subset, we adopt a four-dimensional evaluation framework capturing complementary aspects of answer quality: **Accuracy**, **Relevance**, **Completeness**, and **Parsimony**. These dimensions were chosen based on findings from the long-form QA literature and insights from agricultural experts.

- **Accuracy** measures the factual correctness of the recommendation or diagnosis.
- **Relevance** assesses whether the response remains on-topic and directly addresses the user's query.
- **Completeness** evaluates whether essential details (e.g., causal factors, conditions, or next-step actions) are covered.
- **Parsimony** rewards concise, evidence-based explanations that avoid unnecessary speculation or verbosity.

The first three dimensions align with multi-axis evaluation principles in long-form QA studies [66, 21], which demonstrate that completeness and relevance correlate more strongly with human judgments than surface-level metrics such as ROUGE or BERTScore. The fourth dimension, **Parsimony**, is inspired by the cognitive and philosophical principle of simplicity (Occam's razor), emphasizing clear and minimal explanations in expert communication. We provide a detailed discussion and illustrative examples of diagnostic parsimony in Appendix D.4.

To enable model comparison with a single scalar, we compute a **Weighted-Sum (W-Sum)** score that aggregates these four dimensions. The adopted weight ratio of **2:1:1:1** reflects expert consensus that factual **Accuracy** should carry the greatest influence, since factual errors can lead to misleading or

harmful recommendations. While we also report per-dimension results for transparency, the W-Sum provides a concise, interpretable measure for aggregate ranking.

## D.4 Diagnostic Parsimony

Diagnostic parsimony refers to the principled restraint in offering explanations, favoring the simplest account consistent with observed evidence while avoiding unnecessary speculation. [46] shows that humans have a robust bias toward simpler explanations, not just for cognitive ease but because simplicity aids understanding, memory, and decision-making. In agriculture, this principle is especially critical: farmers and gardeners often seek immediate, actionable advice under time-sensitive or resource-constrained conditions. Overly elaborate responses can obscure key insights, introduce confusion, or even lead to misapplied interventions. As in medicine and law, where expert communication must balance completeness with clarity, parsimony is a cornerstone of effective agronomic consultation. LLM/LVLM answers in such knowledge-intensive domains must therefore strive to be concise yet comprehensive, ensuring the user gets just enough information to act confidently. Studies show that human evaluators, particularly non-experts, tend to prefer shorter, to-the-point responses, even when some details are omitted [66].

However, many instruction-following LLMs/LVLMs often default to verbose outputs, including tangential or overly cautious elaborations. As illustrated in Figure 8, when responding to a user's concern about bark defects on a tree, GPT-4.1 generates a detailed and structured diagnostic explanation covering wound physiology, tree tolerance, preventative care, and multi-step recommendations. Qwen 2 VL 72B, while more concise, still includes speculative possibilities (e.g., fungal infection, water stress) and a broad checklist of actions. In contrast, the human expert offers a brief but decisive interpretation, attributing the damage to non-serious activity and explicitly recommending no treatment. This behavior not only hinders user trust and satisfaction but also confounds traditional QA metrics like exact match (EM) and F1, which are poorly suited to judge informativeness under parsimony constraints. [9].

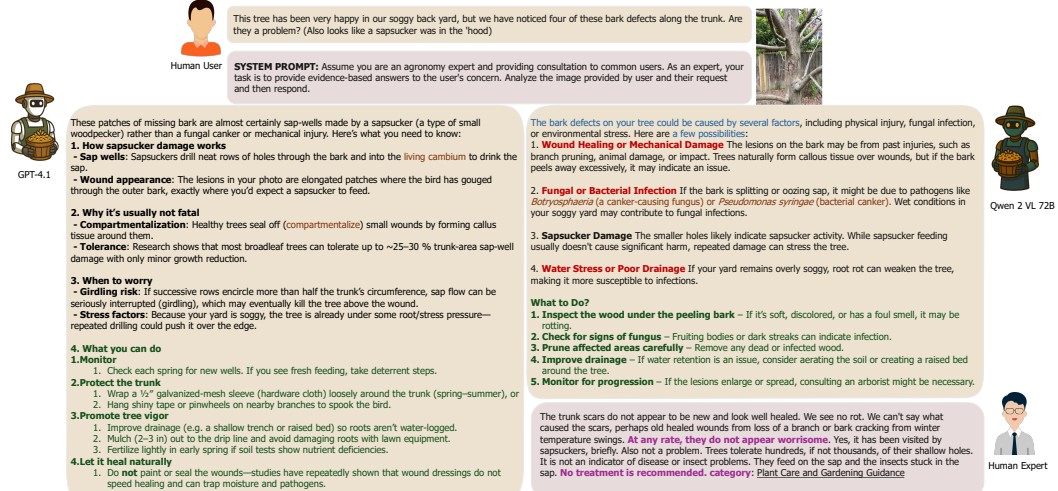

Figure 8: An illustration of three different diagnostic responses to the same user request from MMST-MG: Multiple hypotheses/conditions, speculative statements without direct evidence, technical terminology, actionable intervention recommendations, and definitive expert assessments are each highlighted in the figure to illustrate the taxonomy of statement types.

# E Additional MMST Benchmark Results

## E.1 MMST Benchmark Main Results

### E.1.1 Standard ID Benchmark Results

Table 6 summarises identification accuracy (**Acc**, %) and reasoning accuracy (0–4 scale) for a diverse LVLM on the MMST *Standard-ID* benchmark, reporting scores under three automated judges—DeepSeek-R1-Distill, Qwen3-32B, and Phi-4-reasoning. The table reveals a persistent proprietary lead: GPT-4.1 achieves the highest scores across all judges, retaining a margin of roughly 13 pp in identification accuracy and 0.5 points in reasoning over the strongest open-source model, Qwen-2.5-VL-72B-Instruct.

Table 6: Performance Comparison of Large Language Models on MMST (Standard-ID) Benchmark

| Model | DeepSeek-R1-Distill | | Qwen3-32B | | Phi-4-reasoning | |
|---|---|---|---|---|---|---|
| | Acc (%) | Reasoning | Acc (%) | Reasoning | Acc (%) | Reasoning |
| gpt-4.1 | **44.6** | **3.07** | **44.7** | **2.78** | **42.4** | **3.17** |
| gpt-4.1-mini | 36.3 | 2.79 | 35.0 | 2.51 | 32.6 | 2.95 |
| gpt-4o | 40.9 | 2.52 | 40.5 | 2.29 | 36.6 | 2.65 |
| gpt-4o-mini | 24.3 | 2.19 | 22.4 | 2.01 | 20.4 | 2.34 |
| claude-3-7-sonnet | 34.3 | 2.71 | 34.5 | 2.40 | 32.9 | 2.81 |
| claude-3-5-sonnet | 32.3 | 2.59 | 32.4 | 2.29 | 31.4 | 2.65 |
| claude-3-haiku | 18.7 | 1.83 | 18.4 | 1.63 | 15.8 | 1.92 |
| Llama-4-Scout-17B-16E-Instruct | 20.6 | 2.13 | 21.2 | 1.96 | 18.5 | 2.24 |
| llava-v1.6-mistral-7b-hf | 7.7 | 1.36 | 7.2 | 1.18 | 6.3 | 1.47 |
| llava-onevision-qwen2-7b-ov-hf | 9.9↑29 | 1.63↑20 | 9.4↑31 | 1.45↑23 | 9.0↑43 | 1.70↑16 |
| Qwen2.5-VL-3B-Instruct | 17.4 | 1.55 | 18.1 | 1.37 | 16.0 | 1.53 |
| Qwen2.5-VL-7B-Instruct | 22.5↑29 | 1.91↑23 | 23.3↑29 | 1.70↑24 | 20.5↑28 | 1.95↑28 |
| Qwen2.5-VL-32B-Instruct | 26.1↑16 | 2.54↑33 | 25.3↑9 | 2.18↑28 | 23.8↑16 | 2.57↑32 |
| Qwen2.5-VL-72B-Instruct | 30.8↑18 | 2.59↑2 | 30.3↑20 | 2.22↑2 | 28.4↑19 | 2.60↑1 |
| gemma-3-4b-it | 10.4 | 1.87 | 10.7 | 1.70 | 10.2 | 1.95 |
| gemma-3-12b-it | 16.1↑55 | 2.08↑11 | 15.9↑49 | 1.82↑7 | 15.7↑54 | 2.05↑5 |
| gemma-3-27b-it | 19.3↑20 | 2.28↑10 | 19.2↑21 | 2.03↑12 | 18.1↑15 | 2.35↑15 |
| InternVL3-2B | 10.0 | 1.64 | 8.9 | 1.53 | 8.2 | 1.77 |
| InternVL3-8B | 12.2↑22 | 1.81↑10 | 12.2↑37 | 1.64↑7 | 11.4↑39 | 1.86↑5 |
| InternVL3-14B | 14.7↑21 | 1.95↑8 | 14.2↑16 | 1.76↑7 | 13.6↑19 | 2.02↑9 |
| InternVL3-38B | 20.0↑36 | 2.13↑9 | 19.7↑39 | 1.96↑11 | 17.8↑31 | 2.26↑12 |
| InternVL3-78B | 23.9↑20 | 2.28↑7 | 22.6↑15 | 2.07↑6 | 20.8↑17 | 2.37↑5 |

*Models are color-coded by type: closed-source models in red/light red, open-source models in blue/light blue.*
*↑ values indicate percentage improvements over the previous model size in the same family. Bold purple values highlight the best performance.*

### E.1.2 Standard MG Benchmark Results

Table 7 presents management-task performance on the MMST *Standard-MG* benchmark, reporting four rubric scores—**Accuracy**, **Relevance**, **Completeness**, and **Parsimony** (0–4 scale)—under the same trio of automated judges used for the ID setting. Consistent with the ID results, the proprietary GPT-4.1 model dominates most of metrics across all judges, outscoring the best open-source competitor by roughly $0.4$ absolute points in Accuracy and by $0.3$–$0.5$ in the Relevance and Completeness.

Table 7: Performance Comparison of Large Vision Language Models on MMST (Standard-MG) Benchmark

| Model | DeepSeek-R1-Distill | | | | Qwen3-32B | | | | Phi-4-reasoning | | | |
|---|---|---|---|---|---|---|---|---|---|---|---|---|
| | Acc | Rel | Comp | Pars | Acc | Rel | Comp | Pars | Acc | Rel | Comp | Pars |
| gpt-4.1 | **3.17** | **3.59** | **3.39** | 2.93 | **3.27** | **3.59** | **3.03** | 2.97 | **3.27** | **3.62** | **3.24** | **3.14** |
| gpt-4.1-mini | 2.93 | 3.39 | 3.05 | 2.91 | 2.92 | 3.31 | 2.54 | 2.97 | 2.96 | 3.40 | 2.89 | 3.06 |
| gpt-4o | 2.75 | 3.14 | 2.57 | 2.91 | 2.77 | 3.10 | 2.17 | **3.07** | 2.78 | 3.23 | 2.55 | 3.03 |
| gpt-4o-mini | 2.65 | 3.01 | 2.44 | 2.70 | 2.64 | 2.95 | 2.01 | 2.83 | 2.66 | 3.10 | 2.40 | 2.81 |
| claude-3-7-sonnet | 2.83 | 3.29 | 2.96 | 2.83 | 2.82 | 3.17 | 2.41 | 2.89 | 2.81 | 3.23 | 2.71 | 2.91 |
| claude-3-5-sonnet | 2.75 | 3.22 | 2.80 | 2.88 | 2.75 | 3.11 | 2.29 | 2.94 | 2.74 | 3.18 | 2.62 | 2.95 |
| claude-3-haiku | 2.44 | 2.92 | 2.22 | 2.74 | 2.37 | 2.78 | 1.77 | 2.82 | 2.40 | 2.98 | 2.15 | 2.96 |
| Llama-4-Scout-17B-16E-Instruct | 2.53 | 2.96 | 2.48 | 2.55 | 2.51 | 2.86 | 1.99 | 2.63 | 2.49 | 2.98 | 2.34 | 2.64 |
| llava-v1.6-mistral-7b-hf | 2.22 | 2.55 | 2.05 | 2.14 | 2.20 | 2.44 | 1.60 | 2.26 | 2.17 | 2.50 | 1.93 | 2.19 |
| llava-onevision-qwen2-7b-ov-hf | 2.25 | 2.61 | 2.13 | 2.16 | 2.22 | 2.50 | 1.66 | 2.29 | 2.23 | 2.59 | 2.02 | 2.23 |
| Qwen2.5-VL-3B-Instruct | 2.10 | 2.42 | 1.96 | 2.01 | 2.12 | 2.35 | 1.55 | 2.17 | 2.04 | 2.37 | 1.83 | 2.07 |
| Qwen2.5-VL-7B-Instruct | 2.39 | 2.76 | 2.36 | 2.26 | 2.39 | 2.65 | 1.84 | 2.34 | 2.36 | 2.76 | 2.22 | 2.33 |
| Qwen2.5-VL-32B-Instruct | 2.84 | 3.25 | 3.14 | 2.52 | 2.80 | 3.11 | 2.56 | 2.31 | 2.85 | 3.22 | 2.93 | 2.45 |
| Qwen2.5-VL-72B-Instruct | 2.70 | 3.10 | 2.79 | 2.56 | 2.72 | 3.02 | 2.25 | 2.60 | 2.74 | 3.15 | 2.64 | 2.65 |
| gemma-3-4b-it | 2.33 | 2.82 | 2.60 | 2.18 | 2.28 | 2.61 | 2.04 | 2.06 | 2.23 | 2.69 | 2.31 | 2.25 |
| gemma-3-12b-it | 2.66 | 3.11 | 3.01 | 2.38 | 2.62 | 2.89 | 2.49 | 2.13 | 2.61 | 3.01 | 2.72 | 2.40 |
| gemma-3-27b-it | 2.80 | 3.25 | 3.15 | 2.52 | 2.79 | 3.05 | 2.65 | 2.29 | 2.71 | 3.11 | 2.82 | 2.48 |
| InternVL3-2B | 2.12 | 2.47 | 1.98 | 2.07 | 2.11 | 2.36 | 1.56 | 2.25 | 2.05 | 2.41 | 1.86 | 2.14 |
| InternVL3-8B | 2.36 | 2.75 | 2.28 | 2.39 | 2.34 | 2.64 | 1.83 | 2.50 | 2.34 | 2.74 | 2.19 | 2.50 |
| InternVL3-14B | 2.49 | 2.88 | 2.39 | 2.58 | 2.50 | 2.79 | 1.95 | 2.70 | 2.47 | 2.88 | 2.30 | 2.69 |
| InternVL3-38B | 2.56 | 2.96 | 2.44 | 2.66 | 2.56 | 2.89 | 2.01 | 2.80 | 2.56 | 3.01 | 2.39 | 2.78 |
| InternVL3-78B | 2.60 | 2.99 | 2.48 | **2.97** | 2.59 | 2.90 | 2.03 | 2.82 | 2.60 | 3.04 | 2.42 | 2.82 |

*Models are color-coded by type: closed-source models in red/light red, open-source models in blue/light blue. Scores represent performance on four key metrics (Accuracy / Relevance / Completeness / Parsimony) on a **0-4** scale. Bold purple values highlight the best performance on each metric within each benchmark.*

### E.1.3 Contextual MG Benchmark Results

Table 8: Performance Comparison of Large Vision-Language Models on MMST (Contextual) Benchmark

| Model | DeepSeek-R1-Distill | | | | Qwen3-32B | | | | Phi-4-reasoning | | | |
|---|---|---|---|---|---|---|---|---|---|---|---|---|
| | Acc | Rel | Comp | Pars | Acc | Rel | Comp | Pars | Acc | Rel | Comp | Pars |
| gpt-4.1 | **3.21** | **3.61** | **3.40** | 2.94 | **3.28** | **3.59** | **3.01** | 2.96 | **3.29** | **3.61** | **3.24** | **3.14** |
| gpt-4.1-mini | 2.89 | 3.37 | 3.01 | 2.89 | 2.88 | 3.26 | 2.43 | 2.93 | 2.92 | 3.36 | 2.82 | 3.03 |
| gpt-4o | 2.73 | 3.07 | 2.46 | 2.85 | 2.73 | 3.00 | 2.03 | **3.00** | 2.74 | 3.17 | 2.44 | 2.98 |
| gpt-4o-mini | 2.66 | 2.97 | 2.36 | 2.67 | 2.62 | 2.87 | 1.92 | 2.81 | 2.63 | 3.03 | 2.31 | 2.76 |
| claude-3-7-sonnet | 2.85 | 3.31 | 2.94 | 2.87 | 2.79 | 3.13 | 2.35 | 2.89 | 2.80 | 3.24 | 2.68 | 2.90 |
| claude-3-5-sonnet | 2.79 | 3.24 | 2.80 | 2.92 | 2.75 | 3.09 | 2.26 | 2.96 | 2.76 | 3.20 | 2.62 | 2.95 |
| claude-3-haiku | 2.41 | 2.83 | 2.08 | 2.63 | 2.30 | 2.65 | 1.66 | 2.77 | 2.37 | 2.89 | 2.03 | 2.88 |
| Llama-4-Scout-17B-16E-Instruct | 2.53 | 2.95 | 2.38 | 2.58 | 2.47 | 2.78 | 1.88 | 2.67 | 2.48 | 2.95 | 2.28 | 2.71 |
| llava-v1.6-mistral-7b-hf | 2.29 | 2.57 | 2.03 | 2.17 | 2.22 | 2.44 | 1.57 | 2.31 | 2.22 | 2.53 | 1.91 | 2.24 |
| llava-onevision-qwen2-7b-ov-hf | 2.26 | 2.57 | 2.01 | 2.18 | 2.22 | 2.46 | 1.57 | 2.34 | 2.22 | 2.58 | 1.92 | 2.30 |
| Qwen2.5-VL-3B-Instruct | 2.15 | 2.43 | 1.94 | 2.00 | 2.10 | 2.30 | 1.48 | 2.16 | 2.04 | 2.36 | 1.80 | 2.05 |
| Qwen2.5-VL-7B-Instruct | 2.30 | 2.61 | 2.13 | 2.20 | 2.28 | 2.50 | 1.65 | 2.32 | 2.24 | 2.60 | 2.03 | 2.32 |
| Qwen2.5-VL-32B-Instruct | 2.87 | 3.24 | 3.11 | 2.52 | 2.75 | 3.06 | 2.44 | 2.26 | 2.86 | 3.21 | 2.89 | 2.43 |
| Qwen2.5-VL-72B-Instruct | 2.72 | 3.07 | 2.75 | 2.52 | 2.68 | 2.95 | 2.12 | 2.53 | 2.71 | 3.09 | 2.57 | 2.59 |
| gemma-3-4b-it | 2.34 | 2.83 | 2.57 | 2.19 | 2.24 | 2.56 | 1.94 | 2.00 | 2.22 | 2.69 | 2.26 | 2.22 |
| gemma-3-12b-it | 2.76 | 3.17 | 3.09 | 2.42 | 2.69 | 2.94 | 2.52 | 2.10 | 2.71 | 3.08 | 2.80 | 2.39 |
| gemma-3-27b-it | 2.82 | 3.28 | 3.19 | 2.54 | 2.79 | 3.05 | 2.63 | 2.28 | 2.74 | 3.12 | 2.83 | 2.48 |
| InternVL3-2B | 2.22 | 2.51 | 2.02 | 2.08 | 2.16 | 2.37 | 1.56 | 2.22 | 2.13 | 2.46 | 1.88 | 2.13 |
| InternVL3-8B | 2.86 | 2.76 | 2.26 | 2.45 | 2.37 | 2.64 | 1.76 | 2.56 | 2.36 | 2.76 | 2.14 | 2.53 |
| InternVL3-14B | 2.52 | 2.85 | 2.37 | 2.53 | 2.49 | 2.75 | 1.87 | 2.64 | 2.49 | 2.87 | 2.26 | 2.62 |
| InternVL3-38B | 2.57 | 2.90 | 2.39 | 2.60 | 2.54 | 2.81 | 1.91 | 2.73 | 2.55 | 2.94 | 2.31 | 2.72 |
| InternVL3-78B | 2.57 | 2.92 | 2.36 | **3.05** | 2.54 | 2.81 | 1.90 | 2.78 | 2.55 | 2.95 | 2.30 | 2.77 |

*Models are color-coded by type: closed-source models in red/light red, open-source models in blue/light blue. Scores represent performance on four key metrics (Accuracy / Relevance / Completeness / Parsimony) on a **0-4** scale. Bold purple values denote the best score for each metric across all models.*

## E.2 Model Scaling Results

Increasing model scale consistently boosts both identification accuracy and reasoning quality for all three open-source LVLM families (See Figure 9).

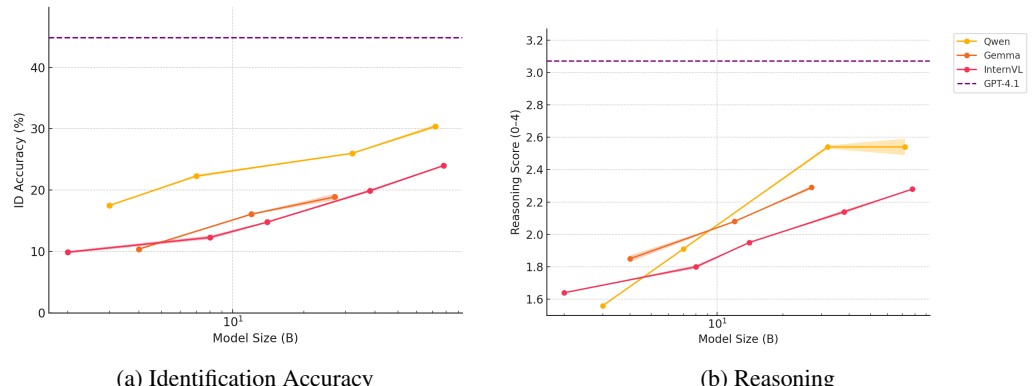

(a) Identification Accuracy         (b) Reasoning

Figure 9: Performance scaling of open-source LVLM families—Qwen (yellow), Gemma (orange), and InternVL (red)—on the standard identification benchmark (Judge: DeepSeek-R1-Distill). Shaded bands denote ±1 std dev across three runs. The dashed purple line shows the closed-source GPT-4.1 result for comparison.

## E.3 With/Without Meta Data Results

**Utility of metadata.** Tables 9 and 10 compare model performance when each question is presented *with* versus *without* the user's location and timestamp. Metadata is incorporated by appending this spatiotemporal context in natural-language form to the image–text input during inference. For the **Identification** benchmark, gains are modest ($\leq 1.6$,pp ID%) with negligible changes in reasoning quality ($\leq 0.05$). Stronger models (GPT-4.1, Qwen-72B) show slight improvements (+1.6 and +0.6,pp), whereas smaller ones (GPT-4.1-mini, Gemma-3B) show minimal or negative shifts, indicating that metadata can add noise without sufficient model capacity.

A similar pattern emerges on the **Management** benchmark (Table 10): absolute deltas across accuracy, relevance, completeness, and parsimony remain within $\pm 0.04$, yet the direction of change is revealing. Large models (GPT-4.1, Gemma-27B, Qwen-72B) exhibit consistent, albeit small, improvements, most notably in *relevance* and *completeness*, while smaller models fluctuate or even decline. Overall, these findings indicate that spatiotemporal cues confer a measurable but limited advantage, and that **leveraging them effectively remains contingent on model scale and training.**

Table 9: Impact of metadata (location + time) on identification accuracy (ID%) and reasoning ability across models on MIRAGE-MMST Standard Identification Benchmark (Judge: DeepSeek-R1-Distill). The table compares performance in the **Image + Text Only** setting and the **Metadata-Augmented** setting. Arrows (↑/↓) indicate change from baseline. Absolute $\Delta$ values are reported on the right.

| Model | Image + Text Only | | + Metadata | | $\Delta$ (Meta – No Meta) | |
|---|---|---|---|---|---|---|
| | ID% | Reason | ID% | Reason | $\Delta$ ID% | $\Delta$ Reason |
| gpt-4.1 | 44.60 | 3.07 | 46.2↑ | 3.12↑ | 1.60 | 0.05 |
| gpt-4.1-mini | 36.30 | 2.79 | 35.6↓ | 2.81↑ | −0.60 | 0.02 |
| gemma-3-4b-it | 10.40 | 1.87 | 10.2↓ | 1.88↑ | −0.20 | 0.01 |
| gemma-3-27b-it | 19.30 | 2.28 | 19.0↓ | 2.33↑ | −0.20 | 0.04 |
| Qwen2.5-VL-3B-Instruct | 17.40 | 1.55 | 18.0↑ | 1.57↑ | 0.60 | 0.02 |
| Qwen2.5-VL-32B-Instruct | 26.10 | 2.54 | 25.7↓ | 2.54 | −0.30 | 0.00 |
| Qwen2.5-VL-72B-Instruct | 30.80 | 2.59 | 31.4↑ | 2.59 | 0.60 | 0.00 |

Table 10: Performance of large vision–language models on the MIRAGE-MMST Standard Management benchmark (Judge: DeepSeek-R1-Distill), comparing the standard setting (image + text only) against a metadata-augmented setting (including geographic location and time). Results are reported over four metrics: accuracy (Acc), relevance (Rel), completeness (Comp), and parsimony (Pars). Arrows in the metadata columns indicate the direction of change, and absolute $\Delta$ values are reported.

| Model | Image + Text Only | | | | + Metadata | | | | $\Delta$ (Meta – No Meta) | | | |
|---|---|---|---|---|---|---|---|---|---|---|---|---|
| | Acc | Rel | Comp | Pars | Acc | Rel | Comp | Pars | $\Delta$ A | $\Delta$ R | $\Delta$ C | $\Delta$ P |
| gpt-4.1 | 3.17 | 3.59 | 3.39 | 2.93 | 3.21↑ | 3.62↑ | 3.41↑ | 2.96↑ | 0.04 | 0.03 | 0.02 | 0.03 |
| gpt-4.1-mini | 2.93 | 3.39 | 3.05 | 2.91 | 2.90↓ | 3.41↑ | 3.05↓ | 2.94↑ | −0.03 | 0.02 | 0.00 | 0.03 |
| gemma-3-4b-it | 2.33 | 2.82 | 2.60 | 2.18 | 2.32↓ | 2.83↑ | 2.62↑ | 2.18 | −0.01 | 0.01 | 0.02 | 0.00 |
| gemma-3-27b-it | 2.80 | 3.25 | 3.15 | 2.52 | 2.81↑ | 3.28↑ | 3.19↑ | 2.55↑ | 0.01 | 0.03 | 0.04 | 0.03 |
| Qwen2.5-VL-3B-Instruct | 2.10 | 2.42 | 1.96 | 2.01 | 2.13↑ | 2.44↑ | 1.99↑ | 2.00↓ | 0.03 | 0.02 | 0.03 | −0.01 |
| Qwen2.5-VL-32B-Instruct | 2.84 | 3.25 | 3.14 | 2.52 | 2.83↓ | 3.25 | 3.14 | 2.54↑ | −0.01 | 0.00 | 0.00 | 0.02 |
| Qwen2.5-VL-72B-Instruct | 2.70 | 3.10 | 2.79 | 2.56 | 2.71↑ | 3.12↑ | 2.83↑ | 2.56 | 0.01 | 0.02 | 0.04 | 0.00 |

## E.4 Fine-Turning Results

**Fine-tuning setup.** All models are adapted on the MMST *standard* training dataset, which contains 17 532 single-turn consultations, each paired with up to three images. Given the limited corpus size, we employ parameter-efficient LoRA fine-tuning: a global batch size of 128, `lora_alpha = 64`, `lora_dropout = 0.05`, and `bfloat16` precision. Optimisation uses AdamW with a cosine learning-rate schedule and a warm-up ratio of 0.03. Hardware resources: one NVIDIA H200 GPU suffices for the QwenVL-2.5-3 B and 7 B models, while the 32 B model is trained on two H200 cards—enabling the full eight-epoch run.

**Effect of LoRA fine-tuning.** Figure 10 tracks identification accuracy and reasoning accuracy on *seen* vs. *unseen* entities, as Qwen2.5-VL models undergo progressively longer LoRA fine-tuning. For both the 32 B (left) and 7 B (right) variants, the bulk of the improvement is realised within the first four epochs: ID accuracy on seen entities rises from 32.9% to 37.6% for 32 B and from 27.7% to 34.8% for 7 B. Beyond epoch 4 the gains plateau or slightly regress, hinting at *diminishing returns* and possible over-fitting to the fine-tuning set. Reasoning accuracy follow a similar but more muted trend, increasing by at most 0.2–0.3 points before flattening. The persistently low curves for *unseen* entities are unsurprising. Identification requires the model to emit an explicit entity name; if that name never appeared in the fine-tuning set, fine-tuning does not help.

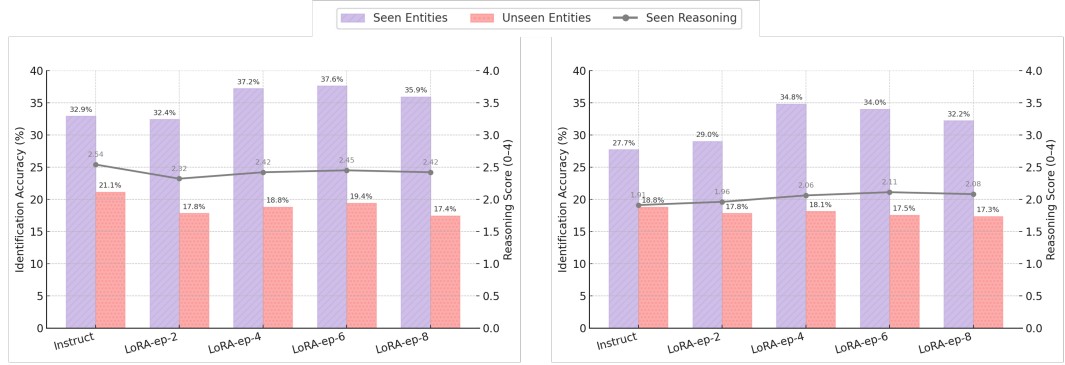

Figure 10: LoRA fine-tuning results on identification accuracy and reasoning on Standard Identification Benchmark (Judge: DeepSeek-R1-Distill). Bars show ID Accuracy (%) on *seen entities* and *unseen entities* for Qwen2.5-VL-32B **(Left)** & 7B **(Right)** at epochs: Instruct (0), LoRA-ep-2, 4, 6, 8. The grey line marker ● traces the Reasoning Score (0–4 scale) on seen entities. Values above each bar/point give the exact percentages and scores.

# F  Error Analysis

We conducted a quantitative error analysis to complement our main evaluation results for deeper insights into model failure modes. We performed a stratified manual error review across three scenarios: (i) GPT-4.1 failures ($n = 100$ of 2,395), (ii) Qwen2.5-VL-32B failures where GPT-4.1 succeeds ($n = 100$ of 1,078), and (iii) joint failures ($n = 100$ of 300). We focused on instances with zero identification accuracy and compared the failure modes and root causes between the proprietary GPT-4.1 and the open-source Qwen2.5-VL-32B. The observed errors cluster into two tiers: *Fundamental Domain Challenges* and *Open-Source Model Systematic Failures*.

## F.1  Tier 1: Fundamental Domain Challenges

These represent the current boundaries of what even the most advanced LVLMs can achieve in agricultural reasoning and consultation contexts. Failures in this tier occur in GPT-4.1, the best-performing model in our evaluation, indicating inherent limitations of current LVLMs in agricultural expert reasoning. Percentages reflect the relative frequency of each error type within sampled failures

- **Specialist Knowledge Gaps (35% of cases):** These involve missing domain-specific understanding of biological phenomena requiring deep expertise beyond general training data. The LVLMs fail to recognize or reason about biological processes. *Example: GPT-4.1 could not interpret "appleleaf blister mite" damage or its developmental process.*

- **Complex Diagnostic Reasoning (10% of cases):** These involve correct symptom observation but incorrect causal inference or disease conclusion. Despite accurate visual analysis, the models fail to reason over underlying biological causality. *Example: Observing insect damage but misdiagnosing the pest species.*

- **Finegrained visual understanding (55% of cases):** Majority of the errors occured due to model's inability to pick up finegrained visual cues and distinguishing factors among different adjacent biological species or part occlusions of the main entity in images.

## F.2  Tier 2: Open-Source Model Systematic Failures

These failure types occur primarily in Qwen2.5-VL-32B but not in GPT-4.1, revealing systematic capability gaps between open-source and proprietary LVLMs.

- **Vision-Language Integration Issues (40% of cases):** Failures in integrating visual cues with domain-specific terminology lead to misidentification among visually similar species. Domain-specific finetuning mitigates this issue. *Example: Qwen2.5-VL-32B confusing a longhorn beetle with a June beetle.*

- **Diagnostic Hedging Bias (25% of cases):** The open-source model frequently exhibits hedging behavior, avoiding decisive expert-level conclusions.

- **Generalist Reasoning Bias (20% of cases):** The model applies broad, non-specific reasoning, listing multiple generic possibilities instead of providing a targeted diagnosis.

## F.3  Quantitative Error Pattern Summary

**Critical Finding:** In joint failures, GPT-4.1 still demonstrates superior reasoning quality in 60% of cases, reflecting deeper integration of visual and textual cues and more expert-like reasoning patterns, even when incorrect in final identification.

## F.4  Performance on Common and Rare Entities

The MIRAGE dataset exhibits a long-tail entity distribution, where models consistently perform worse on rare entities than on common ones. This imbalance exposes a central challenge for agricultural AI: models must recognize thousands of rare species that occur infrequently in training data. Addressing this limitation will require domain adaptation and long-tail learning approaches beyond standard large-scale training.

Table 11: Quantitative distribution of failure categories.

| Failure Category | Percentage | Model |
|---|---|---|
| **Tier 1: Fundamental Domain Challenges (GPT-4.1 Failures)** | | |
| Specialist Knowledge Gaps | 35% | GPT-4.1 |
| Complex Diagnostic Reasoning | 10% | GPT-4.1 |
| Finegrained visual understanding | 55% | GPT-4.1 |
| **Tier 2: Open-Source Model Systematic Failures (Qwen2.5-VL-32B vs. GPT-4.1)** | | |
| Vision-Language Integration Issues | 40% | Qwen2.5-VL-32B |
| Diagnostic Hedging Bias | 25% | Qwen2.5-VL-32B |
| Generalist Reasoning Bias | 20% | Qwen2.5-VL-32B |

Table 12: Performance (%) and average reasoning score on common entities.

| Model | DeepSeek-R1 | Qwen3-32B | Phi-4-reasoning |
|---|---|---|---|
| GPT-4.1 | 53.1, 3.15 | 52.7, 2.84 | 52.2, 3.20 |
| GPT-4.1-mini | 42.1, 2.82 | 41.3, 2.53 | 40.5, 2.95 |
| Qwen2.5-VL-32B-Instruct | 32.9, 2.62 | 32.2, 2.23 | 31.6, 2.62 |
| Qwen2.5-VL-72B-Instruct | 38.1, 2.65 | 38.4, 2.27 | 37.1, 2.63 |

Table 13: Performance (%) and average reasoning score on rare entities.

| Model | DeepSeek-R1 | Qwen3-32B | Phi-4-reasoning |
|---|---|---|---|
| GPT-4.1 | 38.5, 3.01 | 38.9, 2.73 | 35.4, 3.14 |
| GPT-4.1-mini | 32.1, 2.76 | 30.4, 2.50 | 26.8, 2.94 |
| Qwen2.5-VL-32B-Instruct | 21.1, 2.47 | 20.4, 2.14 | 18.2, 2.53 |
| Qwen2.5-VL-72B-Instruct | 25.5, 2.56 | 24.5, 2.18 | 22.1, 2.57 |

Building on these findings, we outline a roadmap for advancing multimodal large vision-language models (VLMs) toward robust, interactive agricultural consultation systems. Future versions MI-RAGE should integrate **agentic capabilities** that enable models to proactively utilize **time- and location-based context**—for example, adapting responses based on seasonal patterns, regional crop profiles, or current environmental conditions. This contextual grounding will allow evaluations to move beyond static reasoning and toward dynamic, situation-aware dialogue.

To achieve this, future VLM development should emphasize the following directions:

- **Contextual and Temporal Grounding:** Incorporate temporal signals (e.g., crop cycles, weather timelines) and geospatial information (e.g., soil types, local pest occurrences) to support temporally and regionally coherent reasoning in consultation tasks.

- **Interactive Conversational Reasoning:** Extend MIRAGE's multi-turn benchmark to simulate fully interactive dialogues, where models must perform clarification, conversational repair, and feedback incorporation, key aspects of real-world agricultural advisory systems.

- **Domain-Specific Knowledge Integration:** Bridge VLMs with structured agronomic knowledge graphs and expert-curated datasets to reduce specialist knowledge gaps identified in our error analysis.

- **Adaptive Visual-Linguistic Understanding:** Improve VLMs' capacity to detect fine-grained visual cues and align them with domain terminology, addressing the vision-language integration failures seen in open-source models.

- **Long-Tail Entity Recognition:** Develop targeted fine-tuning and retrieval-augmented methods for handling rare species and diseases that dominate real-world agricultural problem distributions.

These extensions will move MIRAGE and future VLMs from static perception and reasoning benchmarks toward **fully interactive, contextually grounded expert systems**. Such systems could ultimately assist human agricultural specialists in diagnosing issues, providing adaptive recommendations, and integrating multimodal data streams for decision support in diverse agricultural environments.

## F.5 Category-Wise Breakdown Results

**Contents**

### F.5.1 Plant Identification Reults (MMST Standard)

Table 14: Performance Comparison of Large Language Models on the MMST Standard Benchmark Results for **Plant Identification**

| Model | DeepSeek-R1-Distill | | Qwen3-32B | | Phi-4-reasoning | |
|---|---|---|---|---|---|---|
| | Acc (%) | Reasoning | Acc (%) | Reasoning | Acc (%) | Reasoning |
| gpt-4.1 | **48.7** | **3.13** | **49.0** | **2.89** | **45.9** | **3.28** |
| gpt-4.1-mini | 38.4 | 2.84 | 37.2 | 2.60 | 34.6 | 3.04 |
| gpt-4o | 44.5 | 2.58 | 44.8 | 2.40 | 39.7 | 2.75 |
| gpt-4o-mini | 26.8 | 2.23 | 25.2 | 2.09 | 22.2 | 2.41 |
| claude-3-7-sonnet | 35.8 | 2.76 | 36.3 | 2.51 | 34.8 | 2.95 |
| claude-3-5-sonnet | 35.0 | 2.67 | 35.0 | 2.42 | 33.9 | 2.79 |
| claude-3-haiku | 19.6 | 1.81 | 18.8 | 1.70 | 15.3 | 1.96 |
| Llama-4-Scout-17B-16E-Instruct | 20.1 | 2.12 | 21.4 | 2.01 | 18.1 | 2.28 |
| llava-v1.6-mistral-7b-hf | 6.7 | 1.33 | 6.3 | 1.19 | 5.2 | 1.49 |
| llava-onevision-qwen2-7b-ov-hf | 9.6 | 1.63 | 8.8 | 1.54 | 7.9 | 1.76 |
| Qwen2.5-VL-3B-Instruct | 20.3 | 1.56 | 20.8 | 1.44 | 18.3 | 1.57 |
| Qwen2.5-VL-7B-Instruct | 25.8 | 1.94 | 27.3 | 1.81 | 23.6 | 2.03 |
| Qwen2.5-VL-32B-Instruct | 26.3 | 2.52 | 25.9 | 2.23 | 24.1 | 2.61 |
| Qwen2.5-VL-72B-Instruct | 33.1 | 2.57 | 33.1 | 2.31 | 30.2 | 2.69 |
| gemma-3-4b-it | 11.4 | 1.85 | 11.8 | 1.74 | 11.0 | 2.03 |
| gemma-3-12b-it | 17.0 | 2.08 | 16.8 | 1.93 | 16.0 | 2.17 |
| gemma-3-27b-it | 20.0 | 2.29 | 20.4 | 2.11 | 18.7 | 2.47 |
| InternVL3-2B | 9.4 | 1.61 | 8.1 | 1.57 | 7.3 | 1.80 |
| InternVL3-8B | 10.8 | 1.77 | 10.8 | 1.68 | 9.9 | 1.91 |
| InternVL3-14B | 11.8 | 1.89 | 11.5 | 1.79 | 10.9 | 2.03 |
| InternVL3-38B | 18.4 | 2.10 | 18.6 | 2.01 | 16.1 | 2.30 |
| InternVL3-78B | 22.3 | 2.24 | 21.6 | 2.11 | 19.5 | 2.41 |

*Models are color-coded by type: closed-source models in red/light red, open-source models in blue/light blue. Bold purple values denote the best performance in each column.*

## F.5.2 Insect and Pest Identification Results (MMST Standard)

Table 15: Performance Comparison of Large Language Models on the MMST Standard Benchmark for **Insect and Pest Identification**

| Model | DeepSeek-R1-Distill | | Qwen3-32B | | Phi-4-reasoning | |
|---|---|---|---|---|---|---|
| | Acc (%) | Reasoning | Acc (%) | Reasoning | Acc (%) | Reasoning |
| gpt-4.1 | **34.9** | **2.93** | **34.9** | **2.60** | **32.6** | **3.01** |
| gpt-4.1-mini | 30.1 | 2.68 | 28.6 | 2.38 | 26.0 | 2.83 |
| gpt-4o | 33.0 | 2.43 | 31.2 | 2.16 | 27.3 | 2.53 |
| gpt-4o-mini | 17.5 | 2.12 | 14.9 | 1.88 | 12.8 | 2.23 |
| claude-3-7-sonnet | 27.5 | 2.56 | 27.4 | 2.19 | 24.5 | 2.60 |
| claude-3-5-sonnet | 24.4 | 2.42 | 24.5 | 2.06 | 23.6 | 2.41 |
| claude-3-haiku | 15.4 | 1.83 | 15.1 | 1.54 | 13.3 | 1.84 |
| Llama-4-Scout-17B-16E-Instruct | 16.8 | 2.10 | 16.3 | 1.89 | 14.1 | 2.15 |
| llava-v1.6-mistral-7b-hf | 7.2 | 1.35 | 6.1 | 1.15 | 5.1 | 1.42 |
| llava-onevision-qwen2-7b-ov-hf | 8.6 | 1.57 | 7.7 | 1.33 | 6.5 | 1.60 |
| Qwen2.5-VL-3B-Instruct | 13.5 | 1.48 | 13.8 | 1.26 | 11.9 | 1.45 |
| Qwen2.5-VL-7B-Instruct | 18.3 | 1.90 | 18.2 | 1.60 | 15.8 | 1.92 |
| Qwen2.5-VL-32B-Instruct | 23.9 | 2.52 | 23.1 | 2.13 | 20.2 | 2.54 |
| Qwen2.5-VL-72B-Instruct | 25.6 | 2.46 | 23.6 | 2.11 | 22.5 | 2.49 |
| gemma-3-4b-it | 5.8 | 1.85 | 6.2 | 1.65 | 5.8 | 1.84 |
| gemma-3-12b-it | 10.2 | 1.96 | 10.8 | 1.60 | 10.6 | 1.79 |
| gemma-3-27b-it | 14.4 | 2.16 | 13.8 | 1.89 | 13.4 | 2.16 |
| InternVL3-2B | 9.2 | 1.66 | 7.9 | 1.48 | 6.3 | 1.74 |
| InternVL3-8B | 11.2 | 1.78 | 11.5 | 1.57 | 9.8 | 1.76 |
| InternVL3-14B | 13.9 | 1.92 | 13.0 | 1.69 | 11.9 | 1.95 |
| InternVL3-38B | 17.5 | 2.12 | 17.3 | 1.87 | 15.3 | 2.20 |
| InternVL3-78B | 22.1 | 2.29 | 19.5 | 2.03 | 17.9 | 2.30 |

*Models are color-coded by type: closed-source models in red/light red, open-source models in blue/light blue. Bold purple values denote the best performance in each column.*

### F.5.3 Plant Disease Identification Results (MMST Standard)

Table 16: Performance Comparison of Large Language Models on the MMST Standard Benchmark for **Plant Disease Identification**

| Model | DeepSeek-R1-Distill | | Qwen3-32B | | Phi-4-reasoning | |
|---|---|---|---|---|---|---|
| | Acc (%) | Reasoning | Acc (%) | Reasoning | Acc (%) | Reasoning |
| gpt-4.1 | **45.7** | **3.06** | **44.6** | **2.62** | **46.0** | **2.97** |
| gpt-4.1-mini | 38.8 | 2.79 | 37.7 | 2.37 | 36.5 | 2.77 |
| gpt-4o | 40.5 | 2.46 | 40.0 | 2.02 | 41.0 | 2.46 |
| gpt-4o-mini | 26.5 | 2.15 | 24.9 | 1.88 | 27.2 | 2.22 |
| claude-3-7-sonnet | 40.8 | 2.77 | 40.7 | 2.28 | 41.0 | 2.58 |
| claude-3-5-sonnet | 35.6 | 2.62 | 36.2 | 2.15 | 35.3 | 2.48 |
| claude-3-haiku | 21.5 | 1.91 | 23.2 | 1.54 | 22.8 | 1.91 |
| Llama-4-Scout-17B-16E-Instruct | 30.4 | 2.26 | 30.3 | 1.90 | 29.4 | 2.19 |
| llava-v1.6-mistral-7b-hf | 13.0 | 1.48 | 13.7 | 1.15 | 13.3 | 1.47 |
| llava-onevision-qwen2-7b-ov-hf | 13.8 | 1.71 | 15.7 | 1.30 | 18.3 | 1.67 |
| Qwen2.5-VL-3B-Instruct | 12.3 | 1.63 | 14.5 | 1.25 | 13.8 | 1.52 |
| Qwen2.5-VL-7B-Instruct | 15.9 | 1.77 | 15.4 | 1.38 | 16.1 | 1.70 |
| Qwen2.5-VL-32B-Instruct | 29.4 | 2.61 | 27.0 | 2.04 | 29.4 | 2.44 |
| Qwen2.5-VL-72B-Instruct | 30.6 | 2.95 | 31.0 | 2.03 | 31.1 | 2.41 |
| gemma-3-4b-it | 15.4 | 1.99 | 14.7 | 1.61 | 15.6 | 1.77 |
| gemma-3-12b-it | 23.5 | 2.30 | 22.3 | 1.78 | 24.2 | 2.05 |
| gemma-3-27b-it | 25.6 | 2.48 | 24.4 | 1.96 | 25.1 | 2.18 |
| InternVL3-2B | 14.0 | 1.76 | 14.5 | 1.47 | 15.9 | 1.73 |
| InternVL3-8B | 20.1 | 2.04 | 19.7 | 1.60 | 21.1 | 1.86 |
| InternVL3-14B | 29.8 | 2.23 | 28.4 | 1.78 | 29.2 | 2.08 |
| InternVL3-38B | 31.8 | 2.30 | 29.6 | 1.90 | 30.3 | 2.22 |
| InternVL3-78B | 35.1 | 2.44 | 33.2 | 1.96 | 32.7 | 2.33 |

*Models are color-coded by type: closed-source models in red/light red, open-source models in blue/light blue. Bold purple values denote the best performance in each column.*

### F.5.4   Plant Disease Management Results (MMST Standard)

Table 17: Performance Comparison of Large Vision–Language Models on MMST Standard Benchmark for **Plant Disease Management**

| Model | DeepSeek-R1-Distill | | | | Qwen3-32B | | | | Phi-4-reasoning | | | |
|---|---|---|---|---|---|---|---|---|---|---|---|---|
| | Acc | Rel | Comp | Pars | Acc | Rel | Comp | Pars | Acc | Rel | Comp | Pars |
| gpt-4.1 | **3.05** | **3.54** | **3.31** | **2.87** | **3.17** | **3.54** | **2.90** | **2.92** | **3.15** | **3.53** | **3.12** | **3.05** |
| gpt-4.1-mini | 2.79 | 3.28 | 2.92 | 2.77 | 2.81 | 3.20 | 2.37 | 2.90 | 2.78 | 3.25 | 2.72 | 2.91 |
| gpt-4o | 2.65 | 3.06 | 2.48 | 2.80 | 2.68 | 3.01 | 2.06 | 2.98 | 2.64 | 3.10 | 2.39 | 2.89 |
| gpt-4o-mini | 2.55 | 2.89 | 2.31 | 2.54 | 2.56 | 2.84 | 1.90 | 2.73 | 2.52 | 2.95 | 2.24 | 2.65 |
| claude-3-7-sonnet | 2.67 | 3.19 | 2.82 | 2.71 | 2.66 | 3.06 | 2.25 | 2.79 | 2.61 | 3.07 | 2.55 | 2.81 |
| claude-3-5-sonnet | 2.59 | 3.10 | 2.63 | 2.80 | 2.58 | 2.98 | 2.11 | 2.84 | 2.50 | 3.00 | 2.41 | 2.84 |
| claude-3-haiku | 2.35 | 2.86 | 2.12 | 2.60 | 2.26 | 2.70 | 1.65 | 2.73 | 2.25 | 2.85 | 2.03 | 2.82 |
| Llama-4-Scout-17B-16E-Instruct | 2.45 | 2.91 | 2.32 | 2.49 | 2.42 | 2.80 | 1.85 | 2.63 | 2.36 | 2.88 | 2.20 | 2.61 |
| llava-v1.6-mistral-7b-hf | 2.13 | 2.49 | 1.96 | 2.05 | 2.08 | 2.37 | 1.48 | 2.18 | 2.04 | 2.42 | 1.83 | 2.14 |
| llava-onevision-qwen2-7b-ov-hf | 2.16 | 2.55 | 2.02 | 2.08 | 2.13 | 2.44 | 1.55 | 2.26 | 2.07 | 2.48 | 1.88 | 2.17 |
| Qwen2.5-VL-3B-Instruct | 2.00 | 2.33 | 1.85 | 1.94 | 1.99 | 2.27 | 1.44 | 2.14 | 1.87 | 2.24 | 1.68 | 2.01 |
| Qwen2.5-VL-7B-Instruct | 2.30 | 2.66 | 2.25 | 2.19 | 2.26 | 2.57 | 1.72 | 2.28 | 2.19 | 2.62 | 2.06 | 2.25 |
| Qwen2.5-VL-32B-Instruct | 2.73 | 3.19 | 3.05 | 2.43 | 2.67 | 3.03 | 2.37 | 2.20 | 2.68 | 3.11 | 2.77 | 2.30 |
| Qwen2.5-VL-72B-Instruct | 2.58 | 2.99 | 2.63 | 2.46 | 2.61 | 2.92 | 2.09 | 2.52 | 2.54 | 2.98 | 2.49 | 2.54 |
| gemma-3-4b-it | 2.06 | 2.63 | 1.85 | 2.04 | 1.99 | 2.37 | 1.73 | 1.98 | 1.92 | 2.33 | 1.77 | 2.07 |
| gemma-3-12b-it | 2.48 | 2.99 | 2.84 | 2.29 | 2.40 | 2.75 | 2.26 | 2.04 | 2.31 | 2.78 | 2.47 | 2.26 |
| gemma-3-27b-it | 2.65 | 3.15 | 3.03 | 2.43 | 2.61 | 2.91 | 2.45 | 2.23 | 2.50 | 2.94 | 2.61 | 2.35 |
| InternVL3-2B | 2.07 | 2.47 | 1.95 | 2.06 | 2.04 | 2.34 | 1.48 | 2.20 | 1.92 | 2.33 | 1.77 | 2.07 |
| InternVL3-8B | 2.26 | 2.67 | 2.19 | 2.31 | 2.25 | 2.56 | 1.72 | 2.43 | 2.19 | 2.61 | 2.05 | 2.41 |
| InternVL3-14B | 2.43 | 2.83 | 2.31 | 2.48 | 2.42 | 2.73 | 1.84 | 2.63 | 2.34 | 2.80 | 2.20 | 2.59 |
| InternVL3-38B | 2.51 | 2.90 | 2.38 | 2.57 | 2.51 | 2.84 | 1.93 | 2.73 | 2.47 | 2.91 | 2.28 | 2.72 |
| InternVL3-78B | 2.54 | 2.94 | 2.39 | 2.64 | 2.51 | 2.83 | 1.92 | 2.76 | 2.50 | 2.93 | 2.28 | 2.73 |

*Models are color-coded by type: closed-source models in red/light red, open-source models in blue/light blue. Scores are given on a **0–4** scale for Accuracy (Acc), Relevance (Rel), Completeness (Comp), and Parsimony (Pars). Bold purple numbers denote the best performance for each metric within a column block.*

### F.5.5   Insect and Pest Management Results (MMST Standard)

Table 18: Performance Comparison of Large Vision–Language Models on MMST Standard Benchmark for **Insect and Pest Management**

| Model | DeepSeek-R1-Distill | | | | Qwen3-32B | | | | Phi-4-reasoning | | | |
|---|---|---|---|---|---|---|---|---|---|---|---|---|
| | Acc | Rel | Comp | Pars | Acc | Rel | Comp | Pars | Acc | Rel | Comp | Pars |
| gpt-4.1 | 3.05 | 3.51 | 3.33 | 2.87 | 3.18 | 3.53 | 2.98 | 2.98 | 3.11 | 3.52 | 3.15 | 3.06 |
| gpt-4.1-mini | 2.86 | 3.36 | 3.01 | 2.95 | 2.76 | 3.21 | 2.45 | 2.92 | 2.72 | 3.23 | 2.72 | 2.94 |
| gpt-4o | 2.62 | 3.05 | 2.48 | 2.80 | 2.66 | 3.04 | 2.11 | 3.01 | 2.58 | 3.09 | 2.43 | 2.90 |
| gpt-4o-mini | 2.47 | 2.88 | 2.31 | 2.58 | 2.45 | 2.81 | 1.89 | 2.74 | 2.40 | 2.89 | 2.21 | 2.59 |
| nova-pro | 2.12 | 2.57 | 1.98 | 2.34 | 2.13 | 2.45 | 1.57 | 2.50 | 2.07 | 2.49 | 1.88 | 2.50 |
| claude-3-7-sonnet | 2.64 | 3.14 | 2.83 | 2.71 | 2.66 | 3.03 | 2.30 | 2.80 | 2.57 | 2.99 | 2.53 | 2.71 |
| claude-3-5-sonnet | 2.57 | 3.07 | 2.70 | 2.69 | 2.63 | 3.02 | 2.24 | 2.85 | 2.54 | 2.95 | 2.49 | 2.72 |
| claude-3-haiku | 2.27 | 2.76 | 2.11 | 2.59 | 2.42 | 2.66 | 1.70 | 2.70 | 2.23 | 2.78 | 2.04 | 2.76 |
| Llama-4-Scout-17B-16E-Instruct | 1.29 | 2.77 | 2.34 | 2.42 | 1.33 | 2.71 | 1.90 | 2.54 | 1.18 | 2.69 | 2.12 | 2.42 |
| llava-v1.6-mistral-7b-hf | 1.95 | 2.32 | 1.81 | 1.97 | 1.98 | 2.21 | 1.44 | 2.16 | 1.83 | 2.10 | 1.62 | 1.93 |
| llava-onevision-qwen2-7b-ov-hf | 1.98 | 2.35 | 1.91 | 1.96 | 1.99 | 2.28 | 1.52 | 2.16 | 1.88 | 2.19 | 1.73 | 1.91 |
| Qwen2.5-VL-3B-Instruct | 1.89 | 2.25 | 1.78 | 1.90 | 1.94 | 2.19 | 1.43 | 2.09 | 1.74 | 2.06 | 1.60 | 1.83 |
| Qwen2.5-VL-7B-Instruct | 2.13 | 2.55 | 2.16 | 2.10 | 2.19 | 2.47 | 1.70 | 2.24 | 2.04 | 2.42 | 1.94 | 2.07 |
| Qwen2.5-VL-32B-Instruct | 2.59 | 3.07 | 2.95 | 2.36 | 2.62 | 2.96 | 2.43 | 2.26 | 2.66 | 3.06 | 2.82 | 2.39 |
| Qwen2.5-VL-72B-Instruct | 2.48 | 2.96 | 2.63 | 2.45 | 2.56 | 2.90 | 2.16 | 2.56 | 2.45 | 2.95 | 2.45 | 2.46 |
| gemma-3-4b-it | 1.99 | 2.51 | 2.28 | 1.94 | 1.99 | 2.34 | 1.81 | 1.93 | 1.81 | 2.24 | 1.93 | 1.98 |
| gemma-3-12b-it | 2.29 | 2.83 | 2.70 | 2.15 | 2.28 | 2.62 | 2.22 | 2.03 | 2.15 | 2.58 | 2.31 | 2.13 |
| gemma-3-27b-it | 2.47 | 2.98 | 2.87 | 2.32 | 2.51 | 2.84 | 2.43 | 2.22 | 2.31 | 2.74 | 2.47 | 2.23 |
| InternVL3-2B | 1.87 | 2.22 | 1.77 | 1.90 | 1.91 | 2.13 | 1.40 | 2.11 | 1.73 | 2.06 | 1.58 | 1.88 |
| InternVL3-8B | 2.06 | 2.53 | 2.08 | 2.22 | 2.07 | 2.44 | 1.64 | 2.40 | 1.99 | 2.39 | 1.93 | 2.24 |
| InternVL3-14B | 2.19 | 2.64 | 2.19 | 2.40 | 2.25 | 2.60 | 1.79 | 2.59 | 2.11 | 2.49 | 2.00 | 2.44 |
| InternVL3-38B | 2.32 | 2.80 | 2.31 | 2.50 | 2.38 | 2.73 | 1.90 | 2.72 | 2.24 | 2.72 | 2.17 | 2.55 |
| InternVL3-78B | 2.36 | 2.81 | 2.33 | 2.54 | 2.41 | 2.75 | 1.92 | 2.72 | 2.31 | 2.77 | 2.23 | 2.62 |

*Models are color-coded by type: closed-source (red) and open-source (blue). Scores range from 0-4 for Accuracy (Acc), Relevance (Rel), Completeness (Comp), and Parsimony (Pars). Bold purple indicates the best score in each column block.*

### F.5.6 Plant Care and Gardening Guidance Results (MMST Standard)

Table 19: Performance Comparison of Large Vision–Language Models on MMST Standard Benchmark for **Plant Care and Gardening Guidance**

| Model | DeepSeek-R1-Distill | | | | Qwen3-32B | | | | Phi-4-reasoning | | | |
|---|---|---|---|---|---|---|---|---|---|---|---|---|
| | Acc | Rel | Comp | Pars | Acc | Rel | Comp | Pars | Acc | Rel | Comp | Pars |
| gpt-4.1 | **3.38** | **3.73** | **3.55** | **3.01** | **3.45** | **3.73** | **3.17** | **3.12** | **3.52** | **3.80** | **3.43** | **3.24** |
| gpt-4.1-mini | 3.15 | 3.54 | 3.23 | 3.01 | 3.15 | 3.50 | 2.72 | 3.03 | 3.28 | 3.64 | 3.15 | 3.20 |
| gpt-4o | 2.93 | 3.26 | 2.69 | 3.01 | 2.93 | 3.21 | 2.28 | 3.12 | 3.03 | 3.41 | 2.74 | 3.16 |
| gpt-4o-mini | 2.85 | 3.18 | 2.64 | 2.87 | 2.84 | 3.12 | 2.17 | 2.94 | 2.95 | 3.35 | 2.65 | 3.03 |
| nova-pro | 2.60 | 2.95 | 2.31 | 2.67 | 2.52 | 2.81 | 1.87 | 2.73 | 2.63 | 3.04 | 2.30 | 2.88 |
| claude-3-7-sonnet | 3.08 | 3.48 | 3.15 | 3.00 | 3.06 | 3.39 | 2.59 | 3.02 | 3.14 | 3.53 | 2.97 | 3.12 |
| claude-3-5-sonnet | 3.02 | 3.41 | 2.99 | 3.05 | 3.00 | 3.32 | 2.47 | 3.07 | 3.09 | 3.49 | 2.87 | 3.17 |
| claude-3-haiku | 2.66 | 3.10 | 2.39 | 2.93 | 2.56 | 2.95 | 1.92 | 2.94 | 2.67 | 3.25 | 2.35 | 3.16 |
| Llama-4-Scout-17B-16E-Instruct | 2.79 | 3.15 | 2.69 | 2.68 | 2.75 | 3.02 | 2.15 | 2.70 | 2.85 | 3.29 | 2.62 | 2.82 |
| llava-v1.6-mistral-7b-hf | 2.50 | 2.77 | 2.27 | 2.28 | 2.47 | 2.66 | 1.80 | 2.35 | 2.50 | 2.82 | 2.19 | 2.37 |
| llava-onevision-qwen2-7b-ov-hf | 2.55 | 2.86 | 2.39 | 2.35 | 2.50 | 2.73 | 1.85 | 2.39 | 2.61 | 2.96 | 2.33 | 2.47 |
| Qwen2.5-VL-3B-Instruct | 2.33 | 2.61 | 2.15 | 2.15 | 2.34 | 2.52 | 1.67 | 2.25 | 2.37 | 2.67 | 2.09 | 2.27 |
| Qwen2.5-VL-7B-Instruct | 2.65 | 2.97 | 2.55 | 2.41 | 2.63 | 2.83 | 2.00 | 2.42 | 2.71 | 3.08 | 2.49 | 2.54 |
| Qwen2.5-VL-32B-Instruct | 3.14 | 3.45 | 3.40 | 2.68 | 3.07 | 3.32 | 2.80 | 2.39 | 3.21 | 3.50 | 3.22 | 2.62 |
| Qwen2.5-VL-72B-Instruct | 2.96 | 3.29 | 3.02 | 2.69 | 2.95 | 3.18 | 2.42 | 2.65 | 3.09 | 3.42 | 2.92 | 2.80 |
| gemma-3-4b-it | 2.77 | 3.14 | 2.97 | 2.40 | 2.68 | 2.96 | 2.21 | 2.57 | 2.76 | 3.20 | 2.76 | 2.54 |
| gemma-3-12b-it | 3.09 | 3.41 | 3.36 | 2.60 | 3.03 | 3.19 | 2.84 | 2.25 | 3.16 | 3.48 | 3.23 | 2.64 |
| gemma-3-27b-it | 3.17 | 3.53 | 3.45 | 2.72 | 3.14 | 3.32 | 2.96 | 2.39 | 3.19 | 3.53 | 3.23 | 2.72 |
| InternVL3-2B | 2.38 | 2.67 | 2.18 | 2.22 | 2.34 | 2.54 | 1.71 | 2.35 | 2.38 | 2.73 | 2.11 | 2.35 |
| InternVL3-8B | 2.67 | 2.99 | 2.52 | 2.57 | 2.63 | 2.88 | 2.04 | 2.61 | 2.74 | 3.11 | 2.49 | 2.72 |
| InternVL3-14B | 2.78 | 3.11 | 2.62 | 2.77 | 2.76 | 3.01 | 2.15 | 2.82 | 2.87 | 3.26 | 2.60 | 2.91 |
| InternVL3-38B | 2.82 | 3.15 | 2.62 | 2.84 | 2.79 | 3.06 | 2.16 | 2.89 | 2.90 | 3.30 | 2.63 | 2.97 |
| InternVL3-78B | 2.84 | 3.18 | 2.67 | 2.88 | 2.82 | 3.08 | 2.20 | 2.92 | 2.93 | 3.32 | 2.67 | 3.01 |

*Models are color-coded by type: closed-source (red) and open-source (blue). Each metric is scored on a 0–4 scale: Accuracy (Acc), Relevance (Rel), Completeness (Comp), and Parsimony (Pars). Bold purple values indicate the best performance within each column block.*

### F.5.7 Weeds/Invasive Plants Management Results (MMST Standard)

Table 20: Performance Comparison of Large Vision–Language Models on MMST Standard Benchmark for **Weeds/Invasive Plants Management**

| Model | DeepSeek-R1-Distill | | | | Qwen3-32B | | | | Phi-4-reasoning | | | |
|---|---|---|---|---|---|---|---|---|---|---|---|---|
| | Acc | Rel | Comp | Pars | Acc | Rel | Comp | Pars | Acc | Rel | Comp | Pars |
| gpt-4.1 | **2.90** | **3.41** | **3.14** | 2.85 | **3.01** | **3.37** | **2.87** | 2.96 | **2.89** | **3.40** | **3.00** | 3.10 |
| gpt-4.1-mini | 2.62 | 3.19 | 2.80 | 2.82 | 2.67 | 3.09 | 2.43 | 2.97 | 2.60 | 3.14 | 2.67 | 3.06 |
| gpt-4o | 2.56 | 3.06 | 2.50 | **2.95** | 2.62 | 3.04 | 2.18 | **3.17** | 2.59 | 3.12 | 2.46 | **3.11** |
| gpt-4o-mini | 2.43 | 2.88 | 2.25 | 2.65 | 2.39 | 2.80 | 1.91 | 2.81 | 2.36 | 2.92 | 2.22 | 2.79 |
| nova-pro | 1.92 | 2.33 | 1.78 | 2.28 | 1.95 | 2.33 | 1.53 | 2.51 | 1.84 | 2.33 | 1.74 | 2.50 |
| claude-3-7-sonnet | 2.59 | 3.06 | 2.82 | 2.72 | 2.56 | 2.93 | 2.28 | 2.82 | 2.48 | 2.94 | 2.49 | 2.73 |
| claude-3-5-sonnet | 2.48 | 3.03 | 2.66 | 2.74 | 2.46 | 2.84 | 2.19 | 2.82 | 2.35 | 2.86 | 2.39 | 2.77 |
| claude-3-haiku | 2.19 | 2.67 | 2.05 | 2.64 | 2.14 | 2.58 | 1.66 | 2.81 | 2.11 | 2.68 | 1.95 | 2.92 |
| Llama-4-Scout-17B-16E-Instruct | 2.18 | 2.71 | 2.31 | 2.38 | 2.20 | 2.65 | 1.87 | 2.52 | 2.06 | 2.63 | 2.08 | 2.44 |
| llava-v1.6-mistral-7b-hf | 1.90 | 2.29 | 1.82 | 2.07 | 1.89 | 2.19 | 1.47 | 2.27 | 1.85 | 2.23 | 1.73 | 2.09 |
| llava-onevision-qwen2-7b-ov-hf | 1.85 | 2.28 | 1.83 | 1.99 | 1.80 | 2.16 | 1.45 | 2.22 | 1.81 | 2.17 | 1.73 | 2.04 |
| Qwen2.5-VL-3B-Instruct | 1.86 | 2.20 | 1.82 | 1.88 | 1.92 | 2.20 | 1.53 | 2.11 | 1.79 | 2.09 | 1.68 | 1.87 |
| Qwen2.5-VL-7B-Instruct | 2.10 | 2.57 | 2.23 | 2.14 | 2.14 | 2.48 | 1.81 | 2.34 | 2.03 | 2.50 | 2.06 | 2.21 |
| Qwen2.5-VL-32B-Instruct | 2.42 | 2.96 | 2.78 | 2.37 | 2.47 | 2.86 | 2.33 | 2.32 | 2.33 | 2.80 | 2.48 | 2.34 |
| Qwen2.5-VL-72B-Instruct | 2.43 | 2.92 | 2.64 | 2.52 | 2.48 | 2.85 | 2.20 | 2.67 | 2.44 | 2.90 | 2.49 | 2.65 |
| gemma-3-4b-it | 1.95 | 2.55 | 2.38 | 2.11 | 1.96 | 2.38 | 1.90 | 2.03 | 1.87 | 2.37 | 2.06 | 2.14 |
| gemma-3-12b-it | 2.19 | 2.78 | 2.66 | 2.21 | 2.21 | 2.60 | 2.20 | 2.12 | 2.11 | 2.55 | 2.31 | 2.28 |
| gemma-3-27b-it | 2.36 | 2.92 | 2.82 | 2.37 | 2.39 | 2.76 | 2.40 | 2.21 | 2.18 | 2.68 | 2.46 | 2.32 |
| InternVL3-2B | 1.74 | 2.13 | 1.71 | 1.87 | 1.79 | 2.17 | 1.45 | 2.24 | 1.68 | 2.01 | 1.61 | 1.96 |
| InternVL3-8B | 1.94 | 2.43 | 2.00 | 2.24 | 1.97 | 2.34 | 1.66 | 2.47 | 1.83 | 2.31 | 1.86 | 2.34 |
| InternVL3-14B | 2.07 | 2.56 | 2.13 | 2.44 | 2.13 | 2.48 | 1.75 | 2.62 | 1.96 | 2.38 | 1.95 | 2.51 |
| InternVL3-38B | 2.19 | 2.71 | 2.20 | 2.54 | 2.18 | 2.65 | 1.85 | 2.76 | 2.11 | 2.66 | 2.13 | 2.66 |
| InternVL3-78B | 2.25 | 2.76 | 2.25 | 2.54 | 2.26 | 2.67 | 1.87 | 2.78 | 2.17 | 2.72 | 2.16 | 2.68 |

*Models are color-coded by type: closed-source (red) and open-source (blue). Scores range from 0–4 across Accuracy (Acc), Relevance (Rel), Completeness (Comp), and Parsimony (Pars). Bold purple marks the highest score for each metric within a column block.*

## F.5.8 Plant Disease Management Results (MMST Contextual)

Table 21: Performance Comparison of Large Vision–Language Models on MMST Contextual Benchmark for **Plant Disease Management**

| Model | DeepSeek-R1-Distill | | | | Qwen3-32B | | | | Phi-4-reasoning | | | |
|---|---|---|---|---|---|---|---|---|---|---|---|---|
| | Acc | Rel | Comp | Pars | Acc | Rel | Comp | Pars | Acc | Rel | Comp | Pars |
| gpt-4.1 | 3.07 | 3.52 | 3.34 | 2.86 | 3.16 | 3.49 | 2.91 | 2.86 | 3.09 | 3.44 | 3.08 | 2.96 |
| gpt-4.1-mini | 2.74 | 3.20 | 2.90 | 2.70 | 2.68 | 3.08 | 2.24 | 2.74 | 2.63 | 3.07 | 2.58 | 2.78 |
| gpt-4o | 2.62 | 2.96 | 2.40 | 2.68 | 2.60 | 2.89 | 1.89 | 3.17 | 2.55 | 2.96 | 2.28 | 2.78 |
| gpt-4o-mini | 2.57 | 2.85 | 2.30 | 2.48 | 2.52 | 2.75 | 1.80 | 2.65 | 2.46 | 2.81 | 2.15 | 2.51 |
| nova-pro | 2.17 | 2.55 | 1.94 | 2.37 | 2.08 | 2.37 | 1.43 | 2.48 | 2.05 | 2.47 | 1.81 | 2.60 |
| claude-3-7-sonnet | 2.66 | 3.16 | 2.83 | 2.68 | 2.58 | 2.97 | 2.18 | 2.71 | 2.52 | 2.95 | 2.47 | 2.67 |
| claude-3-5-sonnet | 2.59 | 3.06 | 2.67 | 2.73 | 2.54 | 2.89 | 2.05 | 2.77 | 2.46 | 2.91 | 2.37 | 2.74 |
| claude-3-haiku | 2.32 | 2.75 | 2.03 | 2.48 | 2.18 | 2.56 | 1.53 | 2.64 | 2.20 | 2.73 | 1.89 | 2.71 |
| Llama-4-Scout-17B-16E-Instruct | 2.36 | 2.79 | 2.24 | 2.43 | 2.27 | 2.59 | 1.67 | 2.52 | 2.24 | 2.69 | 2.07 | 2.59 |
| llava-v1.6-mistral-7b-hf | 2.18 | 2.47 | 1.97 | 2.04 | 2.07 | 2.33 | 1.43 | 2.18 | 2.04 | 2.36 | 1.75 | 2.07 |
| llava-onevision-qwen2-7b-ov-hf | 2.17 | 2.50 | 1.95 | 2.05 | 2.09 | 2.35 | 1.45 | 2.19 | 2.04 | 2.38 | 1.78 | 2.11 |
| Qwen2.5-VL-3B-Instruct | 2.04 | 2.33 | 1.89 | 1.92 | 1.96 | 2.21 | 1.35 | 2.04 | 1.86 | 2.16 | 1.67 | 1.92 |
| Qwen2.5-VL-7B-Instruct | 2.15 | 2.49 | 2.04 | 2.06 | 2.09 | 2.33 | 1.47 | 2.18 | 1.99 | 2.33 | 1.81 | 2.13 |
| Qwen2.5-VL-32B-Instruct | 2.69 | 3.07 | 2.98 | 2.37 | 2.54 | 2.85 | 2.22 | 2.06 | 2.56 | 2.92 | 2.62 | 2.18 |
| Qwen2.5-VL-72B-Instruct | 2.59 | 2.91 | 2.64 | 2.35 | 2.49 | 2.77 | 1.94 | 2.34 | 2.44 | 2.82 | 2.33 | 2.33 |
| gemma-3-4b-it | 2.07 | 2.60 | 2.02 | 2.04 | 1.93 | 2.27 | 1.63 | 1.94 | 1.87 | 2.31 | 1.93 | 1.94 |
| gemma-3-12b-it | 2.52 | 3.03 | 2.93 | 2.29 | 2.38 | 2.68 | 2.20 | 1.93 | 2.27 | 2.67 | 2.39 | 2.13 |
| gemma-3-27b-it | 2.57 | 3.09 | 2.99 | 2.38 | 2.48 | 2.78 | 2.29 | 2.12 | 2.33 | 2.72 | 2.42 | 2.23 |
| InternVL3-2B | 2.18 | 2.48 | 2.06 | 2.01 | 2.07 | 2.31 | 1.47 | 2.10 | 1.98 | 2.32 | 1.80 | 2.02 |
| InternVL3-8B | 2.31 | 2.63 | 2.21 | 2.29 | 2.22 | 2.48 | 1.63 | 2.40 | 2.16 | 2.52 | 1.96 | 2.32 |
| InternVL3-14B | 2.42 | 2.76 | 2.32 | 2.38 | 2.36 | 2.59 | 1.74 | 2.44 | 2.29 | 2.66 | 2.12 | 2.43 |
| InternVL3-38B | 2.47 | 2.79 | 2.34 | 2.45 | 2.39 | 2.66 | 1.76 | 2.56 | 2.36 | 2.72 | 2.14 | 2.50 |
| InternVL3-78B | 2.46 | 2.79 | 2.30 | 2.48 | 2.40 | 2.66 | 1.76 | 2.61 | 2.33 | 2.72 | 2.12 | 2.58 |

*Models are color-coded by type: closed-source (red) and open-source (blue). Scores range from 0–4 across Accuracy (Acc), Relevance (Rel), Completeness (Comp), and Parsimony (Pars). Bold purple marks the highest score for each metric within a column block.*

### F.5.9    Insect and Pest Management Results (MMST Contextual)

Table 22: Performance Comparison of Large Vision–Language Models on MMST Contextual Benchmark for Insect and Pest Management

| Model | DeepSeek-R1-Distill | | | | Qwen3-32B | | | | Phi-4-reasoning | | | |
|---|---|---|---|---|---|---|---|---|---|---|---|---|
| | Acc | Rel | Comp | Pars | Acc | Rel | Comp | Pars | Acc | Rel | Comp | Pars |
| gpt-4.1 | **3.14** | **3.59** | **3.32** | **2.99** | **3.25** | **3.61** | **2.99** | **3.06** | **3.22** | **3.60** | **3.20** | **3.21** |
| gpt-4.1-mini | 2.73 | 3.33 | 2.91 | 2.88 | 2.80 | 3.22 | 2.42 | 2.97 | 2.79 | 3.30 | 2.75 | 3.02 |
| gpt-4o | 2.66 | 3.07 | 2.44 | 2.89 | 2.71 | 3.03 | 2.08 | 3.03 | 2.63 | 3.16 | 2.42 | 3.00 |
| gpt-4o-mini | 2.52 | 2.88 | 2.25 | 2.62 | 2.50 | 2.82 | 1.88 | 2.78 | 2.41 | 2.92 | 2.18 | 2.70 |
| nova-pro | 2.13 | 2.58 | 1.95 | 2.46 | 2.17 | 2.49 | 1.62 | 2.64 | 2.10 | 2.59 | 1.91 | 2.70 |
| claude-3-7-sonnet | 2.74 | 3.27 | 2.89 | 2.81 | 2.74 | 3.13 | 2.35 | 2.92 | 2.68 | 3.18 | 2.63 | 2.87 |
| claude-3-5-sonnet | 2.68 | 3.19 | 2.76 | 2.87 | 2.67 | 3.02 | 2.22 | 2.92 | 2.60 | 3.11 | 2.57 | 2.84 |
| claude-3-haiku | 2.30 | 2.75 | 1.98 | 2.59 | 2.22 | 2.61 | 1.63 | 2.74 | 2.22 | 2.76 | 1.95 | 2.78 |
| Llama-4-Scout-17B-16E-Instruct | 2.37 | 2.87 | 2.30 | 2.71 | 2.37 | 2.75 | 1.89 | 2.63 | 2.28 | 2.82 | 2.20 | 2.61 |
| llava-v1.6-mistral-7b-hf | 1.99 | 2.35 | 1.78 | 2.04 | 1.99 | 2.24 | 1.44 | 2.23 | 1.92 | 2.21 | 1.68 | 2.01 |
| llava-onevision-qwen2-7b-ov-hf | 1.97 | 2.33 | 1.81 | 1.99 | 1.99 | 2.26 | 1.42 | 2.19 | 1.87 | 2.22 | 1.67 | 2.02 |
| Qwen2.5-VL-3B-Instruct | 1.97 | 2.28 | 1.80 | 1.90 | 1.98 | 2.20 | 1.43 | 2.10 | 1.79 | 2.14 | 1.59 | 1.87 |
| Qwen2.5-VL-7B-Instruct | 2.13 | 2.54 | 2.05 | 2.12 | 2.18 | 2.43 | 1.65 | 2.32 | 2.05 | 2.45 | 1.91 | 2.17 |
| Qwen2.5-VL-32B-Instruct | 2.62 | 3.11 | 2.91 | 2.44 | 2.57 | 2.97 | 2.36 | 2.29 | 2.60 | 3.05 | 2.71 | 2.35 |
| Qwen2.5-VL-72B-Instruct | 2.56 | 3.03 | 2.66 | 2.50 | 2.62 | 2.95 | 2.14 | 2.60 | 2.52 | 3.01 | 2.47 | 2.58 |
| gemma-3-4b-it | 1.95 | 2.60 | 2.35 | 2.02 | 1.97 | 2.40 | 1.84 | 1.92 | 1.94 | 2.41 | 2.03 | 2.02 |
| gemma-3-12b-it | 2.43 | 2.94 | 2.80 | 2.26 | 2.45 | 2.77 | 2.34 | 2.04 | 2.35 | 2.75 | 2.38 | 2.22 |
| gemma-3-27b-it | 2.57 | 3.09 | 2.98 | 2.41 | 2.61 | 2.93 | 2.51 | 2.27 | 2.47 | 2.89 | 2.60 | 2.38 |
| InternVL3-2B | 1.94 | 2.28 | 1.79 | 1.92 | 1.95 | 2.20 | 1.45 | 2.14 | 1.81 | 2.15 | 1.65 | 1.91 |
| InternVL3-8B | 2.05 | 2.58 | 2.05 | 2.26 | 2.19 | 2.45 | 1.67 | 2.45 | 2.04 | 2.48 | 1.93 | 2.34 |
| InternVL3-14B | 2.26 | 2.67 | 2.18 | 2.40 | 2.31 | 2.62 | 1.79 | 2.60 | 2.22 | 2.61 | 2.06 | 2.42 |
| InternVL3-38B | 2.31 | 2.74 | 2.21 | 2.50 | 2.36 | 2.70 | 1.86 | 2.70 | 2.28 | 2.72 | 2.11 | 2.59 |
| InternVL3-78B | 2.42 | 2.85 | 2.29 | 2.62 | 2.43 | 2.77 | 1.88 | 2.81 | 2.35 | 2.81 | 2.19 | 2.69 |

*Models are color-coded by type: closed-source (red) and open-source (blue). Scores range from 0–4 for Accuracy (Acc), Relevance (Rel), Completeness (Comp), and Parsimony (Pars). Bold purple highlights the best score for each metric within a column block.*

### F.5.10 Plant Care and Gardening Guidance Results (MMST Contextual)

Table 23: Performance Comparison of Large Vision–Language Models on MMST Contextual Benchmark for **Plant Care and Gardening Guidance**

| Model | DeepSeek-R1-Distill | | | | Qwen3-32B | | | | Phi-4-reasoning | | | |
|---|---|---|---|---|---|---|---|---|---|---|---|---|
| | Acc | Rel | Comp | Pars | Acc | Rel | Comp | Pars | Acc | Rel | Comp | Pars |
| gpt-4.1 | **3.40** | **3.73** | **3.54** | 3.03 | **3.46** | **3.72** | **3.16** | 3.03 | **3.53** | **3.80** | **3.42** | **3.27** |
| gpt-4.1-mini | 3.13 | 3.55 | 3.19 | 3.05 | 3.11 | 3.47 | 2.65 | 3.07 | 3.27 | 3.64 | 3.09 | 3.23 |
| gpt-4o | 2.87 | 3.20 | 2.57 | 2.97 | 2.86 | 3.11 | 2.16 | **3.08** | 2.95 | 3.35 | 2.62 | 3.11 |
| gpt-4o-mini | 2.82 | 3.13 | 2.50 | 2.85 | 2.79 | 3.03 | 2.06 | 2.93 | 2.90 | 3.29 | 2.55 | 2.98 |
| nova-pro | 2.59 | 2.92 | 2.24 | 2.72 | 2.52 | 2.77 | 1.81 | 2.82 | 2.59 | 3.02 | 2.25 | 2.94 |
| claude-3-7-sonnet | 3.09 | 3.48 | 3.13 | 3.03 | 3.02 | 3.33 | 2.53 | 3.01 | 3.12 | 3.53 | 2.92 | 3.09 |
| claude-3-5-sonnet | 3.02 | 3.44 | 2.98 | 3.09 | 2.99 | 3.31 | 2.46 | 3.12 | 3.07 | 3.48 | 2.87 | 3.13 |
| claude-3-haiku | 2.57 | 2.97 | 2.19 | 2.81 | 2.45 | 2.78 | 1.79 | 2.87 | 2.59 | 3.11 | 2.19 | 3.04 |
| Llama-4-Scout-17B-16E-Instruct | 2.77 | 3.13 | 2.57 | 2.74 | 2.70 | 2.98 | 2.05 | 2.78 | 2.78 | 3.24 | 2.49 | 2.86 |
| llava-v1.6-mistral-7b-hf | 2.52 | 2.77 | 2.20 | 2.32 | 2.45 | 2.63 | 1.73 | 2.41 | 2.49 | 2.82 | 2.13 | 2.43 |
| llava-onevision-qwen2-7b-ov-hf | 2.51 | 2.79 | 2.21 | 2.39 | 2.45 | 2.65 | 1.75 | 2.51 | 2.55 | 2.90 | 2.17 | 2.55 |
| Qwen2.5-VL-3B-Instruct | 2.36 | 2.60 | 2.11 | 2.14 | 2.29 | 2.45 | 1.61 | 2.26 | 2.33 | 2.64 | 2.01 | 2.24 |
| Qwen2.5-VL-7B-Instruct | 2.52 | 2.79 | 2.30 | 2.36 | 2.48 | 2.67 | 1.81 | 2.43 | 2.54 | 2.91 | 2.27 | 2.51 |
| Qwen2.5-VL-32B-Instruct | 3.14 | 3.46 | 3.35 | 2.69 | 3.02 | 3.29 | 2.70 | 2.38 | 3.23 | 3.53 | 3.21 | 2.63 |
| Qwen2.5-VL-72B-Instruct | 2.92 | 3.26 | 2.94 | 2.67 | 2.89 | 3.12 | 2.29 | 2.62 | 3.04 | 3.38 | 2.84 | 2.79 |
| gemma-3-4b-it | 2.71 | 3.13 | 2.88 | 2.39 | 2.58 | 2.87 | 2.26 | 2.16 | 2.67 | 3.16 | 2.67 | 2.50 |
| gemma-3-12b-it | 3.12 | 3.43 | 3.39 | 2.61 | 3.06 | 3.21 | 2.87 | 2.25 | 3.21 | 3.54 | 3.25 | 2.65 |
| gemma-3-27b-it | 3.18 | 3.54 | 3.48 | 2.73 | 3.15 | 3.34 | 2.99 | 2.41 | 3.20 | 3.54 | 3.25 | 2.71 |
| InternVL3-2B | 2.45 | 2.70 | 2.17 | 2.24 | 2.38 | 2.55 | 1.70 | 2.33 | 2.42 | 2.74 | 2.09 | 2.33 |
| InternVL3-8B | 2.69 | 2.99 | 2.45 | 2.66 | 2.61 | 2.86 | 1.94 | 2.72 | 2.70 | 3.11 | 2.42 | 2.77 |
| InternVL3-14B | 2.76 | 3.05 | 2.54 | 2.74 | 2.72 | 2.95 | 2.04 | 2.81 | 2.80 | 3.20 | 2.51 | 2.87 |
| InternVL3-38B | 2.79 | 3.10 | 2.55 | 2.76 | 2.75 | 3.00 | 2.07 | 2.87 | 2.85 | 3.26 | 2.56 | 2.94 |
| InternVL3-78B | 2.78 | 3.10 | 2.51 | **3.50** | 2.74 | 2.99 | 2.05 | 2.91 | 2.85 | 3.24 | 2.52 | 2.99 |

*Models are color-coded by type: closed-source (red) and open-source (blue). Scores range from 0–4 for Accuracy (Acc), Relevance (Rel), Completeness (Comp), and Parsimony (Pars). Bold purple highlights the best score for each metric within a column block.*

### F.5.11 Weeds/Invasive Plants Management Results (MMST Contextual)

Table 24: Performance Comparison of Large Vision–Language Models on MMST Contextual Benchmark for **Weeds/Invasive Plants Management**

| Model | DeepSeek-R1-Distill | | | | Qwen3-32B | | | | Phi-4-reasoning | | | |
|---|---|---|---|---|---|---|---|---|---|---|---|---|
| | Acc | Rel | Comp | Pars | Acc | Rel | Comp | Pars | Acc | Rel | Comp | Pars |
| gpt-4.1 | **3.03** | **3.53** | **3.27** | **2.92** | **3.12** | **3.46** | **2.92** | **3.03** | **3.02** | **3.45** | **3.07** | **3.20** |
| gpt-4.1-mini | 2.74 | 3.31 | 2.89 | 2.93 | 2.74 | 3.19 | 2.43 | 3.01 | 2.69 | 3.25 | 2.75 | 3.18 |
| gpt-4o | 2.62 | 3.13 | 2.49 | 3.05 | 2.65 | 3.06 | 2.13 | 3.19 | 2.63 | 3.17 | 2.47 | 3.23 |
| gpt-4o-mini | 2.47 | 2.90 | 2.32 | 2.75 | 2.44 | 2.82 | 1.96 | 2.90 | 2.45 | 2.95 | 2.28 | 2.92 |
| nova-pro | 2.08 | 2.53 | 1.96 | 2.52 | 2.12 | 2.48 | 1.66 | 2.67 | 2.06 | 2.57 | 1.97 | 2.82 |
| claude-3-7-sonnet | 2.75 | 3.30 | 2.96 | 2.90 | 2.73 | 3.09 | 2.41 | 2.93 | 2.66 | 3.12 | 2.67 | 2.98 |
| claude-3-5-sonnet | 2.68 | 3.22 | 2.81 | 2.93 | 2.67 | 3.07 | 2.36 | 3.02 | 2.65 | 3.14 | 2.66 | 3.05 |
| claude-3-haiku | 2.27 | 2.78 | 2.06 | 2.75 | 2.19 | 2.59 | 1.69 | 2.86 | 2.23 | 2.79 | 2.04 | 3.03 |
| Llama-4-Scout-17B-16E-Instruct | 2.34 | 2.87 | 2.39 | 2.55 | 2.34 | 2.80 | 2.00 | 2.73 | 2.25 | 2.82 | 2.26 | 2.73 |
| llava-v1.6-mistral-7b-hf | 1.98 | 2.38 | 1.87 | 2.18 | 1.99 | 2.32 | 1.58 | 2.38 | 1.93 | 2.30 | 1.77 | 2.23 |
| llava-onevision-qwen2-7b-ov-hf | 1.92 | 2.31 | 1.77 | 2.04 | 1.92 | 2.27 | 1.47 | 2.33 | 1.86 | 2.22 | 1.71 | 2.12 |
| Qwen2.5-VL-3B-Instruct | 1.88 | 2.26 | 1.83 | 1.92 | 1.93 | 2.23 | 1.54 | 2.17 | 1.78 | 2.12 | 1.67 | 1.89 |
| Qwen2.5-VL-7B-Instruct | 2.10 | 2.50 | 2.13 | 2.20 | 2.20 | 2.50 | 1.76 | 2.44 | 2.06 | 2.54 | 2.03 | 2.33 |
| Qwen2.5-VL-32B-Instruct | 2.57 | 3.09 | 2.88 | 2.50 | 2.58 | 2.95 | 2.42 | 2.43 | 2.56 | 2.97 | 2.68 | 2.49 |
| Qwen2.5-VL-72B-Instruct | 2.55 | 3.02 | 2.69 | 2.63 | 2.60 | 2.99 | 2.26 | 2.73 | 2.54 | 3.03 | 2.56 | 2.77 |
| gemma-3-4b-it | 2.12 | 2.69 | 2.51 | 2.18 | 2.10 | 2.52 | 2.03 | 2.11 | 2.03 | 2.60 | 2.22 | 2.29 |
| gemma-3-12b-it | 2.42 | 2.96 | 2.85 | 2.35 | 2.46 | 2.81 | 2.45 | 2.19 | 2.33 | 2.78 | 2.59 | 2.38 |
| gemma-3-27b-it | 2.52 | 3.09 | 3.02 | 2.52 | 2.58 | 2.93 | 2.58 | 2.34 | 2.49 | 2.95 | 2.71 | 2.52 |
| InternVL3-2B | 1.81 | 2.23 | 1.79 | 1.96 | 1.89 | 2.21 | 1.50 | 2.22 | 1.77 | 2.14 | 1.68 | 2.00 |
| InternVL3-8B | 2.04 | 2.52 | 2.06 | 2.33 | 2.03 | 2.43 | 1.68 | 2.54 | 1.95 | 2.43 | 1.92 | 2.47 |
| InternVL3-14B | 2.18 | 2.63 | 2.21 | 2.46 | 2.21 | 2.56 | 1.85 | 2.65 | 2.18 | 2.53 | 2.11 | 2.60 |
| InternVL3-38B | 2.29 | 2.78 | 2.30 | 2.65 | 2.33 | 2.69 | 1.92 | 2.81 | 2.23 | 2.71 | 2.18 | 2.77 |
| InternVL3-78B | 2.26 | 2.77 | 2.24 | 2.65 | 2.35 | 2.74 | 1.93 | 2.86 | 2.27 | 2.77 | 2.21 | 2.81 |

*Models are color-coded by type: closed-source (red) and open-source (blue). Scores range from 0–4 for Accuracy (Acc), Relevance (Rel), Completeness (Comp), and Parsimony (Pars). Bold purple marks the highest score for each metric within a column block.*

# G   MIRAGE-MMMT

## G.1   Benchmark Details

The MIRAGE-MMMT dataset as shown in Table 25, contains 861 multi-turn samples, each annotated with a high-level decision label—either Clarify (56.6%) or Respond (43.4%)—reflecting the expert's intent in continuing the consultation. On average, each sample includes 2.11 images and spans 1.52 turns, capturing compact yet information-rich interactions.

Table 25: Summary statistics for the full dataset

| Overall Statistics | Total |
|---|---|
| Total Samples | 861 |
| **Decision Distribution** | |
| Clarify | 487 (56.6%) |
| Respond | 374 (43.4%) |
| **Per-Sample Statistics** | |
| Avg. Images per Sample | 2.11 |
| Avg. Turns per Sample | 1.52 |
| **Word Count Statistics** | |
| Avg. User-turn Words | 109.91 |
| Avg. Expert-turn Words | 80.57 |
| **Distribution Statistics** | |
| Max Images per Sample | 3 |
| Max Turns per Sample | 14 |
| Max User-turn Words | 1 488 |
| Max Expert-turn Words | 287 |

User and expert utterances are relatively verbose, with average lengths of 109.9 and 80.6 words respectively, and a maximum of 1,488 words in a user turn. Each sample includes up to 3 images and 14 turns, reflecting a wide range of complexity and interaction depth. These characteristics make the dataset well-suited for studying decision-making, goal inference, and clarification strategies in visually grounded, expert-guided dialogues.

## G.2   Task Definition

MIRAGE-MT is a multimodal, multi-turn benchmark designed to evaluate conversational expert agents in a consultative decision-making setting. Given a multi-turn dialogue and associated image(s) as shown in Figure 11, the agent must decide whether to ask a clarification question or provide a helpful response, and then generate the corresponding utterance.

### G.2.1   Input

Let a multi-turn dialogue context be represented as a sequence:

$$D = \{(s_1, u_1), (s_2, u_2), \ldots, (s_n, u_n)\}$$

where:

- $s_i \in \{\texttt{user}, \texttt{expert}\}$ denotes the speaker
- $u_i \in \mathcal{U}$ is the corresponding utterance

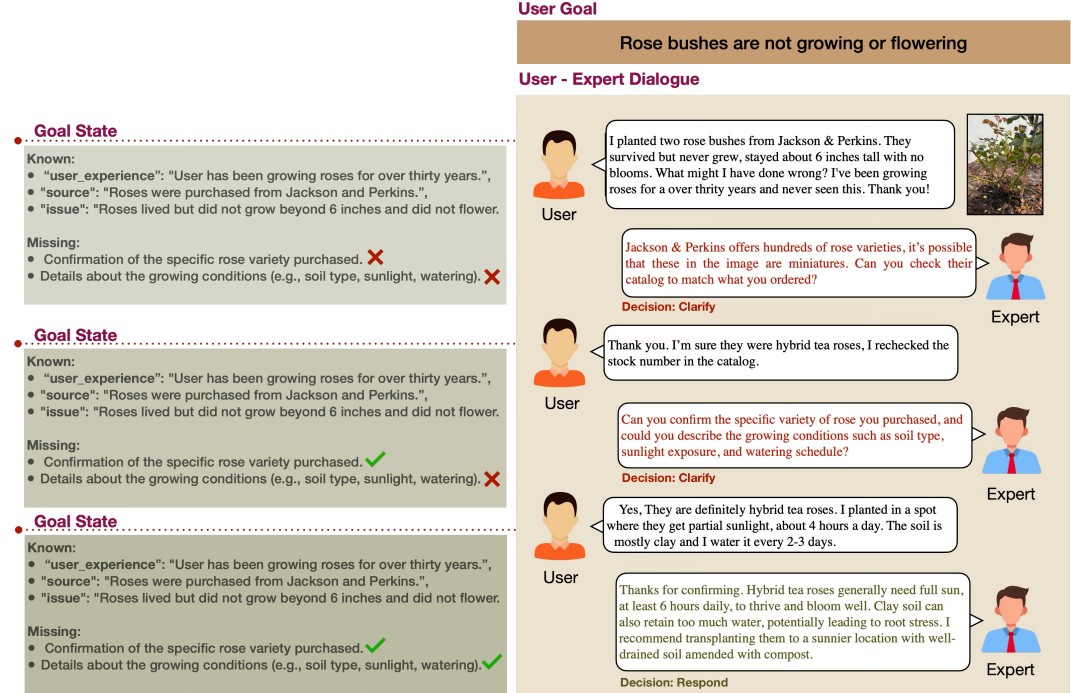

Figure 11: Illustrative example of decision-making in the MIRAGE-MMMT task

Each dialogue is also associated with a set of image inputs:

$$I = \{i_1, i_2, \ldots, i_m\}, \quad I \subset \mathcal{I}$$

which may provide visual context necessary for interpretation (e.g., pest damage, plant structure).

### G.2.2 Output

The model must jointly predict:

1. A **decision** $a \in \mathcal{A} = \{\texttt{<Clarify>}, \texttt{<Respond>}\}$
2. A corresponding **utterance** $r \in \mathcal{U}$, where:
   - If $a = \texttt{<Clarify>}$, then $r$ is a clarification question
   - If $a = \texttt{<Respond>}$, then $r$ is an expert answer

### G.2.3 Goal Inference and Decision Policy

Let $G \in \mathcal{G}$ denote the user's underlying goal (e.g., identifying a plant disease, choosing a planting strategy). The model must infer $G$ and a goal-state representation:

$$S_G = (\texttt{known}, \texttt{missing})$$

The model learns a policy:

$$\pi : (D, I) \to (a, r)$$

and must select the appropriate action:

$$a = \begin{cases} \texttt{<Clarify>}, & \text{if } \exists f \in \texttt{missing} \text{ that is essential to achieve } G \\ \texttt{<Respond>}, & \text{if } \texttt{missing} = \emptyset \text{ or non-essential} \end{cases}$$

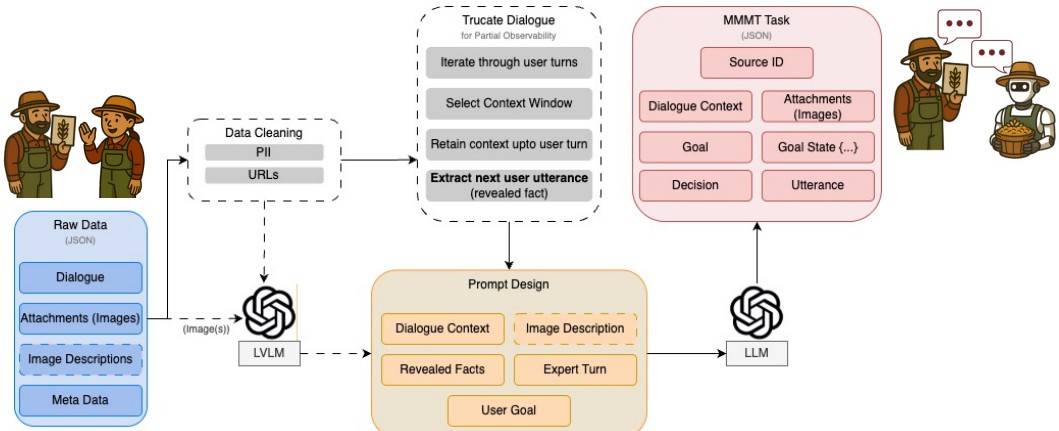

Figure 12: Overview of the MIRAGE-MMMT task generation pipeline. We begin with raw user-expert consultation data including dialogues and attached images. The pipeline applies a series of preprocessing, truncation, and prompting steps to convert each interaction into a structured decision-making task. Green modules denote inputs to the prompt template, while pink boxes indicate components automatically generated using a vision-language model (LVLM). The final structured output includes decision, goal state, and a response or clarification utterance for supervised training or evaluation.

The generation $r$ should then follow:

$$r = \begin{cases} \text{a goal-relevant clarification question,} & \text{if } a = \texttt{<Clarify>} \\ \\ \text{a grounded and helpful expert answer,} & \text{if } a = \texttt{<Respond>} \end{cases}$$

### G.3 Evaluation Criteria

Predictions are evaluated against:

- **Gold revealed fact** $f^*$, obtained from the masked user utterance after expert's turn
- **Gold goal state** $S_G^*$, obtained from the source dialogue
- **LLM-as-a-Judge** ratings:
    - Decision Accuracy
    - Goal Relevance

### G.4 Data Curation Details

Each task sample consists of the dialogue context, referenced images, and metadata such as source dialogue ID and topic. To ensure data safety, we perform automated PII sanitization, replacing all named entities with randomized placeholders while preserving domain relevance. URLs and institutional references are retained when necessary for contextual fidelity. To ensure data quality and task validity, we conduct manual human review on a representative subset of the generated examples. Expert annotators assess the correctness of the decision label, coherence of the generated question or response, and alignment with the revealed user intent. Feedback from this process is used to refine prompt instructions and filter any low-quality generations. Our modular pipeline supports deterministic regeneration of the dataset via fixed seeds and indexing, enabling reproducible experimentation and future extensibility to other domains.

**Release Protocol for MIRAGE-MMMT:** In designing our dataset release, we follow established best practices from recent benchmarks such as MMLU [35], and BIG-Bench [32, 33], which emphasize the importance of separating training data and test targets to prevent leakage and ensure reliable model evaluation. We adopt a protocol that maximizes transparency, reproducibility, and community usability, while preserving the integrity of the held-out test set. We publicly release:

- Full training and validation task datasets, generated from processed source conversations
- Corresponding dialogue context, goal annotations, image references, and model-generated outputs
- Task generation scripts, PII scrubbing utilities, and evaluation tools.

To ensure the credibility and integrity of the test set, we do **not** plan to release the source dialogues or revealed facts used to construct it. Instead, we provide only the test input (dialogue context and image references). This ensures that models are evaluated blind to the gold output, preventing overfitting or prompt leakage. Evaluation of model predictions on the test set can be conducted either via our LLM-based judge or via human assessment.

### G.5 Additional MIRAGE-MMMT Results

Table 26: Classifier performance on the `<Clarify>` vs `<Respond>` decision task using logistic regression with TF-IDF features. Models are grouped by level of input observability.

| Input Variant | Decision Acc. | | F1 (Macro) | | Level |
|---|---|---|---|---|---|
| Dialogue only | 69.79% | | 0.70 | | Realistic |
| Dialogue + Goal | 71.34% | ↑1.55% | 0.71 | ↑0.01 | Semi-Privileged |
| Dialogue + GoalState | **89.27%** | ↑19.48% | **0.89** | ↑0.19 | Oracle |

# H   Prompts

## H.1   Evaluation Prompts for MIRAGE-MMST

Figure 13 presents the inference prompt used for the MIRAGE-MMST Identification Task. `user_query` refers to the original user question. This task evaluates both the model's identification accuracy and reasoning quality, requiring it to generate a clear reasoning chain followed by a final answer. The prompt enforces a standardized output format to facilitate consistent and automatic evaluation. In contrast, for the management task, we impose no format constraints—models are simply given the user question along with the associated images during inference.

---

**EVALUATION Model Inference prompt for MIRAGE-MMST Identification Task**

Please answer the following user question. You should first analyze the provided image, mentioning any visible clues or observations. Then, present the identification result. Write the entire content as one coherent paragraph (analysis + results).

User: {user_query}

---

Figure 13: Model Inference prompt for MIRAGE-MMST Identification Task.

Figure 14 presents the evaluation prompt used for the MIRAGE-MMST Identification Task. Here, `entity_type` denotes the category of the entity—*plant*, *disease*, or *insect/pest*. `user_query` is the original user question, while `expert_answer` contains the expert's full response. The field `entity_name` captures the specific entity mentioned by the expert, with its corresponding scientific name stored in `entity_scientific_name`. The list of `entity_common_names` comprises commonly used names for that entity, collected through external search. Finally, `model_response` refers to the generated answer being evaluated.

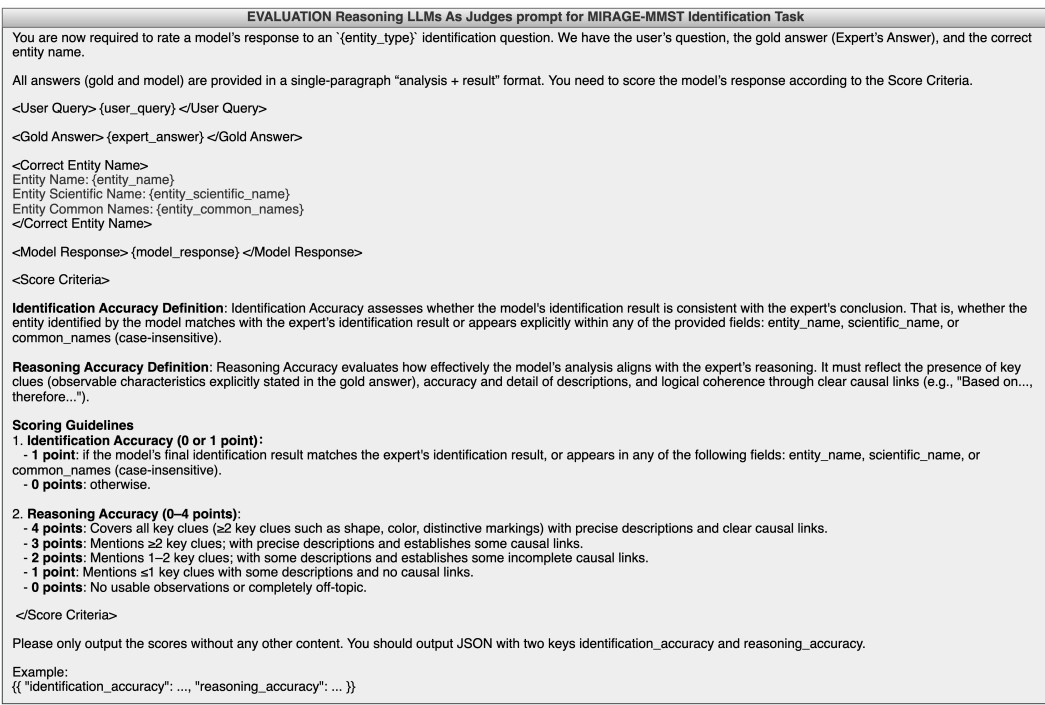

Figure 14: LLM As Judge prompts for MIRAGE-MMST Identification Task.

Figure 15 presents the evaluation prompt used for the MIRAGE-MMST Management Task. Here, `user_query` is the original user question, while `expert_answer` contains the expert's full response. The field `model_response` refers to the generated answer being evaluated.

**EVALUATION Reasoning LLMs As Judges prompt for MIRAGE-MMST Management Task**

You are now required to rate a model's response to an agriculture-related question. We have a gold answer, which is Expert's Answer and based on this gold answer, and the user's question, you need to score the model's answer according to the following four scoring criteria.

<User Query>{user_query}</User Query>

<Gold Answer>{expert_answer}</Gold Answer>

<Model Response>{model_response}</Model Response>

<Score Criteria>
**Accuracy Definition**: Accuracy evaluates whether the agricultural facts, species identification, diagnostic conclusions, and management recommendations provided by the model align with the expert's response. Emphasis is placed on: 1. Correctness of professional terminology (e.g., precise naming of diseases, pests, or invasive species). 2. Accuracy of key details (e.g., descriptions of lesion characteristics, pest behaviors, or plant symptoms). 3. Logical coherence in describing causal relationships (e.g., disease transmission pathways, pest infestation mechanisms). 4. Appropriateness and effectiveness of the proposed management strategies or interventions.

- **4 points**: All agricultural facts, terminologies, diagnostic conclusions, and management recommendations are completely correct, comprehensive, and fully aligned with expert consensus.
- **3 points**: Minor inaccuracies or omissions in terminology, descriptive details, or management advice exist, but the core diagnostic conclusions and recommended management practices remain accurate and effective.
- **2 points**: Noticeable factual errors, misidentifications (species/disease/pests), or suboptimal management suggestions. However, the response still demonstrates partial accuracy or correctness in key aspects.
- **1 point**: Major inaccuracies, such as significant confusion between diseases, pests, or plants, flawed causal logic, or incorrect management practices that could lead to ineffective or detrimental outcomes.
- **0 points**: Entirely incorrect, scientifically invalid, or significantly misleading claims without any alignment with expert consensus.

**Relevance Definition**: This measures how closely the model's response matches the scope and focus of expert answers, ensuring it stays on-topic and avoids tangential information. Responses that digress into unrelated agricultural knowledge or overlook critical points tied to the user's query are considered less relevant.

- **4 points**: The response perfectly mirrors the expert answer and directly addresses the query, using precise terminology and only including question-relevant information.
- **3 points**: The answer is mostly aligned with the expert response and user query, with only minor tangents or slight omissions in details.
- **2 points**: The response contains noticeable deviations or omissions compared to the expert answer, with several off-topic or less relevant points.
- **1 point**: Significant misalignment with the expert answer and the query is evident. The response includes major irrelevant or incorrect content.
- **0 points**: The answer is entirely off-topic, failing to reflect the expert response or address the user query.

**Completeness Definition**: Whether the model's answer covers all key information points mentioned in expert answers to fully address the user's inquiry. If the model omits critical steps or precautions highlighted in expert answers, it is deemed incomplete. Emphasis is placed on: 1. Professional Terminology: Uses precise terms (e.g., names of diseases, pests, invasive species). 2. Key Details: Includes comprehensive descriptions (e.g., lesion characteristics, pest behaviors, plant symptoms). 3. Logical Causal Relationships: Fully explains connections (e.g., disease transmission, pest infestation mechanisms). 4. Management Recommendations: Details all necessary strategies and precautions.

- **4 points**: Covers all key points from the gold answer
- **3 points**: Misses 1-2 minor details but addresses core aspects.
- **2 points**: The response contains noticeable deviations or omissions compared to the expert answer.
- **1 point**: Omits a major component (e.g.,management recommendations).
- **0 points**: Fails to address any key elements of the query.

**Parsimony Definition**: Whether the answer provides actionable guidance that directly addresses the user's core needs, delivering a concise and unambiguous conclusion and specific recommendations without extraneous technical details. The response should adhere to Occam's Razor by avoiding unnecessary complexity and focusing only on what is essential for understanding whether intervention is necessary and what exact steps (if any) need to be taken.

- **4 points**: The answer is succinct, clear, and directly addresses the user's concerns. It offers straightforward, practical guidance that is fully aligned with the visible evidence without any unnecessary details. It embodies the principle of Occam's Razor.
- **3 points**: The answer is generally concise and practical, offering useful advice. However, it may include some extraneous details or slight ambiguity that only minimally detracts from its overall clarity and directness.
- **2 points**: The answer contains relevant information but is overly theoretical or detailed. Extra technical content obscures the key actionable recommendations, making the response less concise and direct.
- **1 point**: The answer is largely indirect or abstract, with a significant amount of unnecessary information. The lack of clarity in actionable guidance leaves the user uncertain about whether any intervention is needed.
- **0 points**: The answer fails to provide practical or actionable recommendations and is cluttered with superfluous details, completely missing the concise, straightforward approach required by Occam's Razor.
</Score Criteria>

Please only output the scores without any other content. You should output JSON with four keys, accuracy, relevance, completeness, parsimony.

Example:
{{ "accuracy": ..., "relevance": ..., "completeness": ..., "parsimony": ... }}

Figure 15: LLM As Judge prompts for MIRAGE-MMST Management Task.

## H.2 Evaluation Prompts for MIRAGE-MMMT

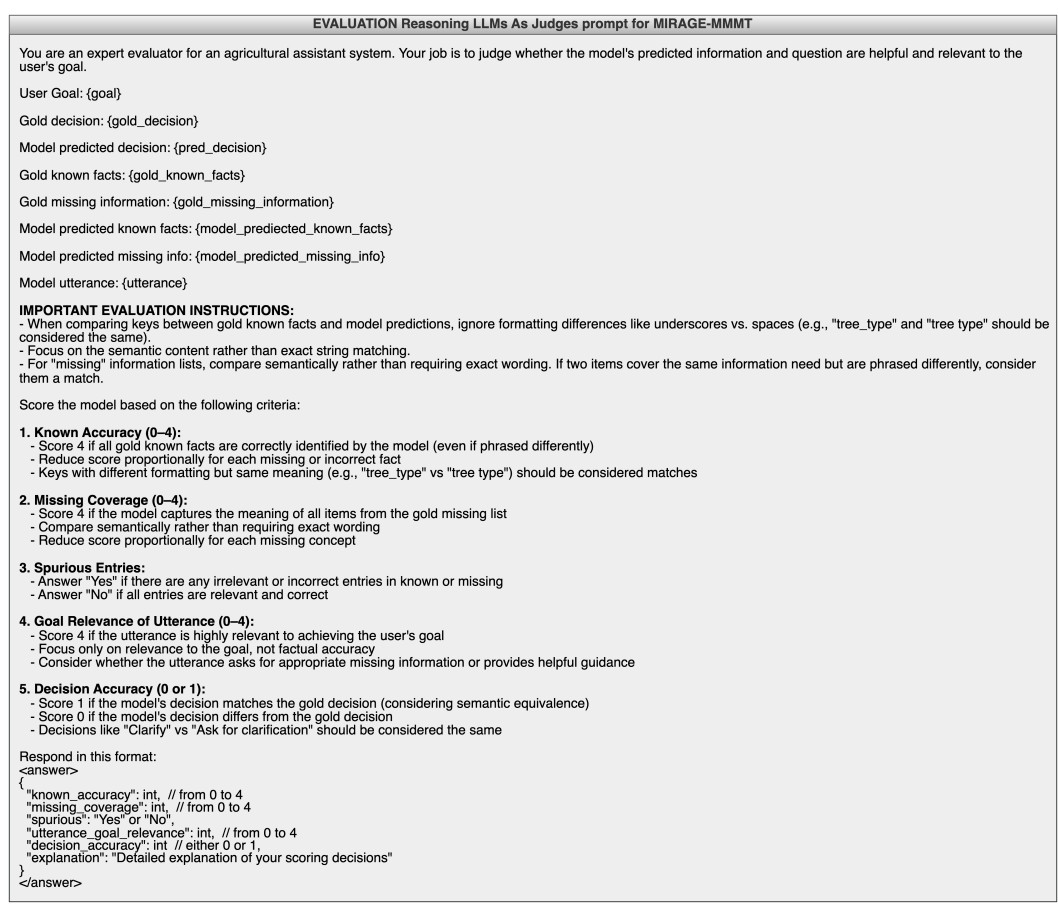

Figure 16: LLM As Judge prompts for MIRAGE-MMMT Prompt.

# I  Case Study

**Contents**

## I.1 Category-Wise Cases

### I.1.1 Plant Identification (MMST Standard)

**Category: Plant Identification** (MMST Standard Benchmark)

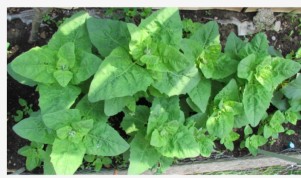 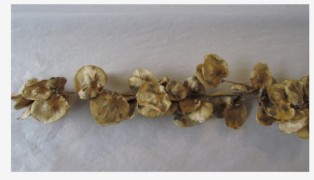 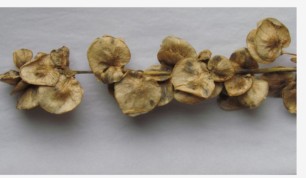

**User:** Can you please tell me the name of this veggie? I got it from my Greek neighbour, who speaks no English. If allowed to grow, it reaches about 3 ft or more. It grows well in Ontario, Canada, and for sure will grow well in Michigan. I think this vegetable is nutritious and can be an additional gardening delight. If it is allowed to seed, it will have a huge number of seeds. Here are pictures when it is still young and a picture of the seeds. I do not want a generic name like \"spinach\" but a specific name to help me find its nutritional details.

**Expert:** The first image shows a healthy cluster of broad, soft, bright green leaves with slightly wrinkled surfaces and a distinct triangular to oval shape, growing closely together from the soil. These leaves resemble those of leafy greens used in cooking rather than typical spinach. The following two images depict dried seed stalks densely packed with large, papery, flattened seed cases in a light to medium brown tone, each almost disc-like, indicative of abundant seed production. These visual characteristics, along with your description of the plant reaching more than 3 feet, thriving in Ontario and similar regions, and producing many seeds, strongly identify this vegetable as Garden Orache (*Atriplex hortensis* L.). Garden Orache is an ancient leafy green prized in Mediterranean and Greek cuisine, belonging to the Amaranth family, notable for its nutritious leaves that can be cooked like spinach.

**Entity Name:** garden orache
**Scientific Name:** `Atriplex hortensis L.`
**Common Names:** `Garden orache`

Figure 17: An example for Plant Identification (MMST Standard).

### I.1.2 Insect and Pest Identification (MMST Standard)

**Category: Insect and Pest Identification** (MMST Standard Benchmark)

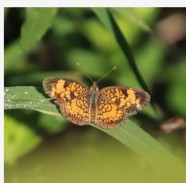

**User:** ID help: Is this a checkerspot?

**Expert:** The butterfly in the image displays a vibrant orange and black pattern with intricate markings covering the wings, including a series of smaller spots and lines that form a complex mosaic pattern. Its forewings and hindwings have a distinct scalloped edge bordered with darker tones. These features match those of the Pearl Crescent (Phyciodes tharos), a common butterfly often mistaken for checkerspots due to some similar color traits. Unlike checkerspots, the Pearl Crescent typically shows finer and more lace-like black markings on an orange background, as seen here. This butterfly is widespread in open areas such as fields and meadows and is known for its association with aster plants as larval hosts. Therefore, based on the pattern and coloration, this butterfly is confirmed as a Pearl Crescent and not a checkerspot.

**Entity Name:** pearl crescent
**Scientific Name:** Phyciodes tharos (Drury, 1773)
**Common Names:** Pearl crescent

Figure 18: An example for Insect and Pest Identification (MMST Standard).

### I.1.3 Plant Disease Identification (MMST Standard)

**Category: Plant Disease Identification** (MMST Standard Benchmark)

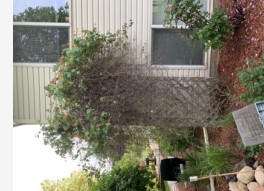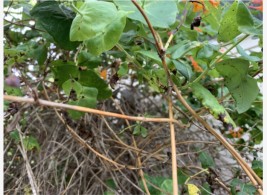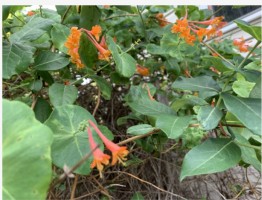

**User:** The top of my bare honeysuckle vine is growing well, however it gets spots and dies below.

**Expert:** The images show a honeysuckle vine with a distinct pattern: the top portion is flourishing with healthy green leaves and bright orange flowers, while the lower portion is suffering significant dieback, with many branches bare or covered in dead twigs and leaves. Closer views reveal numerous small brown to black spots on the older leaves in the lower canopy, which likely represent fungal leaf spots. This type of symptom—healthy growth at the top with decline below—is characteristic of leaf spot infections where the lower parts of the plant, often shaded and wetter, provide ideal conditions for fungal pathogens to thrive, causing leaf tissue necrosis and eventual dieback of stems. Given this, the most probable diagnosis is a leaf spot fungal disease causing lower canopy decline. Management should focus on sanitation by removing and destroying all infected leaves and twigs to reduce fungal inoculum. Improving airflow and sunlight penetration by pruning the vine will help leaves dry faster and inhibit further disease development. Watering should avoid wetting foliage, ideally using drip irrigation or watering at the base early in the day. Mulching while keeping it away from the stems can reduce soil splash-borne pathogens and maintain consistent moisture without stressing the plant. Unless the disease severely impacts the plant year after year, routine fungicide applications are not typically needed, but if desired, preventive fungicides can be applied before symptoms develop, especially during wet weather. These integrated cultural practices will help contain leaf spot spread and promote healthier honeysuckle growth from top to bottom.

**Entity Name:** leaf spot
**Scientific Name:** `Cucumber leaf spot virus`
**Common Names:** `angular leaf spot`

Figure 19: An example for Plant Disease Identification (MMST Standard).

### I.1.4 Plant Disease Management (MMST Standard)

**Category: Plant Disease Management** (MMST Standard Benchmark)

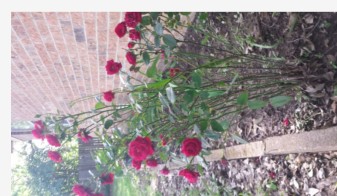

**User:** Is it a fungus causing my miniature rose bushes to get black spots on the leaves and turn yellow, causing them to fall off? They are currently only showing roses with no leaves, and it has just about killed one of them. Please help me, it's not a bug as I have looked and can't see any.

**Expert:** Roses commonly get a fungal leaf spot that is called black spot of rose. Almost all rose cultivars are susceptible. Roses may be sprayed with a fungicide that acts preventatively to keep the foliage clear of the leaf spot. There is no cure for spots that already occur. I suggest that you trim the small rose bush and discard the trimmings when it has finished the first flush of blooms in late May. Also pick up those leaves that fall. This is important in plant disease control. Then spray with a rose or flower and garden spray that lists roses on the label. Spray according to the label and try to get good coverage to all leaves and twigs. You will need to spray again in about 7 to 10 days. Avoid water on the leaves, but try to water the bush around the base of the plant.

**Entity Name:** black spot

Figure 20: An example for Plant Disease Management (MMST Standard).

### I.1.5 Insect and Pest Management (MMST Standard)

**Category: Insect and Pest Management** (MMST Standard Benchmark)

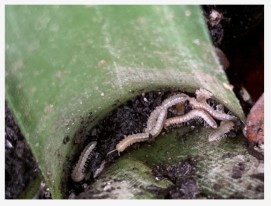 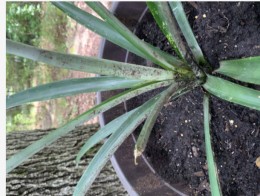

**User:** What are the worms found in my pineapple plant's soil and on the plant? I repotted this 3-year-old indoor pineapple plant in the spring and moved it outside when the weather was warmer. The soil was totally replaced by bagged potting soil. Today I found these worms in the soil and on the plant. Should I treat the ground area where I threw out the wormy soil? If so, with what?

**Expert:** These are millipedes, which are detritivores - they consume dead plant tissue and other organic matter. They are harmless, though some species can secrete odorous and mildly irritating fluid from glands on their bodies if disturbed. They should not harm the pineapple, but their presence might indicate the beginnings of rot on the stem or in its roots. You can keep the pot raised off the deck/patio with 'pot feet,' which hold the bottom of the pot an inch or so up to allow for better airflow and drainage. Millipedes are beneficial in the garden because they help recycle dead plant matter and release nutrients into the soil, so no treatments are needed. Make sure the pineapple is not being kept wet - they want the soil to dry out a fair amount between waterings, and need bright light to thrive. Larger pots will stay wetter for longer after each soaking, so a smaller container might be beneficial to avoid over-watering. If the pot sits on a saucer (which is not necessary outdoors), make sure it gets emptied right away so the drained water does not re-absorb into the soil to drown the roots.

**Entity Name:** millipede

Figure 21: An example for Insect and Pest Management (MMST Standard).

### I.1.6 Plant Care and Gardening Guidance (MMST Standard)

---

**Category: Plant Care and Gardening Guidance** (MMST Standard Benchmark)

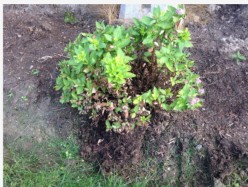 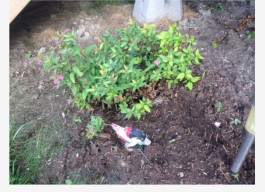 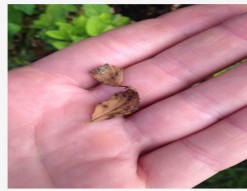

**User:** I got a goldflame spiraea and a gold mound about a month ago. They looked good but are now starting to have leaves turn brown. It had been a drought year but I water them a lot. I see they have grown and keep flowering. It seems the inside of the plant is dying on me. Any advice on why my spiraea is browning?

**Expert:** Spireas grow well in relatively warm and dry conditions. They do not want wet leaves or soggy soil consistently, as this can lead to root rots or fungal leaf diseases. Your spirea may be getting too much moisture from rain and watering. Never water overhead; apply water to the soil at the base of the plant. Most plants grow well with about 1\" of water a week. Supplement rainfall by watering your spirea only when at least the top 1/2 to 1 inch of soil has dried out. Additionally, brush off loose leaves and remove dead leaves.

**Entity Name:** `goldflame spiraea`

---

Figure 22: An example for Plant Care and Gardening Guidance (MMST Standard).

### I.1.7 Weeds/Invasive Plants Management (MMST Standard)

---

**Category: Weeds/Invasive Plants Management** (MMST Standard Benchmark)

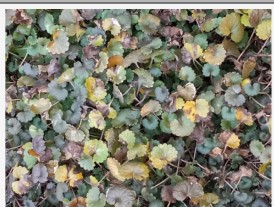 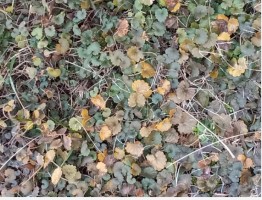

**User:** Can you identify the plant in the attached pictures and suggest the best natural method for eradicating it without using chemicals?

**Expert:** This looks like Ground Ivy (also called Creeping Charlie), and it's a common perennial weed that can be difficult or slow to eradicate without chemicals. The simplest approach is to physically remove it either by hand or with a hard-tined rake (the roots pull up somewhat easily), and then to keep removing any foliage regrowth as soon as it's detected. Eventually, the plants should exhaust their supplies of stored root energy if you can keep it from producing new leaves. If there is little left in the way of lawn or other desirable plants among the Ground Ivy stems, then digging out the entire patch might be faster. Or, you can smother it by denying the Ground Ivy light by covering the patch with dark landscape fabric. Removing what foliage and stems you can and then covering the area with fabric might be a better approach, so the leaf removal weakens the plant first and then the fabric slows its ability to regrow. In either event, pieces of rooted stem left behind can resprout, but as long as they're dealt with promptly, the faster the plant will succumb and stop returning. If growing in lawn, this weed usually prefers sites too shady or damp to support good grass growth, so it might be worth looking into lawn alternatives like a different kind of groundcover to keep the weeds from returning from seed since grass will always struggle to out-compete weeds when growing in challenging conditions. If growing in a garden bed, making sure exposed soil is mulched or is growing a vigorous (but not weedy) groundcover can help prevent future outbreaks.

**Entity Name:** Ground Ivy

---

Figure 23: An example for Weeds/Invasive Plants Management (MMST Standard).

### I.1.8  Plant Disease Management (MMST Contextual)

**Category: Plant Disease Management** (MMST Contextual Benchmark)

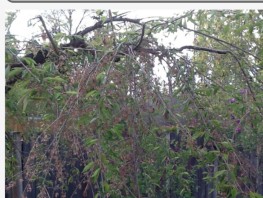 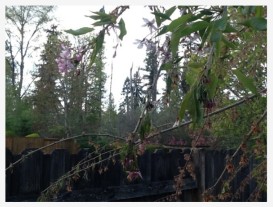 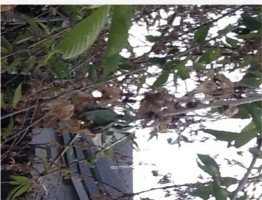

**User:** I spray our "dwarf" flowering cherry each fall and early spring with copper and then sulfur with usually questionable results trying to mitigate shot-hole issues with the first leaf-out. But for some years the tree endures leaf wilt the week after blossom fall/leaf-out, with entire branches exhibiting dying new leaves. It's about the same time as I try to spray draconil for leaf spot, but as usual once again this year the wilt is occurring prior to that. Some branches exhibit resin bursts in places. I'm inclined to think it may be some pest or microbial invader, shutting off nutrient flow to the branch. Wilting starts at the new leaf-outs at the branch terminus and progresses quickly up-stem, but may stop short of the limb base allowing prior leaf nodules to grow normally. I'm tired of having to prune dead branches out each spring and summer, and wonder if there's a systemic or sprayed-on pesticide I should be using for flowering cherry limb/twig wilt. Half the limbs die-back each mid-April, although many revitalize for the second leaf-out, in a few weeks, making it hard to decide right now what to prune out. (asked at 2020-04-16 03:20:40, Clackamas County, Oregon)

**Expert:** Your flowering cherry is experiencing symptoms consistent with both fungal (such as brown rot) and bacterial (specifically bacterial canker caused by Pseudomonas syringae pv. syringae) diseases, both of which are challenging to control, especially in the Willamette Valley's cool, wet climate.\n\nBacterial canker often causes buds not to open, dark brown or greasy spots that become shot holes, blackened shoots, branch cankers, sudden wilting of leaves (especially during hot weather), and resin exudation from canker sites. When infection becomes systemic, new leaves may emerge smaller and discolored, and affected branches or even the whole tree may decline or die. These symptoms match your description of branch tip dieback, leaf wilt soon after blossom fall, and resin bursts on branches.\n\nChemical sprays like copper and sulfur, or even chlorothalonil (Daconil), provide questionable results because copper-resistant bacteria are common, and fungicides do not address bacterial pathogens. Chemical options are limited: fixed copper can be applied before fall rains, during leaf fall, and again late dormant season, but results may not be reliable; there are also products like Arbor-OTC (for trunk injection) and Junction (a copper plus mancozeb product), but even these have variable success and should be used according to label directions.\n\nCultural management is your most effective approach:\n- Prune out cankered or dead branches in dry weather, ideally late dormant season or in summer, to reduce the spread of bacteria and fungi. Disinfect pruning tools between each cut using 10% bleach or 70% rubbing alcohol.\n- Remove and destroy severely infected trees, as older, heavily diseased cherries can serve as inoculum sources.\n- When replanting, choose disease-resistant cultivars if available.\n\nGiven the persistent and severe dieback you are experiencing, consider whether the effort of annual pruning and ongoing chemical treatments is worthwhile. In some cases, replacement with a more disease-resistant tree may be more successful in the long term. If you choose to retain the tree, be vigilant with pruning and sanitation. There are currently no fully effective systemic or sprayed pesticides for controlling limb/twig wilt from bacterial canker or brown rot in flowering cherry.

```
Entity Name: bacterial canker
```

```
Location Related: True
```
**Location Related Analysis:** The expert answer specifies the Willamette Valley's cool, wet climate as a contributing factor to the disease affecting the flowering cherry tree. This is new location-specific context not mentioned by the user, indicating location-dependent advice relevant to the question.

```
Time Related: False
```
**Time Related Analysis:** The expert response discusses diseases and treatment options relevant to the user's description without referencing any timing or seasonal conditions that were not already mentioned by the user. The expert's advice on pruning times (late dormant season or summer) and application periods for copper are general practices and do not rely on knowing the current date or season. The expert does not mention recent or current climatic conditions or specific timing that influences their answer directly. Therefore, the expert's answer does not appear to depend on the question's timestamp information.

Figure 24: An example for Plant Disease Management (MMST Contextual).

### I.1.9 Insect and Pest Management (MMST Contextual)

---

**Category: Insect and Pest Management** (MMST Contextual Benchmark)

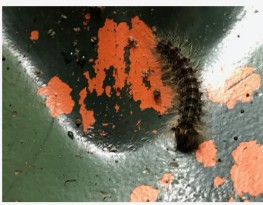 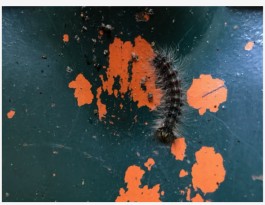

**User:** We have a large amount of black invasive caterpillars on our property this year that may be harming our trees. Can you help me identify them, and let us know if we need to take any action? (asked at 2021-06-18 16:34:58, Muskegon County, Michigan)

**Expert:** The caterpillars in your photos are Spongy moth caterpillars (Lymantria dispar), formerly known as Gypsy moths. These insects can cause noticeable damage to trees, particularly during outbreak years when caterpillar populations are high. The level of harm to your trees depends on several factors, including how many caterpillars are present, the tree species affected, and local weather conditions.\n\nMost deciduous trees can tolerate one or two years of defoliation without lasting impact, as they are generally able to regrow their leaves. However, evergreen species such as spruce and pine are more vulnerable—significant defoliation in a single season can cause lasting damage or even kill these trees. While spongy moth caterpillars typically prefer deciduous trees, during times of population spikes they may feed on evergreens as well.\n\nIf you have only recently noticed the caterpillars or they are not excessively numerous, most deciduous trees should recover. However, if high numbers persist for multiple years or you observe feeding on evergreens, intervention is recommended. Management options include monitoring for egg masses to predict next year's population, applying Bacillus thuringiensis kurstaki (Btk), and using physical barriers like tree bands to limit caterpillar movement.\n\nIt is advisable to regularly inspect your property, particularly at outbreak times, and take action if you have vulnerable trees or repeated infestations. Addressing heavy infestations early can help minimize tree damage and reduce nuisance around your home.

**Entity Name:** `spongy moth caterpillar`

**Location Related: False**
**Location Related Analysis:** The expert's answer provides detailed information about the spongy moth caterpillars, their impact on different tree species, and general management practices. It does not mention any specific or vague locations, local regulations, or location-dependent advice that was not introduced by the user. The user's question already includes the context of 'our property', which refers to their personal space and is not counted as a geographic location per the guidelines. Therefore, the answer is not location-related.

**Time Related: True**
**Time Related Analysis:** The expert references 'outbreak years,' and advises monitoring egg masses to predict next year's population, which implies using current population dynamics and timing to inform management. The advice to inspect property particularly at outbreak times and take action early suggests the expert is using knowledge of the current year's infestation timing to guide recommendations that depend on when the question was asked. This indicates time-related information critical to their guidance.

---

Figure 25: An example for Insect and Pest Management (MMST Contextual).

### I.1.10  Plant Care and Gardening Guidance (MMST Contextual)

**Category: Plant Care and Gardening Guidance** (MMST Contextual Benchmark)

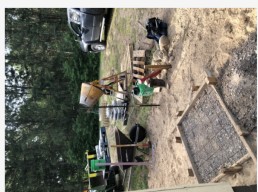

**User:** I have considered the options of planting crops on the property at my location. A farmer in the area recommended that I grow and harvest hemp given the location of the land. I am looking for additional information on hemp harvest. (asked at 2022-08-24 12:01:51, Kent County, Michigan)

**Expert:** At this point, hemp production is still a risky venture given the current markets. Most licenses for hemp growers in Michigan were for those producing hemp for CBD or other cannabinoids, while very few indicated they planned to grow grain or fiber hemp. There are no processing facilities for grain or fiber in the state to my knowledge, so farmers are shipping those to other states. The profit margin for grain and fiber is fairly low - the University of Kentucky has put together enterprise budgets for various types of hemp production. Growing hemp for CBD has a greater likelihood of profit, assuming that you have a buyer in hand, preferably with a contract, prior to planting. It is generally more lucrative to take the biomass (flower and leaves where the CBD is found) further along the value chain by having it processed yourself and selling the crude oil or isolate yourself - again, if you have a buyer for those products in hand. The initial fervor around hemp in 2019 died down considerably after that first year when prices for end-products drastically declined. The license to grow hemp, which you would obtain from the Michigan Department of Agriculture and Rural Development, is $1250 each year, and you would need the processor/handler license in addition at $1350/year if you planned to do any level of processing.

**Entity Name:** hemp

**Location Related: True**
**Location Related Analysis:** The expert answer mentions specific locations such as 'Michigan' and provides location-dependent regulations regarding hemp production and licensing. This introduces significant location-specific information that was not referenced by the user in their question.

**Time Related: True**
**Time Related Analysis:** The expert response addresses the current market conditions and logistical concerns associated with hemp production in Michigan. References to 'At this point,' 'current markets,' and historical context from 'after that first year when prices for end-products drastically declined' suggest that the advice relies on the expert's knowledge of recent developments in the hemp industry, indicating reliance on recent temporal information not provided by the user's inquiry.

Figure 26: An example for Plant Care and Gardening Guidance (MMST Contextual).

### I.1.11 Weeds/Invasive Plants Management (MMST Contextual)

**Category: Weeds/Invasive Plants Management** (MMST Contextual Benchmark)

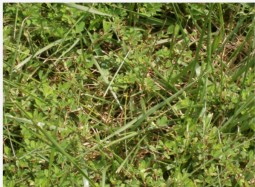

**User:** I reside in the Sykesville/Woodbine area of Carroll. I need to revitalize my lawn. I need to weed, feed, then seed. I am seeking suggestions. There are a variety of weeds in my lawn. There are areas that are laced with rocks - a sort of shale. If possible, I would like to start this process ASAP. Is it advisable to begin weeding now, 8/17/13? When may I begin feeding then seeding? (asked at 2013-08-17 19:39:59, Carroll County, Maryland)

**Expert:** To revitalize your lawn in the Sykesville/Woodbine area, especially with a variety of weeds and rocky sections, start by controlling weeds using appropriate selective broadleaf herbicides. Many effective weed control products require you to wait a few weeks before reseeding; review and precisely follow the label instructions for any chosen herbicide, particularly regarding application timing and intervals before seeding. Based on your lawn's current condition—where much of the area is covered in a clover-like weed—consider whether the desirable turf covers at least 50% of the area. If not, a complete renovation may be more effective than spot treatment. Generally, late summer into early fall is the best time to feed and seed cool-season lawns in Maryland, as soil temperatures remain warm and moisture availability typically improves. Remove as many rocks as possible in the affected areas to ensure good seed-to-soil contact. Begin your weed control now, then proceed with fertilization and seeding as soon as the waiting period after herbicide application concludes. Always ensure thorough watering after seeding and maintain consistent moisture for best seed germination and turf establishment. Proceed stepwise: weed control now, fertilize and seed after the herbicide waiting period has elapsed, and consider full renovation if the turf is mostly weeds.

```
Location Related: True
```
**Location Related Analysis:** The expert answer refers specifically to the 'Sykesville/Woodbine area' and provides timing and treatment recommendations tailored for that location. Although the user mentions the location, the expert also gives timing advice linked to the local climate and conditions in Maryland, which constitutes location-dependent advice not fully explicit in the question. Therefore, this is classified as location-related.

```
Time Related: False
```
**Time Related Analysis:** The expert's response provides general advice about lawn care timing such as 'late summer into early fall is the best time to feed and seed cool-season lawns in Maryland,' which is standard agronomic knowledge rather than specific to the current date. Although the user explicitly mentions the date 8/17/13, the expert does not reference the current date specifically or tailor the advice uniquely based on it. The recommendations about waiting periods after herbicide application and stepwise treatment are generic and apply regardless of the exact time the question was asked.

Figure 27: An example for Weeds/Invasive Plants Management (MMST Contextual).

### I.1.12 Clarify (MMMT)

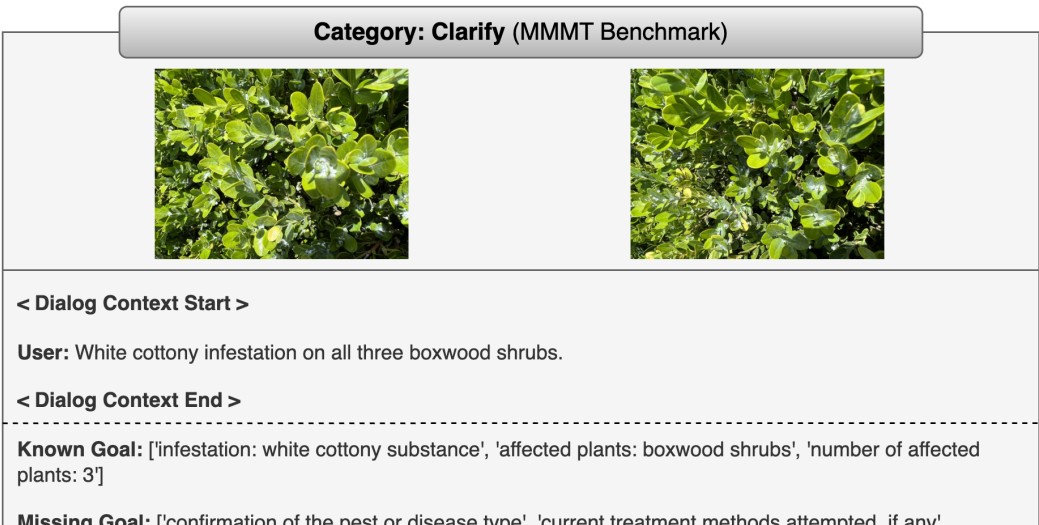

**Category: Clarify** (MMMT Benchmark)

**< Dialog Context Start >**

**User:** White cottony infestation on all three boxwood shrubs.

**< Dialog Context End >**

**Known Goal:** ['infestation: white cottony substance', 'affected plants: boxwood shrubs', 'number of affected plants: 3']

**Missing Goal:** ['confirmation of the pest or disease type', 'current treatment methods attempted, if any', 'geographical location or climate information']

**Decision:** Clarify

**Utterance:** Could you please provide more details about the infestation, such as any specific patterns or behaviors you've noticed? Additionally, have you attempted any treatments so far, and could you confirm your geographical location to better understand the climate conditions?

Figure 28: An example for Clarify Category (MMMT).

### I.1.13 Respond (MMMT)

**Category: Respond** (MMMT Benchmark)

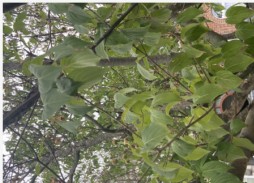 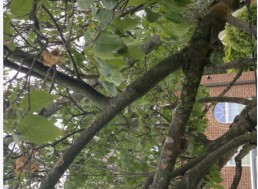 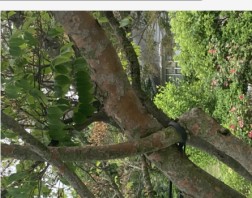

**< Dialog Context Start >**

**User:** I have a 25 year old dogwood whose bark is peeling and branches are dying. There is still green leaves and new growth on the tree. Can it be saved?

**Expert:** Hello, hear about the problem with your dogwood. It's a little difficult to tell from the photos but it looks like Twig and Branch Canker. Symptoms: sunken discolored areas on twigs and small branches. These cankers are caused by fungal pathogens (Botryosphaeria sp. and Diplodia sp.) and are rarely serious but can lead to limb dieback. The fungus Botryosphaeria appears to be more severe on pink flowering cultivars. Healthy plants are less susceptible to these diseases. Water during droughts and avoid over-fertilization to prevent excessive succulent growth. There are no effective chemical controls for this disease once diagnosed, so prune out infected branches promptly. If that doesn't sound right, let us know and we can look further into it. I'll also include the website that this is from, it may be more helpful to see if any of these look more like the problem. Key to Common Problems of Dogwoods | University of Maryland Extension ([umd.edu](http://umd.edu/)) Let us know if we can help you with anything else.

**User:** Thank you for your response. Would it be too much stress if I reshaped the entire tree now? Once I remove the dead branches, should I fertilize? A garden center told me to use holly tone and it's 3 cups for every inch of the trees's diameter. Since there are several branches and not on trunk, they told me to add up all the diameters. They also told me it would be more effective to not put the holly tone directly under the tree trunk but father away so that the roots would be able to absorb it. Is this correct? Thanks again.

**< Dialog Context End >**

- - - - - - - - - - - - - - - - - - - - - - - - - - - - - - - - - - - - - - - - - - - - - - - - -

**Known Goal:** ['tree_age: 25', 'tree_type: dogwood', 'symptoms: peeling bark, dying branches, but still has green leaves and new growth', 'potential issue: Twig and Branch Canker', 'advice sought: pruning and fertilization', 'fertilizer recommendation: holly tone, 3 cups per inch of diameter', 'fertilizer application advice: apply away from the trunk']

**Missing Goal:** [ ]

- - - - - - - - - - - - - - - - - - - - - - - - - - - - - - - - - - - - - - - - - - - - - - - - -

**Decision:** Respond

**Utterance:** Reshaping the entire tree by removing dead branches can be beneficial, but it's important to do it carefully to avoid stressing the tree further. Prune during the dormant season if possible, but if the branches are dead or diseased, it's best to remove them as soon as possible to prevent further spread of any pathogens. As for fertilization, using holly tone is a good choice for dogwoods. The advice to calculate the total diameter by adding up the diameters of all branches and applying the fertilizer away from the trunk is correct. This method encourages the roots to grow outward and absorb nutrients more effectively. Ensure you water the tree adequately, especially during dry periods, to help it recover and thrive.

Figure 29: An example for Respond Category (MMMT).

## I.2 Examples of Reasoning LLM as a Judge

### I.2.1 MMST Identification Task

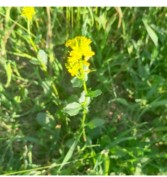
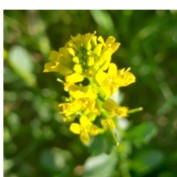
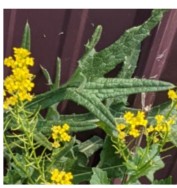

**Instruction:** Please answer the following user question. You should first analyze the provided image, mentioning any visible clues or observations. Then, present the identification result. Write the entire content as one coherent paragraph (analysis + results).

**User:** Is this mustard? Pics taken end of May 2023.

**Expert:** The images reveal a plant with clusters of small, bright yellow flowers arranged in a raceme, typical of the Brassicaceae family. The leaves are notably lobed and somewhat spiny or toothed, distinguishing it from typical mustard species which usually have smoother or less deeply lobed leaves. The overall growth habit is upright with several flowering stalks emerging from a basal rosette. These combined characteristics indicate the plant is not a true mustard but rather Common Winter Cress (Barbarea vulgaris). This species is often mistaken for mustard due to its yellow flowers, yet it is distinct in its leaf morphology and is known to be a weed with toxic properties, especially hazardous to horses.

**Scientific Name:** `Barbarea vulgris (L.) W.T. Aiton`
**Common Names:** `Common Wintercress, Yellow Rocket`

**GPT-4.1:** In analyzing the provided images, I observe plants with clusters of small, bright yellow flowers at the top of slender green stems. The leaves visible in the image are broad with noticeable lobes, and the flowers appear in dense, rounded clusters. These characteristics—yellow, four-petaled flowers, and lobed leaves—are commonly seen in wild mustard (Sinapis arvensis) or a related species in the mustard family (Brassicaceae). The timing of the photos at the end of May aligns with the blooming period for wild mustard in many regions. Based on these visual clues, it is likely that the plants shown in your images are indeed a type of wild mustard, although precise identification down to the exact species would require a closer look at seed pods, leaf structure, and growth habit. Generally, your plant strongly resembles a wild mustard rather than another yellow-flowered weed.

---

Expert ID Label
GPT-4.1 ID Label
Aligned Reasoning
Misaligned Reasoning

**Reasoning LLM as a Judge for MMST Identification Task**

**Judge (DeepSeek-R1-Distill-Llama-70B)**
**Thinking:** Alright, I need to evaluate the model's response based on the given score criteria. Let's start by looking at the Identification Accuracy.

The expert identified the plant as Common Winter Cress (Barbarea vulgaris), and the correct entity name is \"common winter cress.\" The model, however, identified it as wild mustard (Sinapis arvensis), which is a different species. **Since the model's answer doesn't match the expert's identification or any of the common names provided, the Identification Accuracy should be 0.**

Next, for Reasoning Accuracy, I'll assess how well the model's analysis aligns with the expert's reasoning. The model mentioned clusters of small yellow flowers, which the expert also noted, and lobed leaves, which the expert described as spiny or toothed. However, the expert made a distinction based on the leaf morphology, pointing out that mustard usually has smoother leaves. The model didn't mention this key difference and instead focused on similarities with wild mustard. **The model's reasoning is partially correct but misses crucial details that led the expert to a different conclusion. Therefore, the Reasoning Accuracy would be 2 points because it mentions some key clues but lacks the depth and causal links needed for a higher score.**

**Score: {"identification_accuracy": 0, "reasoning_accuracy": 2}**

---

Figure 30: An example illustrating LLM-as-a-Judge framework for MMST Identification Task. The DeepSeek-R1-Distill-Llama-70B judge evaluates both the correctness of the model's prediction and the quality of its reasoning, providing an interpretable feedback.

## I.2.2 MMST Management Task

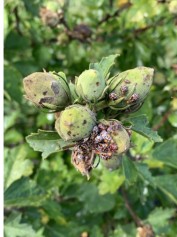

**User:** What are these odd bugs that are eating the rose of Sharon? I have tried soap, seven, water spraying but nothing gets them.

**Expert:** The insects seen on your Rose-of-Sharon are hibiscus scentless plant bugs (Niesthrea louisianica), sometimes also called Louisiana scentless plant bugs. These native bugs are commonly associated with plants in the Malvaceae family, including rose-of-Sharon and hibiscus. Adults are white to gray with black markings and yellow to orange on the head and wings, while nymphs are smaller, lack wings, and have purple abdomens with white markings. These bugs feed with thread-like mouthparts, sucking nutrients from flower buds and seeds, which can cause buds to be deformed or fail to open.\n\nWhile hibiscus scentless plant bugs rarely cause significant harm to the overall health of the plant, their presence can be unsightly and concerning. Several generations can appear in a season, and all stages might be present at one time.\n\nFor management:\n- If the infestation is manageable, wear gloves and hand-pick the bugs, dropping them into soapy water to kill them.\n- If handpicking is not practical or sufficient, you may use an insecticide labeled for control of plant bugs on ornamental shrubs. Insecticidal soap or horticultural oil are effective options.\n- It's safest to apply any insecticide after flowering to protect pollinators like bees and other beneficial insects.\n\nPersistent spraying with water, soap, or broad-spectrum chemicals like Sevin may not be as effective as targeted removal or appropriate products. Always avoid spraying open flowers.\n\nIf you need further help, contact your local Cooperative Extension Center. Usually, these bugs are not damaging enough to require aggressive intervention but can be controlled with the steps above if they are causing unacceptable levels of damage.

**GPT-4.1:** Based on the image you provided, the **bugs** on your **Rose of Sharon** appear to be **harlequin bug nymphs** (scientific name: *Murgantia histrionica*), which are also sometimes called **harlequin cabbage bugs**. \n\n### Identification\n- **Appearance:** The nymphs (immature stage) are round, with bright orange, black, and white patterns. Adults are shield-shaped and also have striking red, orange, and black markings.\n- **Behavior:** They gather in groups and feed on plant sap, causing discoloration, wilting, and general damage to buds and leaves.\n\n### Why Aren't Typical Methods Working?\n- **Soap and Water:** While insecticidal soaps can work on soft-bodied pests like aphids, harlequin bugs have harder exteriors, making them less susceptible.\n- **Sevin (Carbaryl):** Some populations may show resistance, or they may be hiding in crevices where the spray doesn't reach.\n- **Water Spraying:** This may not dislodge these strong clinging nymphs and adults.\n\n---\n\n## **How to Control Harlequin Bugs**\n\n### **1. Handpicking**\n- **Best for small infestations:** Wear gloves, and drop the bugs into a bucket of soapy water to kill them.\n\n### **2. Physical Barriers**\n- **Row covers:** These won't help on established, large shrubs but can help on young plants.\n\n### **3. Botanical Insecticides**\n- **Pyrethrin sprays** can be somewhat effective. Apply **directly** to the pests for the best result, making sure to get under leaves and into flowers.\n- **Neem oil** may also help disrupt their development.\n\n### **4. Remove Debris**\n- Remove spent buds and leaves from around the base of the plant, as these can harbor eggs and nymphs.\n\n### **5. Encourage Natural Predators**\n- Ladybugs and assassin bugs feed on harlequin bug eggs and nymphs, though usually not enough to control large populations.\n\n### **6. Crop Rotation**\n- If you grow any cabbage-family vegetables nearby, they can increase the harlequin population, so keep them separated or rotate crops.\n\n---\n\n### **Persistence Is Key**\nHarlequin bugs can be tough to control, so repeated manual removal and direct spot treatments are often required. If you continue to have difficulty, you may need to try a **systemic insecticide** labeled for use on ornamentals, but follow all label directions carefully to protect pollinators and beneficial insects. \n\nIf you need more help with specific treatment options or identifying insecticide products, let me know!

---

Aligned Reasoning
Misaligned Reasoning

Reasoning LLM as a Judge for MMST Management Task

**Judge (DeepSeek-R1-Distill-Llama-70B)**
**Thinking:** Okay, I need to evaluate the model's response based on the expert's answer using the four criteria: accuracy, relevance, completeness, and parsimony. Let me go through each one step by step.

**First, for accuracy.** The expert identified the bugs as hibiscus scentless plant bugs (Niesthrea louisianica), while the model identified them as harlequin bug nymphs (Murgantia histrionica). This is a clear misidentification, so that's a significant error. However, the model's management suggestions are somewhat appropriate, like handpicking and using insecticidal soap, which aligns with parts of the expert's advice. **But the species identification is wrong, which affects the diagnosis. So, I think accuracy would be low, maybe 1 point because of the major error.**

**Next, relevance.** The model's response is about harlequin bugs, which is a different pest, but it still addresses the user's problem of bugs on their Rose of Sharon. It stays on-topic in trying to help but discusses the wrong species. So, it's somewhat relevant but not fully aligned. **I'd give it 2 points because it's on-topic but misidentifies the pest.**

**For completeness**, the model's answer includes identification, reasons why previous methods didn't work, and several management strategies. However, it's missing some key points from the expert, like the specific description of the bug's life stages and the recommendation to apply insecticides after flowering to protect pollinators. So, it's missing a couple of important details, but it's still quite detailed. **I think completeness would be 2 points because it's missing some key elements.**

**Lastly, parsimony.** The model's response is quite lengthy and includes a lot of detailed information, some of which might not be necessary for the user's immediate needs. **While it's comprehensive, it could be more concise. The expert's answer is more straightforward and to the point. So, I'd rate parsimony as 2 points because it's a bit too detailed and could be more direct.**

**Score: {"accuracy": 1, "relevance": 2, "completeness": 2, "parsimony": 2}**

Figure 31: An example illustrating LLM-as-a-Judge framework for MMST Management Task. The DeepSeek-R1-Distill-Llama-70B judge evaluate the accuracy, relevance, completeness, and parsimony, providing an interpretable feedback.

### I.2.3 MMMT Task

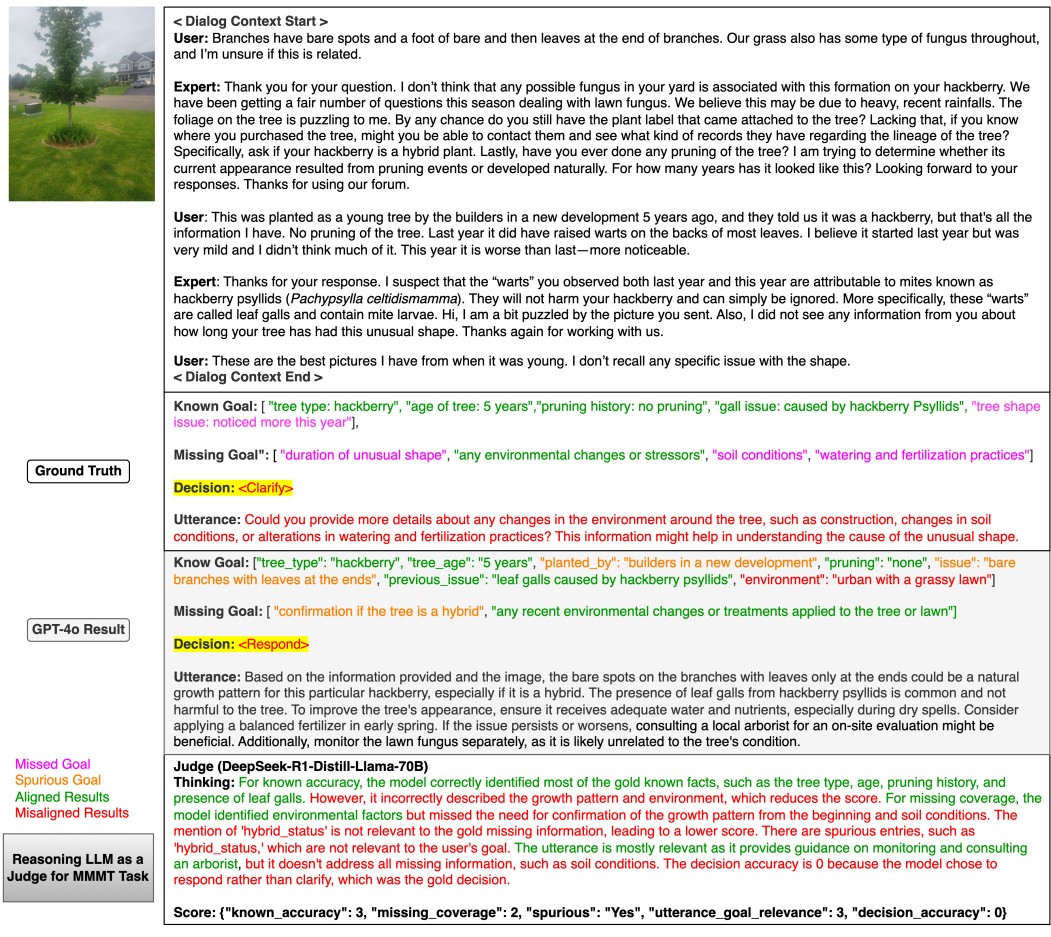

Figure 32: An example illustrating LLM-as-a-Judge framework for MMMT Task. The DeepSeek-R1-Distill-Llama-70B judge evaluate the known_accuracy, missing_coverage, spurious, utterance_goal_relevance, decision_accuracy, providing an interpretable feedback.

