# MIRAGE: A Benchmark for Multimodal Information-Seeking and Reasoning in Agricultural Expert-Guided Conversations

## Supplementary Material

## Table of Contents in Appendix

## A Data Source

The MIRAGE benchmark is constructed from a large-scale archive of real-world agricultural consultations obtained from Ask Extension [22], a U.S. national digital platform maintained by the Extension Foundation. Ask Extension is part of the broader Cooperative Extension System, a federally supported network of land-grant universities that delivers science-based, community-oriented education and services across the United States. The platform connects members of the public, such as farmers, gardeners, or homeowners, with university-affiliated experts who provide timely, research-backed responses to their questions.

Inquiries submitted through the Ask Extension portal are answered by a diverse pool of domain specialists, including university faculty, Extension educators, and trained volunteers such as Master Gardeners. These responses reflect both academic rigor and region-specific expertise, leveraging a unique model of public scholarship that blends localized agricultural knowledge with the latest findings from land-grant institutions. This institutional provenance ensures that the answers used in our dataset are highly reliable, authored by qualified experts, and grounded in scientifically validated practices.

We collected approximately 285,393 interactions (218,431 for single-turn; 66,962 for multi-turn) from the Ask Extension platform, spanning from December 2012 to April 2025. Each entry captures a real question from a user, along with the corresponding expert response, and may include user-uploaded images, time of submission, and geographic metadata.

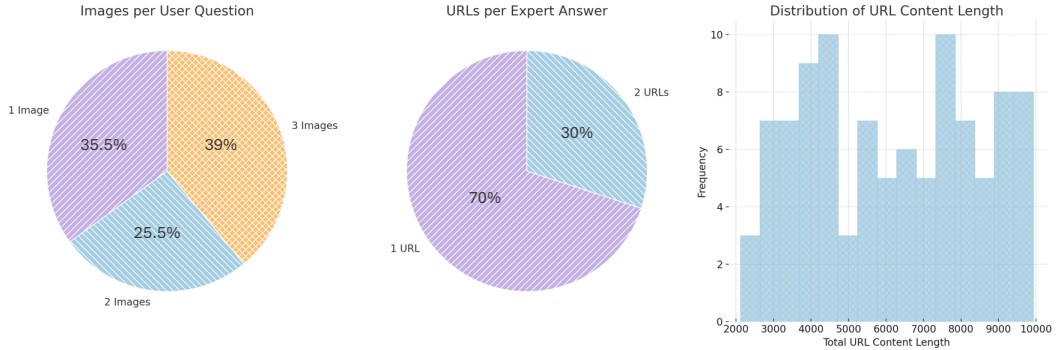

Figure 3: Filtered AskExtension data—(left) number of images per user question, (center) number of URLs per expert answer, and (right) distribution of total URL content length.

As seen in Figure 3, Our filtered AskExtension dialogues are strongly multimodal and reference-rich. Users typically include one to three images in their questions, about 35% of turns have a single image, 26% include two, and 39% include three. Experts in turn ground their advice in external sources: roughly 70% of answers cite a single URL, while the remaining 30% provide two. The total amount of content fetched from those links spans from about 2 000 up to 10 000 tokens per response, indicating that experts draw on substantial external context to support their guidance.

## B Related Works

**Multimodal Large Language Models**: Recent advances in multimodal large language models (LLMs) have markedly expanded vision–language reasoning capabilities. Proprietary models such as *GPT-4* [8], *Claude 3 Sonnet* [10], and Google's *Gemini* [51] demonstrate strong capabilities in unifying visual and textual modalities, achieving notable success across diverse multimodal benchmarks. Concurrently, open-source models—including *Qwen-VL 2.5* [14], *Gemma 3* [52], and *InternVL-3* [68]—have narrowed the performance gap while remaining publicly accessible. Although these models excel on general-domain benchmarks, they underperform in agriculture: they lack fine-grained visual expertise, agronomic terminology, and the ability to reason about rare biological entities and management practices. *MIRAGE* is designed to expose these weaknesses by providing domain-specific, multimodal tasks that require expert-level diagnosis and advice.

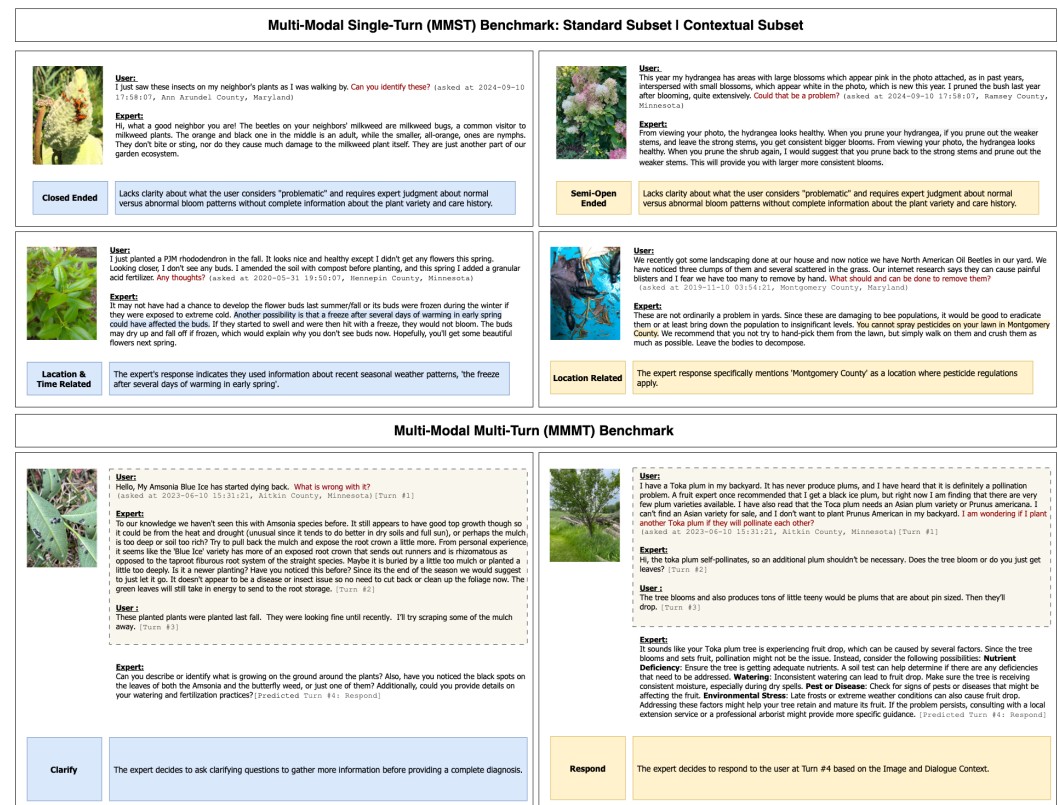

Figure 4: Overview of the tasks in MIRAGE dataset. Dialogues can be single-turn or multi-turn. MMST tasks feature teo subsets: Standard and Contextual