# OpenReview forum: "MIRAGE: A Benchmark for Multimodal Information-Seeking and Reasoning in Agricultural Expert-Guided Conversations"
_NeurIPS.cc/2025/Datasets_and_Benchmarks_Track — NeurIPS 2025 Datasets and Benchmarks Track poster_

### Official Review · Reviewer_636X · 2025-06-16

**Rating:** 5
**Confidence:** 3

**Summary:**

This paper introduces a benchmark to evaluate vision-language models on expert-level reasoning and decision-making in real-world in-domain consultations. It contains two parts: MIRAGE-MMST for single-turn conversation and MIRAGE-MMMT for multi-turn conversations. MIRAGE features underspecified, context-rich scenarios with open-world settings, requiring models to infer latent knowledge gaps, handle rare entities, and either proactively guide the interaction or respond. The paper provides many experiments and shows that there is a large room for recent models to improve.

**Dataset Code Accessibility:**

Yes

**Ethical Considerations:**

No, there are no or only very minor ethics concerns

**Final Justification:**

The rebuttal addresses my concerns and I acknowledge the great value of the benchmark. Thus, I recommend acceptance.

**Limitations Weaknesses:**

1. As the dataset requires professional domain knowledge, I have some doubts about whether it is a suitable dataset for general-purpose ability evaluation.
2. Section 5.3 is poorly organized as the numbering is disordered and the terms MMST-ID and MMST-MG appear for the first time without any explanation.
3. Providing some cases in the main text is necessary (providing them in the Appendix is not suitable), and analyzing some failure cases can offer readers insights into the current model's shortcomings.

**Strengths Contributions:**

1. The dataset is well-organized, each part of the dataset demanding different kinds of ability, posing great challenges to recent models.
2. The dataset derives from real-world questions, reflecting real-world demands.
3. The underspecification characteristic of the dataset distinguishes it from other datasets and reflects the capability asymmetry between users and experts in real-world scenarios, which is an essential ability for model reasoning.
4. The dataset also evaluate the decision-making ability of the model and covers a wide range of entities.
5. The paper provides detailed analysis and experiments on the dataset.

---

> ### Author Rebuttal · Authors · 2025-07-31
>
> **Weakness 1: As the dataset requires professional domain knowledge, I have some doubts about whether it is a suitable dataset for general-purpose ability evaluation.**
>
> We thank the reviewer for raising this important point. We agree that MIRAGE evaluates models in a domain-specific setting, namely, agriculture, and that expert-level benchmarks can differ from general-purpose evaluations in scope and difficulty. However, we believe this is a strength, not a limitation, and we outline our rationale below. While many existing benchmarks assess general-purpose capabilities in relatively clean and well-specified settings (e.g., captioning, short-form VQA, MCQs), MIRAGE is designed to evaluate whether models can perform robust, contextualized, and multimodal reasoning in real-world expert consultation scenarios. Such settings demand not only perception and language fluency, but also the ability to understand under-specified queries, reason over domain-grounded context (e.g., crop health, pest management), and make careful, actionable decisions, capabilities essential for deploying LVLMs in critical, knowledge-intensive domains such as healthcare, legal advising, or technical support. We see MIRAGE as a complementary testbed that highlights important limitations in current frontier models, limitations that remain invisible under standard general-purpose evaluation. We appreciate the reviewer’s perspective and will clarify in our revision that MIRAGE is intended as a stress test for models aspiring toward deployment in grounded, decision-critical tasks and how it can complement general-purpose benchmarks in characterizing a model’s strengths and limitations.
>
> **Weakness 2: Section 5.3 is poorly organized as the numbering is disordered and the terms MMST-ID and MMST-MG appear for the first time without any explanation.**
>
> We sincerely thank the reviewer for pointing this out. We agree that Section 5.3 would benefit from improved structure and clearer terminology. MMST-ID and MMST-MG are the components of our single-turn benchmark: MMST-ID refers to the Single-turn Identification task, and MMST-MG refers to the Single-turn Management Task. We will fix the inline numbering, and in our revision, we will clearly define these terms prior to their appearance.
>
> **Weakness 3: Providing some cases in the main text is necessary (providing them in the Appendix is not suitable), and analyzing some failure cases can offer readers insights into the current model's shortcomings**
>
> We thank the reviewer for this important suggestion. We understand the value of including representative task examples directly in the main text rather than relegating them to the appendix. In our submission, we included detailed task examples for both MMST and MMMT (In the Appendix of Supplementary Material, Figure 4) and qualitative visualizations of failure cases in Appendix Figures 28 and 29, but we agree that presenting a few illustrative examples in the main paper will greatly improve the clarity of the paper. In our revision, we will add some cases for each in the main paper.
>
> Separately, as mentioned in our response to another reviewer (7EZx), we thank the reviewer 636X for suggesting expanding on our error case analysis. In the Supplementary Material (Figures 28 and 29), we already present some qualitative examples of GPT-4.1 failure cases across MMST-ID and MMST-MG tasks. We build on top of this a systematic error analysis, which differentiates performance gaps between frontier models (GPT-4.1) and the Open-Sourced model (Qwen2.5-VL-32B). We systematically manually analyzed failure cases using stratified sampling across three distinct scenarios: GPT-4.1 failures (n=100 from 2,395 cases), cases where Qwen2.5-VL-32B failed but GPT-4.1 succeeded (n=100 from 1,078 cases), and joint failures (n=100 from 300 cases). Our comparison revealed fundamental differences in how these models approach expert-domain reasoning and consultation challenges. We categorize these challenges into 2 tiers: **Fundamental Domain Challenges** and **Open-Source Model Systematic Failures**.
>
> **Quantitative Error Pattern Analysis**
> | **Failure Category** | **Percentage** | **Model** |
> |---------------------|----------------|----------------|
> | **Tier 1: Fundamental Domain Challenges (GPT-4.1 Failures)** |
> | Specialist Knowledge Gaps | 35% | GPT-4.1 |
> | Complex Diagnostic Reasoning | 10% | GPT-4.1 |
> | **Tier 2: Open-Source Model Systematic Failures (Qwen2.5-VL-32B vs GPT-4.1)** |
> | Vision-Language Integration Issues | 40% | Qwen2.5-VL-32B |
> | Diagnostic Hedging Bias | 25% | Qwen2.5-VL-32B |
> | Generalist Reasoning Bias | 20% | Qwen2.5-VL-32B |
>
> **Critical Finding:** In joint failures, GPT-4.1 maintains superior reasoning quality in 60% of cases, demonstrating a deeper understanding and integration of visible clues, more accurately mimicking expert reasoning patterns even when incorrect in identifying the exact entity.
>
> Please see our response to **Reviewer 7EZx – Weakness 3** for further details.

---

> ### Comment · Reviewer_636X · 2025-08-04
>
> Thanks for the rebuttal. It is a valuable dataset.

---

### Official Review · Reviewer_QXVi · 2025-07-01

**Rating:** 5
**Confidence:** 5

**Summary:**

This paper introduce MIRAGE, a high-fidelity benchmark for evaluating vision-language models (VLMs) in  expert-level agricultural consultations. MIRAGE is publicly releasedtosupport the development of vision-language systems that go beyond narrow question  answering, toward models capable of engaging in natural interactions that involve ambiguity.

**Dataset Code Accessibility:**

Yes

**Ethical Considerations:**

No, there are no or only very minor ethics concerns

**Final Justification:**

I am happy to keep my current rating as 5.

**Limitations Weaknesses:**

1.In line 238, what is the rationale behind using a 2:1:1:1 ratio for aggregating the scores? Please clarify the motivation or justification for this specific weighting scheme.

2.In line 265, what does “base-instruct” refer to? Please provide a clear definition or explanation for this term, especially if it represents a specific model variant or training configuration.

3.The description of the evaluation datasets is not sufficiently comprehensive. For example, the benchmark B-AVIBench [ref1],et al., which is designed to evaluate model robustness, should be included as a relevant dataset for more complete coverage.

[ref1]B-AVIBench: Towards Evaluating the Robustness of Large Vision-Language Model on Black-box Adversarial Visual-Instructions. IEEE Transactions on Information Forensics and Security.
[ref2]Convbench: A multi-turn conversation evaluation benchmark with hierarchical ablation capability for large vision-language models. Advances in Neural Information Processing Systems, 37, 100734-100782.

**Strengths Contributions:**

This paper introduce MIRAGE, a high-fidelity benchmark for evaluating vision-language models (VLMs) in  expert-level agricultural consultations. MIRAGE is publicly releasedtosupport the development of vision-language systems that go beyond narrow question  answering, toward models capable of engaging in natural interactions that involve ambiguity.

---

> ### Author Rebuttal · Authors · 2025-07-31
>
> **Weakness 1: Rationale behind using a 2:1:1:1 ratio for aggregating the scores**
>
> We thank the reviewer for this thoughtful question. We agree that the choice of aggregation weights is critical in multi-dimensional evaluation, and we are happy to clarify our rationale for using the 2:1:1:1 weighting scheme when computing the W-Sum score.
> This weighting reflects expert-informed prioritization among the four evaluation dimensions: Accuracy, Relevance, Completeness, and Parsimony. Through discussions with agricultural extension experts, we consistently observed that Accuracy, the factual correctness of the response, was regarded as the most crucial dimension. Experts consistently prioritized the correctness of the diagnosis above all else, noting that factual errors can mislead users and result in inappropriate or even harmful interventions. To reflect this priority while maintaining balance across all aspects of answer quality, we assign double weight to Accuracy relative to the other three dimensions, resulting in a 2:1:1:1 ratio. This ensures that the final composite score places appropriate emphasis on factual correctness, while still accounting for other dimensions of the consultation response.
>
>
> **Weakness 2: What does “base-instruct” refer to?**
>
> ​​We thank the reviewer for pointing out this ambiguity. In our manuscript, the term “base-instruct” refers to the standard zero-shot prompting of instruction-tuned models using their native chat or completion interface, without any additional scaffolding. This setup is intended to represent a baseline inference configuration i.e., how these models are typically used out-of-the-box for user-facing tasks. We appreciate the reviewer for highlighting this. We will revise the manuscript to define this term more clearly upon its first use to avoid confusion. Thank you again for helping us improve clarity.
>
>
> **Weakness 3 & 4: Description of the evaluation datasets is not sufficiently comprehensive**
>
> We thank the reviewer for these excellent suggestions. We appreciate the pointers to B-AVIBench and ConvBench, both of which represent valuable recent contributions to the growing landscape of vision-language model (VLM) evaluation.
>
> While both benchmarks share our broad interest in evaluating the capabilities of large vision-language models, their design motivations and evaluation settings differ from MIRAGE.
>
> **B-AVIBench** focuses on evaluating robustness under black-box adversarial perturbations, targeting the stability and reliability of VLMs in the face of image and instruction-level attacks. In contrast, MIRAGE is not designed as an adversarial benchmark. It instead evaluates expert-level reasoning and decision-making in consultative, knowledge-intensive, and open-world domains like agriculture, where the challenge arises not from adversarial noise but from underspecification, partial observability, and domain grounding.
>
> **ConvBench**, meanwhile, is designed to assess multi-turn conversation quality, with a focus on evaluating dialog consistency and instruction-following across hierarchical ablations. While MIRAGE-MMMT also evaluates multi-turn reasoning and conversational decision-making, our task formulation is explicitly decision-driven, requiring the model to choose between clarifying or responding, and is grounded in domain-specific user goals, visual inputs, and real-world metadata. This formulation more closely mirrors how experts operate in real consultations, making MIRAGE well-suited for studying goal-driven multimodal interaction in applied settings.
>
> That said, both B-AVIBench and ConvBench are highly relevant in the broader context of robust and interactive VLM evaluation. We agree that they complement the evaluation landscape and help motivate the need for more comprehensive, diverse, and stress-tested evaluation frameworks.
>
> We will revise our manuscript to include these two benchmarks and clarify how MIRAGE complements their contributions.

---

> > ### Comment · Reviewer_QXVi · 2025-08-04
> >
> > I'm happy to maintain my accept rating.

---

### Official Review · Reviewer_SyGM · 2025-07-02

**Rating:** 5
**Confidence:** 4

**Summary:**

This paper proposes a novel benchmark dataset, MIRAGE, for multimodal expert-level reasoning and decision-making in single-turn and consultative interaction settings. The dataset is constructed based on 35,000 real user-expert interactions from AskExtension. 22 LLMs are evaluated on the MIRAGE. 3 settings are tested: zero-shot, chain-of-thought, fine-tuned (Qwen2.5-VL-3B).

**Dataset Code Accessibility:**

Yes

**Ethical Considerations:**

No, there are no or only very minor ethics concerns

**Final Justification:**

Thanks for your explanation. I am happy to keep my current rating as 5.

**Limitations Weaknesses:**

1. In Table 2, can you explain the intuition of the W-Sum score? Why use these weights? What intuition do you use to pick these 4 dimensions for the MG evaluation?
2. In Table 10, how do you incorporate spatiotemporal metadata into the current setting?
3. While LLM-as-a-Judge is widely used in LLM evaluation, I wonder whether any human evaluation have been done to manually check the performance of these LLMs.

**Strengths Contributions:**

1. Comprehensive evaluations have been conducted on the presented dataset with 22 VLMs. Many conclusions are drawn from the experimental results which are invaluable for the users.
2. Figure 1 is very interesting. It shows that while fine-tuning LLMs can improve their performance on the given datasets but also potential harm their generalizability.
3. The dataset MIRAGE-MMMT is useful to test the effectiveness of VLMs in a multi-step conversational setting.

---

> ### Author Rebuttal · Authors · 2025-07-31
>
> We thank the reviewer for raising this important question regarding the intuition behind our choice of evaluation dimensions and the use of a weighted-sum (W-Sum) score in Table 2.
>
> **Weakness 1: In Table 2, can you explain the intuition of the W-Sum score? Why use these weights? What intuition do you use to pick these 4 dimensions for the MG evaluation?**
>
> **Dimensions for Management (MG) Evaluation**: We chose **Accuracy, Relevance, Completeness, and Parsimony** as the four core dimensions for Management (MG) evaluations based on both insights from prior research and our discussions with domain experts. First, **Accuracy, Relevance, and Completeness** are grounded in findings from the long-form QA literature. Xu et al. (2023) emphasize that widely used metrics like ROUGE, BERTScore, BLEURT, and GPT-2 PPL show weak correlation with human preferences, recommending a multi-faceted evaluation that explicitly targets factuality and completeness [1]. Côrtes (2024) further shows that GPT-4-based evaluation prompts and regression models focusing on completeness and relevance correlate most strongly with human ratings [2]. These studies informed our adoption of these three axes as essential for capturing answer quality beyond surface-level overlap. The fourth dimension, **Parsimony**, is rooted in the principle of simplicity (Occam’s razor), which has deep roots in both philosophy and cognitive science, where it’s shown that people prefer explanations that minimize complexity while retaining informativeness (Feldman, 2016) [3]. In controlled experiments, Pacer & Lombrozo (2017) demonstrate that, given multiple causal accounts of equal fit, human judges systematically endorse the simpler alternative [4]. In our domain, overly verbose or speculative answers risk undermining user confidence and generating unnecessary anxiety, so we explicitly reward parsimonious, evidence-based recommendations. This pattern was also empirically observed in our evaluations, as illustrated in Figure 9 in the Appendix of the Supplementary Material, where we see most models tend to hedge their bets by offering long, multiple possible explanations. While such answers may appear comprehensive, they often confuse users in consultation settings, where actionable guidance is paramount. A real-world user typically wants to know what to do, not navigate a list of speculative possibilities. Our parsimony criterion helps penalize such hedging and rewards clarity and decisiveness in expert-like responses.
>
> **W-Sum score**: To enable model comparison with a single scalar, we adopt a weighted-sum (W-Sum) aggregation, a standard approach in multi-criteria evaluation. While we recognize the value of preserving a multi-dimensional metric, a combined score is necessary for ranking and model selection, especially when reporting aggregate performance. The specific weights (2:1:1:1) are grounded in qualitative discussions with agricultural experts, where Accuracy was repeatedly emphasized as the most critical factor. Experts consistently prioritized the correctness of the diagnosis above all else, noting that factual errors could mislead users and result in inappropriate or even harmful interventions.
>
> [1] Xu, F., Song, Y., Iyyer, M., & Choi, E. (2023). A critical evaluation of evaluations for long-form question answering. arXiv preprint arXiv:2305.18201.
>
> [2] Côrtes, E. G. (2024). Beyond accuracy: completeness and relevance metrics for evaluating long answers.
>
> [3] Feldman, Jacob. "The simplicity principle in perception and cognition." Wiley Interdisciplinary Reviews: Cognitive Science 7, no. 5 (2016): 330-340.
>
> [4] Pacer, Michael, and Tania Lombrozo. "Ockham’s razor cuts to the root: simplicity in causal explanation." Journal of Experimental Psychology: General 146, no. 12 (2017): 1761.
>
> We appreciate the reviewer’s request for clarification and will make this rationale more explicit in our revised manuscript, including a clearer explanation of the expert-informed motivations for our dimension selection and weighting strategy.
>
> **Weakness 2: Incorporation of Spatiotemporal Metadata (Table 10)**
>
> We appreciate the opportunity to clarify how spatiotemporal metadata is incorporated into our model evaluations. For each user query, we extracted geographic location and time of year based on the metadata associated with the original user post. This data was explicitly provided to the models at inference time as natural language context, e.g., “This query originates from Hennepin County, Minnesota, on 2020-05-31 at 19:50:07.” We included this information in the input prompt alongside the image and question.
>
> This setup simulates a real-world expert consultation scenario, where such context would be made available to a human expert. Our goal was to evaluate whether current vision-language models (VLMs) can effectively utilize this additional context to enhance their responses. Tables 9 and 10 of the Supplementary Material show that in the Identification (ID) task, all models’ identification accuracy varied by at most ±1.60 percentage points and their reasoning scores by at most ±0.05. In the Management (MG) task, variations for every metric remain below ±0.04. These consistently modest changes highlight a critical limitation of current VLMs—their underdeveloped ability to utilize structured contextual information in expert-level settings effectively.
>
> We will make this implementation detail clearer in our revision to avoid ambiguity around how time and location were used during inference.
>
> **Weakness 3: While LLM-as-a-Judge is widely used in LLM evaluation, I wonder whether any human evaluation has been done to manually check the performance of these LLMs.**
>
> We acknowledge that human expert evaluation is invaluable, particularly in domains like agriculture. In designing our benchmark, we carefully considered this and structured our evaluation framework to maximize rigor and interpretability at scale. Given the size and breadth of our benchmark with an ensemble of 3 different judge models, multiple tasks, and over 7000 unique entities, we opted for a scalable and reproducible approach grounded in expert-authored reference answers, while recognizing that targeted human studies remain an important complementary direction for future work. Our judge ensemble is firmly grounded in expert-authored reference answers, ensuring alignment with domain-specific expectations. To enhance reliability and avoid the pitfalls of single-pass evaluations, we had each judge rate every model output three times, capturing intra-judge variability. We assessed inter-judge agreement across all scores using Fleiss’ κ (0.75–0.88) and Kendall’s W (0.69–0.87), indicating good to excellent consensus on both classification and ranking tasks. Intra-judge consistency for the repeated ratings was measured via ICC(2), which confirmed high stability for DeepSeek-R1-Distill-Llama-70B and Qwen3-32B (see Appendix D in Supplementary Material, Figs. 6–7). In future work, we will complement this automated framework with human expert studies to further validate and refine our evaluation methodology.

---

> > ### Comment · Reviewer_SyGM · 2025-08-07
> >
> > Thanks for your explanation. I am happy to keep my current rating as 5.

---

### Official Review · Reviewer_7EZx · 2025-07-02

**Rating:** 4
**Confidence:** 3

**Summary:**

This paper presents MIRAGE, a large-scale, multimodal benchmark designed to evaluate vision-language models (VLMs) in realistic agricultural expert consultation settings. The benchmark is divided into two main tasks: MMST (Multimodal Single-Turn) and MMMT (Multimodal Multi-Turn), testing models on identification, causal reasoning, management recommendations, and clarify-or-respond decision-making. Extensive experiments on 22 vision-language models demonstrate that even state-of-the-art systems—especially open-source models—struggle with MIRAGE’s real-world complexity.

**Dataset Code Accessibility:**

Yes

**Ethical Considerations:**

No, there are no or only very minor ethics concerns

**Final Justification:**

They address my concern on task setting. I will maintain my accept score.

**Limitations Weaknesses:**

- MIRAGE does not simulate fully interactive dialogues or user feedback loops. Multi-turn tasks are limited to text-only user turns after the initial image, and do not model visual follow-ups.
- From the experiment, the results show that including location and time metadata has surprisingly little impact, suggesting either models underutilize such information or that it is not sufficiently integrated in the current benchmark/design.
- While challenges are documented, the work could provide deeper qualitative analysis of error types and the sources of failure, particularly for open-source models, to further guide future improvements.
- No more questions. A valuable benchmark.

**Strengths Contributions:**

- MIRAGE is constructed from a large set of authentic user-expert conversations, moving beyond synthetic or closed-world benchmarks.
- The dataset incorporates user images, metadata (location, time), and naturalistic language, as well as detailed, expert-authored responses. It covers a taxonomically diverse set of real-world problems.
- MIRAGE includes unseen and long-tail entities, ambiguous queries, and requires open-ended decision-making and proactive clarification.

---

> ### Author Rebuttal · Authors · 2025-07-31
>
> We thank the reviewer for their valuable feedback and appreciate their recognition of MIRAGE as a “valuable benchmark” that "poses great challenges to recent models."
>
> **Weakness 1: MIRAGE does not simulate fully interactive dialogues or user feedback loops. Multi-turn tasks are limited to text-only user turns after the initial image, and do not model visual follow-ups.**
>
> We concur that extending MIRAGE to simulate fully interactive dialogues, including visual follow-up interactions, is an essential future direction. Currently, MIRAGE only isolates the initial image-based interaction to target a core decision-making challenge (Whether to respond or to clarify) that an expert must make during a consultation. We acknowledge this constraint in section 6, lines 315-318. Additionally, simulating interactive dialogue would require a domain-aware user simulator capable of engaging effectively with the expert model, which was beyond the scope of the current work. In our revision, we will further emphasize our plans for future enhancements, specifically:
> - Incorporating interactive visual follow-ups.
> - Developing dynamic user–agent feedback loops.
> - Integrating a domain-specific user simulator.
>
> **Weakness 2: From the experiment, the results show that including location and time metadata has surprisingly little impact, suggesting either models underutilize such information or that it is not sufficiently integrated in the current benchmark/design.**
>
> Thank you for highlighting this observation. We extracted the geographic location from where the user posted the request, as well as the time of the request’s origin. Through in-depth discussions with agricultural extension experts, we found that location and time metadata are critical signals they rely on while consulting. They routinely draw on seasonal trends, region-specific pest cycles, and historical climate patterns to contextualize and tailor actionable advice. This form of experiential, situational recall is deeply embedded in their domain expertise. In contrast, current vision-language models lack mechanisms to reason over such temporally and geographically grounded patterns, and instead treat metadata as static, isolated inputs. Our controlled ablation experiments (Tables 9 and 10 in the Appendix of the Supplementary Material) were specifically designed to assess how effectively current VLMs utilize contextual metadata. We provided the time and location of origin of the user’s request to supply additional contextual information to the model. In the Identification (ID) Task, all models showed changes within ±1.60 in identification accuracy and within ±0.05 in reasoning score. In the Management (MG) Task, performance differences across all metrics for all models remained within ±0.04. These consistently modest changes highlight a critical limitation of current VLMs—their underdeveloped ability to utilize structured contextual information in expert-level settings effectively.
>
> **Weakness 3: While challenges are documented, the work could provide a deeper qualitative analysis of error types and the sources of failure, particularly for open-source models, to further guide future improvements.**
>
> We appreciate the reviewer’s call for deeper qualitative insights. In our Supplementary Material submission, we added two illustrative case figures (Figs. 28 and 29) showcasing representative GPT-4.1 failures on both MMST tasks (MMST-ID and MMST-MG). To further address this feedback, we conducted a stratified manual error analysis across three scenarios: (i) GPT-4.1 failures (n = 100 of 2,395), (ii) Qwen2.5-VL-32B failures where GPT-4.1 succeeds (n = 100 of 1,078), and (iii) joint failures (n = 100 of 300). Focusing on instances with zero identification accuracy, we compared failure modes and root causes between the proprietary GPT-4.1 and the open-source Qwen2.5-VL-32B. The errors cluster into two tiers: Fundamental Domain Challenges and Open-Source Model Systematic Failures.
>
> **Tier1 Fundamental Domain Challenges**: These represent the current boundaries of what even the most advanced LVLMs can achieve in agricultural reasoning and consultation settings. These are failures that occur in GPT-4.1, the best-performing model in our evaluation, indicating inherent limitations in current LVLMs for our task settings.
>
> - **Specialist Knowledge Gaps (20% cases)** involve missing domain-specific understanding of biological phenomena that require deep expertise beyond general training. The LVLMs fail in knowing what the phenomenon represents or how biological systems work. *Example: GPT 4.1 could not understand what “appleleaf blister mite” damage looks like or how it develops.*
>
> - **Complex Diagnostic Reasoning (10% cases)** This is mainly correct symptom observation but wrong disease/condition conclusion. The model’s failure in causal reasoning about biological processes, despite accurate visual analysis is the cause for such errors. *Example: Seeing insect damage but misdiagnosing the type of pest or disease involved*
>
>
> **Tier 2 Open-Source Model Systematic Failures**: These represent additional failure modes that we identified occurring in Qwen2.5-VL-32B but are absent in GPT-4.1. These are cases where GPT-4.1 succeeds but the open-source model fails, revealing systematic gaps in open-source model capabilities.
>
> - **Vision-Language Integration Issues (40% cases)** Involves the inability to distinguish between closely related species, requiring subtle morphological expertise. The models fail in detecting minute visual differences that separate similar organisms. This is due to poor integration of visual features with specialist terminology and knowledge, we observe an improvement in this by finetuning the models on domain specific training data. *Example: Qwen2.5-VL-32B confusing longhorn beetle for June beetle despite similar appearance*
>
> - **Generalist Reasoning Bias (20% cases)**: Qwen2.5-VL-32B model applied broad, non-specific analysis in 20% of the cases rather than focused specialist diagnosis; this results in them listing multiple generic possibilities instead of definitive expert identification.
>
> - **Diagnostic Hedging Bias (25% cases)**: Open source models showed a higher tendency towards hedging and were unable to commit to a decisive specialist-level conclusion/identification response.
>
> **Quantitative Error Pattern Analysis**
> | **Failure Category** | **Percentage** | **Model** |
> |---------------------|----------------|----------------|
> | **Tier 1: Fundamental Domain Challenges (GPT-4.1 Failures)** |
> | Specialist Knowledge Gaps | 35% | GPT-4.1 |
> | Complex Diagnostic Reasoning | 10% | GPT-4.1 |
> | **Tier 2: Open-Source Model Systematic Failures (Qwen2.5-VL-32B vs GPT-4.1)** |
> | Vision-Language Integration Issues | 40% | Qwen2.5-VL-32B |
> | Diagnostic Hedging Bias | 25% | Qwen2.5-VL-32B |
> | Generalist Reasoning Bias | 20% | Qwen2.5-VL-32B |
>
> **Critical Finding:** In joint failures, GPT-4.1 maintains superior reasoning quality in 60% of cases, demonstrating a deeper understanding and integration of visible clues, more accurately mimicking expert reasoning patterns even when incorrect in identifying the exact entity.
>
> **Model Performance on Common & Rare Entities**
>
> The MIRAGE entity distribution follows a long-tail pattern, and we observe that all models perform worse on these rare entities than on the more common ones, as shown in the following table. The disproportionate failure rate on rare species exposes a fundamental challenge in agricultural AI: models must handle thousands of uncommon entities that practitioners encounter in real-world scenarios yet that have minimal representation in training data. Addressing this gap will require specialized techniques beyond standard large-scale training paradigms.
>
> **Performance on Common Entities**
> | Model              	| DeepSeek-R1  | Qwen3-32B   | Phi-4-reasoning  |
> |----------------------- |--------------|-------------|------------------|
> | gpt-4.1            	| 53.1%, 3.15  | 52.7%, 2.84 | 52.2%, 3.20  	|
> | gpt-4.1-mini       	| 42.1%, 2.82  | 41.3%, 2.53 | 40.5%, 2.95  	|
> | Qwen2.5-VL-32B-Instruct| 32.9%, 2.62  | 32.2%, 2.23 | 31.6%, 2.62  	|
> | Qwen2.5-VL-72B-Instruct| 38.1%, 2.65  | 38.4%, 2.27 | 37.1%, 2.63  	|
>
> **Performance on Rare Entities**
> | Model             	| DeepSeek-R1 | Qwen3-32B     	| Phi-4-reasoning  |
> |-----------------------|------------ |-------------------|------------------|
> | gpt-4.1           	| 38.5%, 3.01 | 38.9%, 2.73   	| 35.4%, 3.14  	|
> | gpt-4.1-mini      	| 32.1%, 2.76 | 30.4%, 2.50   	| 26.8%, 2.94  	|
> | Qwen2.5-VL-32B-Instruct| 21.1%, 2.47 | 20.4%, 2.14   	| 18.2%, 2.53  	|
> | Qwen2.5-VL-72B-Instruct| 25.5%, 2.56 | 24.5%, 2.18   	| 22.1%, 2.57  	|
>
> **Effects of Image Quality on Model Performance**
>
> To further understand failure sources, we conducted a systematic analysis of image quality effects on model performance using three key metrics: resolution (total pixels, log-scaled), sharpness (Laplacian variance indicating edge clarity), and RMS contrast (pixel brightness range indicating visual distinctiveness). We examined both linear relationships (Pearson correlation) and monotonic trends (Spearman correlation) between these image quality metrics and model identification accuracy. Our analysis mostly revealed weak correlations between most of the image quality metrics and performance across all evaluated models, confirming that failures arise from model reasoning capabilities and knowledge representation gaps rather than visual preprocessing limitations. This validates that our identified error patterns represent core vision-language understanding challenges rather than low-level perceptual constraints.

---

> > ### Comment · Reviewer_7EZx · 2025-08-04
> >
> > I thank the authors for providing the clarifications, and I'm happy to maintain my accept rating.

---

### Decision · Program_Chairs · 2025-09-18

**Decision:**

Accept (poster)

**Comment:**

This paper introduce MIRAGE, a high-fidelity benchmark for evaluating vision-language models (VLMs) in expert-level agricultural consultations. MIRAGE is publicly released to support the development of vision-language systems that go beyond narrow question answering, toward models capable of engaging in natural interactions that involve ambiguity.

The reviewers all scored this submission positively, and all reviewers had their concerns addressed during the rebuttal. Some of the key strengths include that the benchmark is well-organized, that the analysis is comprehensive and detailed with 22 VLMs, and incorporates user images, metadata (location, time), and naturalistic language from a large set of authentic user-expert conversations.

There were only minor weaknesses, mostly related to improvements to writing and clarifications on experimental details. Some more important weaknesses include the need for professional domain knowledge in a rather niche application (agricultural consultations) and the lack of fully interactive dialogues or user feedback loops.

Nevertheless, given the unanimously positive reviews, I believe the paper warrants presentation at neurips.